# OPEN SET OPPONENT MODELING

## ABSTRACT

In multi-agent systems, opponent modeling aims to reduce environmental uncertainty by modeling other agents. Existing research has utilized opponent information to enhance decision-making capabilities based on various methodologies. However, they generally lack good generalization when opponents adopt an open set of policies. In particular, no work has managed to effectively identify never-before-seen opponents. To address these issues, we propose an end-to-end **O**pen **S**et **O**pponent **M**odeling (**OSOM**) training approach, which for the first time enables explicit identification and response to open set opponents. First, OSOM overcomes the challenges of partial observability by distilling opponent policies into information encodings of controlled agent through representation learning. Second, using randomly generated opponent type embeddings as prompts, OSOM achieves identification of opponent types with variable numbers and semantics by maximizing the probability of selecting the true opponent type embedding via contrastive learning. Finally, with the aggregated opponent type embeddings selected from recent history as context, OSOM learns to best respond to sampled opponents through online reinforcement learning. At test time, OSOM only needs to randomly generate opponent type embeddings as prompts again to achieve effective on-the-fly identification and response to non-stationary open set opponents. Extensive controlled experiments in competitive, cooperative, and mixed environments quantitatively validate the significant advantages of OSOM over existing approaches in terms of identification accuracy and response performance.

## 1 INTRODUCTION

**O**pponent **M**odeling (**OM**) is a long-standing and far-reaching research topic that aims to develop a *self-agent*[1] capable of modeling behaviors, goals, intentions, and other properties about *other agents* (collectively referred to as **opponents**) within a multi-agent system. The purpose of such modeling is to allow the self-agent to flexibly adapt to opponents, thereby reducing its environmental uncertainty and enhancing its decision-making abilities (He et al., 2016; Foerster et al., 2018a; Albrecht & Stone, 2018; Nashed & Zilberstein, 2022; Fu et al., 2022; Yu et al., 2022; Ma et al., 2024; Jing et al., 2024a; 2025b). Existing research has utilized various methodologies, such as Representation Learning (Hong et al., 2018; Grover et al., 2018; Papoudakis et al., 2021), Bayesian Inference (Hernandez-Leal et al., 2016; Zheng

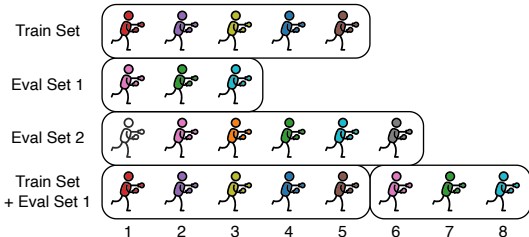

Figure 1: An illustration of **O**pen **S**et **O**pponent (**OSO**) setting. The number and semantics of opponent types in the test opponent set $\Pi^{\text{test}}$ can differ significantly from those in train opponent set $\Pi^{\text{train}}$. In this example, $\Pi^{\text{train}}$ adopts 'Train Set', while $\Pi^{\text{test}}$ may be 'Eval Set 1', 'Eval Set 2', or 'Train Set + Eval Set 1'. OSO poses a formidable challenge for explicit opponent identification.

et al., 2018; DiGiovanni & Tewari, 2021), and Meta-learning (Zintgraf et al., 2021; Al-Shedivat et al., 2018; Kim et al., 2021), to model opponents. Most OM approaches are typically first trained on a fixed *training set of opponent policies* $\Pi^{\text{train}}$ and then benchmarked on a given *testing set of opponent policies* $\Pi^{\text{test}}$ to evaluate their online adaptation abilities (Jing et al., 2024b; 2025a).

---

[1]We refer to the agent under our control in the multi-agent environment as 'self-agent'.

However, when *the set of all possible policies that opponents can adopt is variable*, which we refer to as the **O**pen **S**et **O**pponents (**OSO**s) setting, existing approaches generally exhibit poor generalization. We illustrate the OSO setting in Fig. 1, where different colors denote distinct opponent types. The challenge of OSOs setting is that the opponent types in training and testing can differ significantly in number and semantics. In particular, no work has yet managed to achieve effective identification of opponents that were never seen during training. Existing work either assumes that the opponents in training and testing have very similar policies or implicitly circumvents identification, focusing solely on responding to the opponents. We argue that this lack of identification for unseen opponents largely limits the adaptability of OM approaches. On one hand, a self-agent that can only adapt implicitly often employs a 'blurred' response policy for all potential opponents, making it difficult to precisely exploit the specific opponent it faces. On the other hand, when facing opponents whose policies may change, a self-agent who cannot identify has difficulty quickly detecting shifts in opponent behavior patterns, which leads to low response efficiency.

In this work, we propose a novel **O**pen **S**et **O**pponent **M**odeling (**OSOM**) training approach to address above issues. OSOM is trained end-to-end using the optimization objectives of Representation Learning, *Contrastive Learning* (CL) (Jaiswal et al., 2020), and Online *Reinforcement Learning* (RL), which for the first time achieves **explicit identification** and **effective response** to OSOs.

Without loss of generality, OM typically assumes that its environment is partially-observable. This means that during testing, the self-agent cannot immediately observe information about opponents. To mitigate the challenges of partial observability, we introduce a Representation Learning objective that uses an additional decoder to predict opponent observations and actions, thereby distilling the opponent's policy into a hidden state encoded from the self-agent's observations and actions.

To achieve explicit identification of OSOs, we further propose a CL-based training procedure. Before each iteration, we first sample $K$ opponent policies from a large and diverse Train Set, and then randomly generate $K$ **O**pponent **T**ype **E**mbeddings (**OTE**s) to characterize these $K$ opponent types. Next, we use these OTEs as prompts for the input of model, and autoregressively output a predicted OTE at each timestep. Finally, we use a CL objective during updating to maximize the similarity between the predicted OTEs and the ground truth ones. This procedure makes it possible to identify opponent types that are variable in both number and semantics, and avoids the limitations of using a classifier (Ma et al., 2024), which can only identify a fixed number and semantics of opponents.

Building upon explicit identification, how to effectively respond to opponents is also crucial. Specifically, we first aggregate all OTEs selected from recent episodes to obtain a *compact semantic representation* of the opponent. Next, using this aggregated representation as context, we directly train the self-agent through an online RL objective to learn the best response to the sampled opponent.

By integrating the three objectives, OSOM iteratively samples opponents from the Train Set to train a Transformer-based model end-to-end. Recent research has shown that Transformers pre-trained on high-quality data for decision-making tasks exhibit in-context learning abilities (Lee et al., 2023; Lin et al., 2024). Our model design adopts this idea, which enables the opponent identification and response capabilities learned by OSOM during training to generalize well to testing. Once training is complete, when facing a set of opponents with unseen types in terms of both number and semantics, we only need to randomly generate new OTEs as prompts. This allows OSOM to achieve effective on-the-fly identification and response to non-stationary OSOs as it interacts. Intuitively, OSOM first identifies the opponent based on the self-agent's information, which contains opponent policy semantic. It then summarizes the identification results from a recent period and finally generates the most suitable response according to the determined opponent's identity.

In extensive comparative and ablation experiments across competitive, cooperative, and mixed environments, our approach consistently and significantly outperforms representative OM approaches in terms of both identification accuracy and response performance, fully validating OSOM's effectiveness against OSO. To our best knowledge, OSOM is the first to achieve explicit identification of unseen opponent policies during testing, which prior work has been unable to accomplish.

## 2 PRELIMINARIES

We use an $n$-agent *Partially-Observable Stochastic Game* (POSG) (Hansen et al., 2004; Yang & Wang, 2020) $\langle \mathcal{S}, \{\mathcal{O}^i\}_{i=1}^n, \{\mathcal{A}^i\}_{i=1}^n, \mathcal{P}, \{R^i\}_{i=1}^n, \{\Omega^i\}_{i=1}^n, T, \gamma \rangle$ to formalize the multi-agent envi-

ronment. Here, $\mathcal{S}$ denotes the state space. $\mathcal{O}^i$ is the observation space of agent $i \in [n]$, $\mathcal{O} = \prod_{i=1}^{n} \mathcal{O}^i$ is the joint observation space. $\mathcal{A}^i$ denotes the action space for agent $i$, $\mathcal{A} = \prod_{i=1}^{n} \mathcal{A}^i$ is the joint action space. $\mathcal{P} : \mathcal{S} \times \mathcal{A} \times \mathcal{S} \to [0, 1]$ denotes the transition probabilities. $R^i : \mathcal{S} \times \mathcal{A} \to \mathbb{R}$ denotes the reward function of agent $i$. $\Omega^i : \mathcal{S} \times \mathcal{A} \times \mathcal{O}^i \to [0, 1]$ denotes the agent $i$'s observation function. $T$ is the horizon for each game episode. $\gamma$ is the discount factor.

In line with the tradition in OM, we mark the agent under our control, *i.e.*, the *self-agent*, with the superscript $1$, while the other $n - 1$ agents, regarded as *opponents*, are marked with the superscript $-1$. The joint policy of opponents is denoted as $\pi^{-1}(a^{-1}|o^{-1}) = \prod_{j \neq 1} \pi^j(a^j|o^j)$, where $a^{-1}$ is the joint actions of opponents and $o^{-1}$ is the joint observations of opponents.

Let the opponent's trajectory at timestep $t$ be denoted as $\tau_t^{-1} = (o_0^{-1}, a_0^{-1}, r_0^{-1}, \ldots, o_t^{-1}, a_t^{-1}, r_t^{-1})$, and his complete trajectory be $\tau^{-1} := \tau_{T-1}^{-1} + (o_T^{-1})$. At any episode $h$ and any timestep $t$, *opponent historical trajectories* $\mathfrak{T}_{t(h)}^{-1} := (\tau_{(0)}^{-1}, \ldots, \tau_{(h-1)}^{-1})$ is available to the self-agent.

In OM, the policy of the self-agent can be typically represented as $\pi^1(a^1|o^1, D)$, which adjusts based on the *opponent information data* $D$ (Jing et al., 2024b; 2025a). The data $D$ can either be directly derived from a subset of $\mathfrak{T}^{-1}$ or obtained by learning a representation from $\mathfrak{T}^{-1}$. Under the OSO setting, the number and semantics of opponent types contained within both the *training set of opponent policies* $\Pi^{\text{train}}$ and the *testing set of opponent policies* $\Pi^{\text{test}}$ are variable. Leveraging the training process, the objective of the self-agent is to maximize its expected *return* (*i.e.*, cumulative discounted reward) during testing, *i.e.*, $\max_{\pi^1} \mathbb{E}_{\pi^1 \leftarrow \text{Train}(\Pi^{\text{train}}), \pi^{-1} \sim \Pi^{\text{test}}} \left[ \sum_{t=0}^{T-1} \gamma^t R_t^1 \right]$.

To supplement more background knowledge, we provide a comprehensive review of related work concerning *Opponent Modeling*, *Open Set Learning*, and *RL with Transformers* in Sec. A.

## 3 METHODOLOGY

In this work, we propose an **O**pen **S**et **O**pponent **M**odeling (**OSOM**) training approach, which trains a Transformer-based model end-to-end, achieving **explicit identification** and **effective response** to **O**pen **S**et **O**pponents (**OSO**s). Specifically, the OSO setting presents three main challenges: (1) How to handle the *partial observability* that is inherent in multi-agent environments? (2) How to achieve *explicit identification* of a *variable set of opponent policies*? (3) How to *effectively respond to opponents* based on *historical identification results*? In this section, we will sequentially elaborate on how OSOM addresses these challenges. We provide an overview of the OSOM's training procedure in Fig. 2, and supplement this with the corresponding algorithmic pseudocode in Sec. B. Although we conceptually decompose OSOM into three components to address the three challenges of the OSO setting, in implementation it is a **single Transformer-based model**: all modules share parameters and are trained in **one end-to-end optimization loop** with three coupled objectives, rather than a fragile three-stage pipeline.

### 3.1 OPPONENT POLICY DISTILLATION WITH REPRESENTATION LEARNING

Consistent with the POSG formalization in Sec. 2, multi-agent environments are typically partially-observable. Under these conditions, the self-agent's inability to instantaneously access the opponent information during testing significantly impedes the effective distinction of different opponent types. However, it is a reasonable assumption that opponent data is often accessible during training (Papoudakis et al., 2021; Gronauer & Diepold, 2022). This corresponds to a standard *Centralized-Training-with-Decentralized-Execution* (CTDE) assumption: during training we may log opponents' trajectories, but at test time OSOM only observes its own local information. Consequently, we can manage to distill the opponent policy into the self-agent's representation, thereby enabling the self-agent to more effectively distinguish various opponent policies solely through its own observation-action sequence.

Enlightened by this, we propose to distill the opponent policy into the self-agent's information encoding via Representation Learning. We employ an encoder $f$ to encode the self-agent's observation-action tuple, yielding a corresponding $d$-dimensional latent state $e = f(o^1, a^1) \in \mathbb{R}^d$. Without loss of generality, assuming a continuous observation space and a discrete action space, we introduce an auxiliary decoder $g$ that uses $e$ as input to predict the opponent's observation-action tuple at the

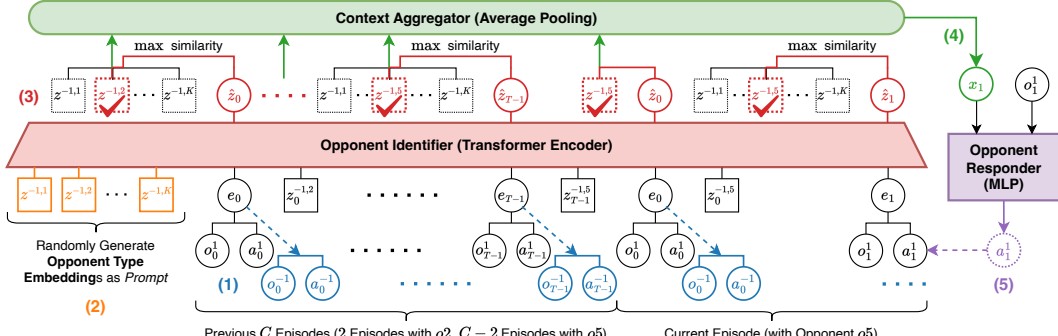

Figure 2: Training procedure of OSOM. Taking timestep 1 of the current episode against the opponent policy $o5$ as an example, we illustrate the five steps of OSOM's end-to-end training: (1) *Opponent Policy Distillation* (Sec. 3.1): Distill the opponent policy into the latent state $e$ of the self-agent's observation-action encoding via a Representation Learning objective that predicts the opponent's observation and action. (2) *Embedding Generation and Prompting* (Sec. 3.2): Sample $K$ distinct opponent policies $\{\pi^{-1,k}\}_{k=1}^{K}$ from the Train Set, randomly generate $K$ corresponding OTEs $\{z^{-1,k}\}_{k=1}^{K}$, and use this set as the prompt for the input sequence. (3) *Explicit Identification* (Sec. 3.2): Use the **Opponent Identifier** to output the predicted OTE $\hat{z}$, maximizing its similarity with the ground truth OTE via a CL objective. (4) *Context Aggregation* (Sec. 3.3): Employ the **Context Aggregator** to aggregate all historically selected OTEs from the previous $C$ episodes, yielding the compact semantic representation $x$. (5) *Best Response Learning* (Sec. 3.3): Use the **Opponent Responder** to output the self-agent's action, using the self-agent's observation as input along with $x$ as context, and employ an online RL objective to learn the best response to the $o5$.

same timestep, specifically $o^{-1} \leftarrow g^{\text{obs}}(e), a^{-1} \leftarrow g^{\text{act}}(\cdot|e)$. Supposing that during training, we consistently sample $K$ opponent policies $\Pi^{\text{train}} = \{\pi^{-1,k}\}_{k=1}^{K}$ from a large and diverse Train Set (comprising more than $K$ policies), the objective function can be written as:

$$\mathcal{J}_{\text{distill}} = \mathbb{E}_{\pi^{-1} \in \Pi^{\text{train}}} \left[ \frac{1}{T} \sum_{t=0}^{T-1} \left( -(g^{\text{obs}}(e_t) - o_t^{-1})^2 + \log g^{\text{act}}(a_t^{-1}|e_t) \right) \right]. \tag{1}$$

Where $e_t = f(o_t^1, a_t^1)$. This method ensures that, even when opponent information is inaccessible during testing, the trained encoder can utilize the self-agent's information to implicitly infer and incorporate the opponent policy details, thus mitigating the challenges posed by partial observability.

In practice, the distillation term $J_{\text{distill}}$ is an auxiliary objective: in the ablation 'OSOM w/o Distill Loss' we set its coefficient to 0, remove all opponent trajectories from training, and still obtain a functioning identification-and-response agent trained only with $J_{\text{identify}}$ and $J_{\text{respond}}$ (Sec. 4.2).

## 3.2 OPPONENT IDENTIFICATION WITH CONTRASTIVE LEARNING

In the domain of *Contrastive Learning* (CL) (Weng, 2021), work such as CLIP (Radford et al., 2021) has demonstrated the feasibility of text classification for images unseen during training by maximizing the similarity between image and text embeddings that share the same semantics. Motivated by this, we introduce a novel training procedure based on CL that enables the explicit identification of opponent types, where both the number and semantics of these types can be variable.

Before each training iteration, we first sample $K$ opponent policies $\Pi^{\text{train}}$ from the Train Set mentioned in Sec. 3.1. Subsequently, we randomly generate $K$ corresponding $d$-dimensional **O**pponent **T**ype **E**mbeddings (OTEs) $\mathcal{Z}^{\text{train}} = \{z^{-1,k} \in \mathbb{R}^d\}_{k=1}^{K}$ to uniquely characterize these $K$ distinct opponent policies. The core intuition behind using random OTEs is to eliminate any prior knowledge regarding the structural space of the opponent types within our model. This design choice is motivated by the observation that employing learnable OTEs becomes impractical when encountering novel opponent types unseen during training. A new opponent type would lack a pre-trained embedding, and assigning an arbitrary embedding could introduce severe domain shift because the model would be unable to identify it. The utilization of random OTEs ensures that the model does not

rely on extracting any information from the embedding itself, but rather on interpreting the context provided by the historical interactions with the environment.

Furthermore, we encourage approximate pairwise orthogonality among the $K$ OTEs by sampling random unit vectors on the sphere and, when the embedding dimension $d$ is at least $K$, optionally applying Gram-Schmidt orthogonalization. This is inspired by Elhage et al. (2022); Ganesh et al. (2023). The unit sphere standardizes the scale of the embeddings, and near-orthogonality makes it easier for the model to adjust the probability assigned to each opponent type, since a predicted vector $\hat{z}$ can align strongly with one OTE without inadvertently aligning with others. This improves the conditioning of the contrastive objective in Eq. (2) by sharpening the contrastive signal for OSOM; when $K > d$, one can simply use random unit OTEs without enforcing exact orthogonality.

During the rollout process of training, we instantiate a Transformer-based model (Vaswani et al., 2017), which we term the **Opponent Identifier**. This model utilizes the above $K$ OTEs as a prompt for the input sequence and, at each timestep, takes the latent state $e$ of the self-agent's observation and action as input to autoregressively output a predicted OTE $\hat{z} \in \mathbb{R}^d$. The inclusion of the sequence of all legal OTEs prior to the main input serves to prevent the model from lacking awareness of the current structural space of opponent types. In addition, by directly outputting the embedding $\hat{z}$ instead of predicting a probability distribution over the opponent types, the model achieves independence from both the number and the permutation order of the available opponent types.

During the update process of training, we employ an objective of CL to maximize the similarity between the predicted OTE $\hat{z}$ and the ground truth OTE $z^{-1,j}$, while simultaneously minimizing the similarity between $\hat{z}$ and all other OTEs $z^{-1,k \neq j}$. Specifically, this objective can be formulated as:

$$\mathcal{J}_{\text{identify}} = \mathbb{E}_{z^{-1,j} \in \mathcal{Z}^{\text{train}}} \left[ \frac{1}{T} \sum_{t=0}^{T-1} \log \frac{\exp\left(\hat{z}_t \cdot z_t^{-1,j}/\kappa\right)}{\sum_{k=1}^{K} \exp\left(\hat{z}_t \cdot z_t^{-1,k}/\kappa\right)} \right]. \tag{2}$$

Where $\kappa$ represents the temperature coefficient. This procedure draws inspiration from CL methods such as CLIP, InfoNCE (Oord et al., 2018), and SupCon (Khosla et al., 2020), bypassing the limitation of traditional classifiers which can only identify opponent types with fixed numbers and semantics (Sukthankar & Sycara, 2007; Iglesias et al., 2008; Bombini et al., 2010; Ma et al., 2024). During the rollout processes of both training and testing, we probabilistically sample from all legal OTEs as the *selected OTE* $z^{\text{sel}}$ according to the following probability distribution:

$$\forall l \in [K], \quad P(z^{-1,l}) = \text{softmax}(\hat{z} \cdot z^{-1,l}) = \frac{\exp\left(\hat{z} \cdot z^{-1,l}\right)}{\sum_{k=1}^{K} \exp\left(\hat{z} \cdot z^{-1,k}\right)}. \tag{3}$$

### 3.3 OPPONENT RESPONSE WITH ONLINE REINFORCEMENT LEARNING

Despite the importance of explicit opponent identification, the challenge of how to effectively respond to the opponent based on the history of identification results is equally critical. Adhering to the best practices of OM, we assume that the opponent during testing is both *unknown* and *non-stationary*. Here, 'unknown' signifies that the opponent's true policy is black-box and inaccessible. 'Non-stationary' indicates that the opponent may switch policies in some manner as the interaction progresses. Under these conditions, relying solely on the most recent identification result or using the entirety of the historical identification results could potentially lead to model confusion.

To address the above issues, we introduce a sliding window-based **Context Aggregator** to aggregate the historically generated identification results. Assuming the current timestep is $t$ in episode $h$, we employ Average Pooling (Gholamalinezhad & Khosravi, 2020) to aggregate all historically selected OTEs from the current episode up to $t - 1$ and the previous $C$ episodes, *i.e.*, $(z_{0(h-C)}^{\text{sel}}, \cdots, z_{T-1(h-1)}^{\text{sel}}, z_{0(h)}^{\text{sel}}, \cdots, z_{t-1(h)}^{\text{sel}})$, yielding a *compact semantic representation* $x_{t(h)}$. The manner in which we select the opponent type is detailed in Eq. (3). This aggregation allows us to confine the identification results primarily to the current opponent, thereby mitigating interference introduced by other potential opponents.

Subsequently, using the aggregated representation $x$ as context, we propose an online RL (Li, 2017; Schulman et al., 2017) objective to train an **Opponent Responder** that learns the best response to the opponents. The resulting trained self-agent policy can be concisely denoted as $\pi^1(a^1|o^1, x)$.

Given the $K$ sampled opponent policies $\Pi^{\text{train}}$ (same as in Sec. 3.1), the objective can be written as:

$$\mathcal{J}_{\text{respond}} = \mathbb{E}_{\pi^{-1} \sim \Pi^{\text{train}}, \pi^1} \left[ \sum_{t=0}^{T-1} \gamma^t R_t^1 \right]. \tag{4}$$

The Opponent Responder is trained with a standard on-policy RL algorithm PPO (Schulman et al., 2017), which provides a policy-gradient estimator of $\nabla \mathcal{J}_{\text{respond}}$; in practice we optimize a PPO surrogate objective jointly with the auxiliary losses rather than differentiating through the environment dynamics. To summarize, OSOM iteratively samples opponents from the Train Set and end-to-end trains a Transformer-based model. Its final optimization objective is:

$$\max \alpha_1 \mathcal{J}_{\text{distill}} + \alpha_2 \mathcal{J}_{\text{identify}} + \alpha_3 \mathcal{J}_{\text{respond}}. \tag{5}$$

Where $\alpha_1, \alpha_2, \alpha_3$ are tunable coefficients. Upon completion of training, when faced with an opponent whose policy is drawn from the set of $M$ unknown policies $\Pi^{\text{test}} = \{\pi^{-1,m}\}_{m=1}^M$, OSOM only needs to randomly generate $M$ corresponding OTEs $\mathcal{Z}^{\text{test}} = \{z^{-1,m} \in \mathbb{R}^d\}_{m=1}^M$ as a prompt. This allows for effective on-the-fly identification and response to the non-stationary opponent as interactions proceed. The testing procedure of OSOM is detailed in Sec. C. We adopt Transformer as the backbone for our model, which further facilitates OSOM to generalize the learned identification and response abilities from the training opponents to the testing opponents. This design choice is enlightened by numerous studies demonstrating In-Context RL abilities of Transformer in decision-making tasks (Wang et al., 2016; Duan et al., 2016; Laskin et al., 2023; Grigsby et al., 2023).

Crucially, the Responder never memorizes fixed semantics for individual OTE vectors. Because we regenerate a fresh random OTE set whenever we sample a new opponent subset during training and again at test time, the only stable semantics lie in the *geometry* of the codebook: for a given interaction history, the Identifier learns to assign high similarity to exactly one OTE and low similarity to the others, and the Responder learns to map the resulting aggregated context $x_t$ to a good response. Re-sampling the OTE set at test time therefore corresponds to an orthogonal rotation of the label space to which the jointly trained Identifier–Responder pair is invariant.

## 4 EXPERIMENTS

In this section, Sec. 4.1 provides detailed experimental setups. Sec. 4.2 poses a series of questions and provides empirical results to answer them, with the aim of analyzing the effectiveness of OSOM.

### 4.1 EXPERIMENTAL SETUP

**Environments.** We consider three partially-observable environments widely used in MARL. For further details of these environments, see Sec. D.

- Kuhn Poker (KP) (Hoehn et al., 2005; Kuhn, 2016): A two-player zero-sum (competitive) game with a discrete state space. In KP, the self-agent's objective is to maximize chip gain while minimizing chip loss, with the opponent having the same objective. No player knows the other's hand until one player folds or both proceed to a showdown. KP encompasses the key challenges found in real-world Poker, where strategic deception or conservatism must be learned.
- Partially-Observable Overcooked (POO) (Carroll et al., 2019; Ma et al., 2024): A two-player cooperative game with a discrete state space. In POO, the self-agent aims to coordinate closely with its partner to complete a series of sub-tasks and serve dishes. This version of POO introduces partial observability and multiple recipes, requiring the self-agent to both infer the global state of the kitchen from local observations and learn to deduce its partner's recipe preferences.
- Predator-Prey with Watchtowers (PPW) (Lowe et al., 2017; Ma et al., 2024): A four-player mixed-incentive game with a continuous state space. In PPW, the self-agent plays the role of a predator, whose objective is to cooperate with its fellow predator to capture the two preys as many times as possible. The challenge of PPW lies in the fact that predators can only gain momentary global observation by actively touching watchtowers. Furthermore, the self-agent must accurately model the movement preferences of every agent, both teammates and opponents.

**Baselines.** We select the following representative OM approaches as baselines. The training recipes for all the OM approaches are supplemented in Sec. F.

- **PPO** (Schulman et al., 2017): a non-recurrent agent that receives only the current self observation as input (no cross-episode context and no opponent information), serving as a naive baseline.
- **Generalist**: A plain recurrent policy with access to cross-episode contexts. These contexts encompass all of the self-agent's historical observations and actions.
- **LIAM** (Papoudakis et al., 2021): Use the observations and actions of the self-agent to reconstruct those of the opponent through an auto-encoder, thereby embedding the opponent policy into a latent space to assist in the self-agent's response learning against opponents.
- **LIAMX** (Papoudakis et al., 2021): A variant of LIAM with cross-episode contexts. It further extends the horizon over which LIAM can infer the opponent's policy from its own information.
- **LILI** (Xie et al., 2021): Model the observation-action-reward-next observation transitions observed by self-agent in the last episode, implicitly encoding the opponent as environmental dynamics to enhance the self-agent's policy optimization.
- **GSCU** (Fu et al., 2022): 1) Employ offline policy embedding learning to train well-structured representations 2) Utilize offline conditional RL to learn responses to various opponents 3) Apply an online multi-armed bandit algorithm to balance conservative or greedy self-agent policies.
- **PACE** (Ma et al., 2024): Introduce an opponent identification reward to maximize the mutual information between the opponent's policy and the self-agent's cross-episode trajectory, thereby encouraging self-agent actions that aid in identifying the opponent's behavioral patterns.
- **PACE-TF** (Ma et al., 2024): A variant of PACE in which the original GRU-based context encoder is replaced with the official Transformer architecture from the PACE paper, while keeping all other components, training recipes, and hyperparameters unchanged. This baseline isolates the effect of the Transformer backbone from OSOM's open-set identification mechanism.

**Opponent Policies.** We design a relatively large and diverse *opponent pool* for training and testing OM approaches against OSOs. To ensure both policy diversity and semantic interpretability, we incorporate human priors in constructing the opponent pool. For KP, we follow Hoehn et al. (2005), using two parameters to parameterize the opponent's policy space after eliminating dominated strategies. For POO and PPW, we follow Ma et al. (2024) by using rule-based methods to construct opponent policies with clearly distinct preferences. After constructing the opponent pool, we selected a subset of these policies to form a Train Set, and used all remaining policies to constitute an Eval Set. Concretely, we construct dozens of distinct opponent policies in each domain and then split them into disjoint Train and Eval pools, so that every evaluation trajectory involves behaviors that never appeared in Train while still exhibiting meaningful semantic diversity. We have supplemented more detailed process of constructing the opponent pool in Sec. E.

**Training and Testing Protocols.** We train all the OM approaches for the same number of iterations. During training, in each iteration, we randomly sample $K$ opponent policies from the Train Set to instantiate $\Pi^{\text{train}}$. The final checkpoints obtained from training all approaches are used to test against unknown non-stationary opponents (as explained in Sec. 3.3) for the same number of episodes. Non-stationarity is defined as the opponent switching its policy by sampling from $\Pi^{\text{test}}$ every $H$ episodes. In KP we set the opponent-switch frequency to $H = 20$ episodes, and in POO and PPW we use $H = 5$ episodes. These values correspond to a moderate level of non-stationarity: the agent must track and adapt to switches that are neither too frequent (near-i.i.d. noise) nor too rare (almost stationary). All figures and tables report the *mean* and *standard deviation* of the results averaged over 5 random seeds. All the hyperparameters are detailed in Sec. G. Unless otherwise stated, all learning curves are plotted against **total environment timesteps** rather than training iterations, so that the sample efficiency of methods with different update-to-data ratios can be compared fairly.

## 4.2 EMPERICAL ANALYSIS

**Question 1.** *Can OSOM effectively adapt to OSOs with different $\Pi^{\text{test}}$ configurations?*

Fig. 3 shows the overall performance of all OM approaches against non-stationary opponents whose policies are sampled from three distinct $\Pi^{\text{test}}$ configurations. We establish three types of $\Pi^{\text{test}}$ from which the non-stationary opponent could sample policies: (1) Train Set; (2) Eval Set, where all policies are unseen during training; and (3) All Set, which is the union of the Train Set and the Eval Set. The latter two configurations are specifically designed to examine the generalization ability to opponent types that are unseen in terms of both number and semantics.

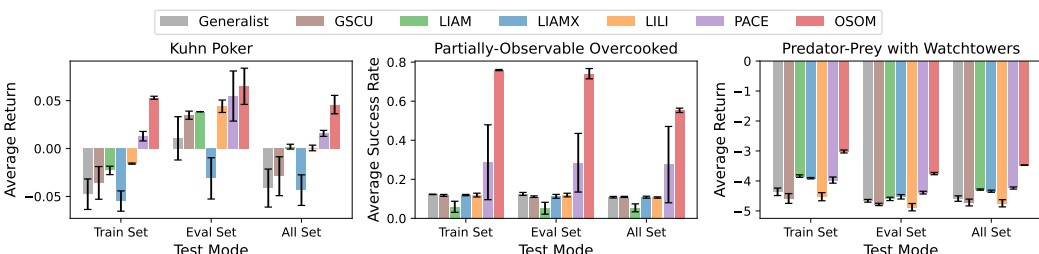

Figure 3: The overall results of testing OM approaches against unknown non-stationary OSOs employing different $\Pi^{\text{test}}$ configurations. Specifically, the performance for KP and PPW is measured by the average return, while POO's performance is measured by the average success rate.

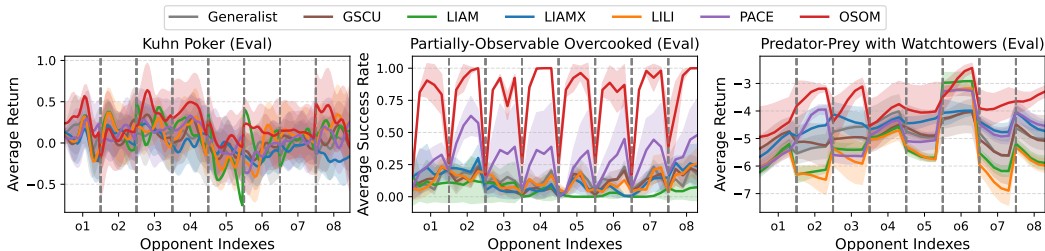

Figure 4: Detailed per-episode performance curves for OM approaches when tested against each specific opponent policy, where $\Pi^{\text{test}}$ is set to Eval Set. We use $o1, o2, \ldots, o8$ to denote the indexes of the different opponent types, respectively.

It can be observed that in all three environments, OSOM achieves a higher average return or success rate compared to other baselines under various $\Pi^{\text{test}}$ configurations. This is particularly evident in the challenging POO, where most other approaches fail to work properly. These consistent results indicate that the overall design of OSOM enables it to effectively respond to non-stationary OSOs, enabling it to robustly adapt to seen opponent types and successfully generalize to unseen ones.

**Question 2.** *Can OSOM effectively respond to every specific policy adopted by OSOs?*

In Fig. 4, we present the smoothed per-episode performance curves for all approaches against non-stationary opponents whose policies are sampled from the Eval Set. Specifically, each environment displays the average results against 8 unseen opponent policies from the Eval Set. The frequency $H$ for the non-stationary opponent to switch policies is 20, 5, and 5 in KP, POO, and PPW, respectively. Similar to the observation in Fig. 3, OSOM generally exhibits better results against every specific opponent policy compared to other baselines. Notably, OSOM shows a trend of gradually increasing performance against most opponent policies in POO and PPW, which suggests that it may possess a degree of test-time gradient-free learning capability.

**Question 3.** *Can OSOM make explicit identification of OSOs feasible?*

Table 1 shows the accuracy metric for identifying non-stationary OSOs under various $\Pi^{\text{test}}$ configurations during testing. For OSOM, we use the Opponent Identifier to select the OTEs according to Eq. (3), and then compute the proportion of times it selects the true opponent type out of all prediction attempts. Furthermore, we introduce **Random Guess** as a comparative baseline. In contrast, the other OM approaches are all unable to explicitly predict the opponent type. For instance, while PACE uses a classifier to predict opponent type during training, all three $\Pi^{\text{test}}$ configurations we established contain a greater number of opponent types (and semantics) than those used during training. Observation reveals that, under this challenging open-set protocol, OSOM consistently achieves substantially higher identification accuracy than Random Guess across all environments, especially on the unseen opponents in the Eval Set. For example, in the most challenging All Set OSOM obtains identification accuracies in the range of 9.75%-36.52% while Random Guess remains around 2%-3.7%, *i.e.*, improvements of approximately 2-15× over chance. Given the large label spaces (up to 40-50 opponent types), the open-set and non-stationary test protocol, and partial observability, such 2-15× gains already indicate substantial and practically useful identification

Table 1: Per-policy identification accuracy statistics against non-stationary OSOs under various $\Pi^{\text{test}}$ configurations during testing. The Accuracy, expressed as a percentage, measures the quality of opponent type identification for each approach (higher is better). Results marked with '✗' indicate that a meaningful evaluation is not possible.

| Avg. Acc. (%) ↑ of Approaches | Kuhn Poker | | | Partially-Observable Overcooked | | | Predator-Prey with Watchtowers | | |
|---|---|---|---|---|---|---|---|---|---|
| | Train Set | Eval Set | All Set | Train Set | Eval Set | All Set | Train Set | Eval Set | All Set |
| OSOM | $13.27 \pm 0.09$ | $24.09 \pm 0.36$ | $9.75 \pm 0.11$ | $32.42 \pm 4.24$ | $23.33 \pm 1.32$ | $14.77 \pm 1.40$ | $38.99 \pm 1.33$ | $34.87 \pm 0.05$ | $36.52 \pm 0.44$ |
| Random Guess | 2.50 | 10.00 | 2.00 | 5.56 | 11.11 | 3.70 | 6.25 | 4.17 | 2.50 |
| Other OM Approaches | ✗ | ✗ | ✗ | ✗ | ✗ | ✗ | ✗ | ✗ | ✗ |

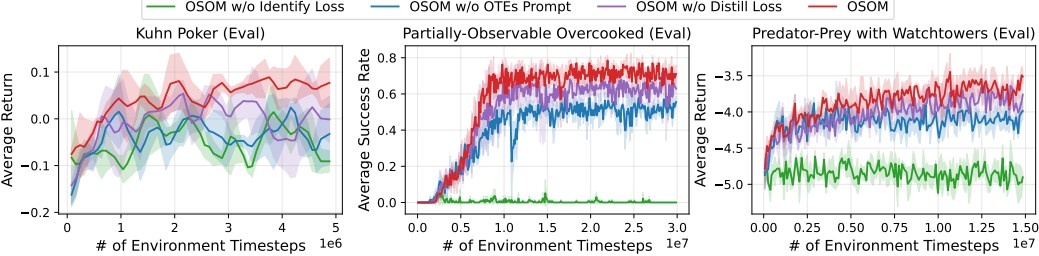

Figure 5: Average performance curves of the various ablation variants of OSOM against all opponent policies in the Eval Set during training. We select checkpoints at equal intervals during the training process to conduct this evaluation.

ability. Combining this with the results in Fig. 3, we hypothesize that OSOM's strong capability for responding to OSOs is founded upon its effective identification of opponent types.

**Question 4.** *Do all key design choices in OSOM contribute positively?*

We design the following ablation variants of OSOM: (1) **OSOM w/o Identify Loss**: Remove the opponent identification objective described in Eq. (2), meaning $\alpha_2 = 0$; (2) **OSOM w/o OTEs Prompt**: Remove the OTEs prompt part of the Opponent Identifier's input sequence; (3) **OSOM w/o Distill Loss**: Remove the policy distillation objective described in Eq. (1), meaning $\alpha_1 = 0$.

Fig. 5 presents the average performance curves against all opponent policies in the Eval Set during the training process. We observe that 'OSOM w/o Identify Loss' is essentially unable to function properly, suggesting that our Contrastive Learning-based opponent identification method plays a crucial role in effective opponent adaptation. Furthermore, 'OSOM w/o OTEs Prompt' and 'OSOM w/o Distill Loss' generally show performance degradation of varying degrees compared to the full OSOM. These observations support the conclusion that both using OTEs to prompt the Opponent Identifier regarding the structural space of opponent types, and distilling the opponent policy into the self-agent's information encoding, contribute positively to OSOM's performance.

**Question 5.** *Compared to other OM approaches, does OSOM have higher training efficiency?*

In Fig. 6, we show the average performance curves for each OM approach against all opponent policies in the Train Set and the Eval Set during the training process. Specifically, we test the performance by periodically evaluating checkpoints during training of the OM approaches. It is observed that OSOM consistently surpasses the other baselines and achieves superior opponent adaptation results more efficiently, regardless of whether the opponent type has been previously encountered or not. This is particularly true in the challenging P00, where the other baselines consistently fail to learn a workable self-agent policy. This demonstrates that OSOM possesses a higher algorithmic training efficiency, given that it requires fewer iterations to reach the same performance.

As expected, the naive PPO baseline performs significantly worse than all OM approaches, especially on the unseen Eval opponents, confirming that modeling cross-episode information and opponent types is crucial for open-set adaptation. The additional baseline PACE-TF shows that merely upgrading the context encoder from a GRU to a Transformer does not close the gap to OSOM; the performance gains stem primarily from OSOM's open-set identification and context aggregation mechanisms rather than from a stronger sequence model alone.

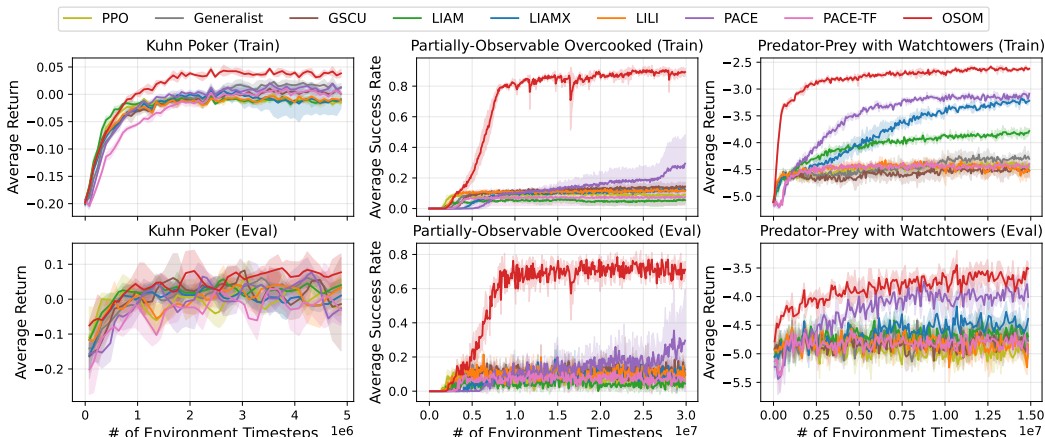

Figure 6: Average performance curves of various OM approaches against different opponent sets during the training process. The upper subplot shows results against the Train Set, and the lower subplot shows results against the Eval Set.

**Additional Experiments.** We provide further experiments in Sec. I to investigate (1) the feasibility of multiple self-agents control for OSOM in CTDE settings; (2) the effect of opponent switch frequency $H$; (3) how does OSOM behave in the degenerate case using a single OTE.

## 5 DISCUSSION

**Summary.** This paper introduces OSOM, a novel end-to-end training approach that enables explicit identification and effective response to unseen OSOs in multi-agent systems—an ability not achieved by prior work. OSOM addresses the three core challenges of OSO modeling by combining Representation Learning (to handle partial observability), CL (for explicit, variable-type identification), and online RL (to learn the best response). Experiments in various multi-agent environments show that OSOM significantly outperforms existing approaches in both identification accuracy and response performance against non-stationary OSOs, confirming its robustness and generalizability.

**Limitations and Future Work.** While OSOM already achieves explicit identification and strong responses to open-set opponents, it still has several limitations. First, our current experiments adopt a CTDE assumption: during training we log opponents' trajectories and use them in the distillation loss $J_{\text{distill}}$ to improve representation quality under partial observability, whereas at test time the agent only observes its own information. Ablations show that an OSOM variant trained without $J_{\text{distill}}$ (using only $J_{\text{identify}}$ and $J_{\text{respond}}$) remains functional but with reduced sample efficiency, suggesting future work on fully decentralized or unsupervised alternatives that remove this privileged training signal. Second, in the OSO benchmarks we set the prompt size $M$ equal to the number of test opponents and generate approximately orthogonal $d$-dimensional OTEs with $d \geq M$; more generally, $M$ and $d$ should be understood as capacity hyperparameters that control how many distinct opponent behaviors can be separated within a single prompt. Scaling OSOM to regimes with very large or unknown numbers of opponent types will require choosing $M$ as a design budget, allowing behaviors to be clustered into at most $M$ latent types, increasing $d$, or exploring richer OTE parameterizations (*e.g.*, learnable or hierarchical codebooks) that relax the need for strict orthogonality.

Our empirical study further focuses on three standard multi-agent benchmarks (KP, POO, PPW) built from structured, semantically diverse pools of opponent policies, and on settings where OSOM controls a single self-agent against piecewise-stationary opponents. Although our formulation and algorithms directly extend to centralized control of a team of agents (treating the team as a joint self-agent) and to CTDE-style decentralized control with one OSOM instance per controllable agent, a systematic evaluation of such multi-agent extensions is left for future work. Future work also includes developing variants that can cope with fully learning, continuously evolving opponents beyond the piecewise-stationary regime considered here.

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

# A   RELATED WORK

**Opponent Modeling.**   Prior OM work generally falls into two major categories based on their focus: (1) those *focused on training* and (2) those *focused on testing*.

The first category focuses on learning high-level knowledge of responding to different opponents during training and generalizing it to testing. Some employ the idea of *Representation Learning* (Jaiswal et al., 2020), aiming to learn high-quality representations of opponent policies during training to aid in policy optimization (He et al., 2016; Hong et al., 2018; Grover et al., 2018; Papoudakis & Albrecht, 2020; Zintgraf et al., 2021; Papoudakis et al., 2021; Papoudakis & Albrecht, 2020; Jing et al., 2024a; Ma et al., 2024). *Non-gradient Meta-learning* methods (Duan et al., 2016; Wang et al., 2016) are also utilized by some researchers, who seek to leverage recurrent architectures to acquire knowledge about the intrinsic policy structure of each opponent and the differences between them throughout training (Wang et al., 2016; Zintgraf et al., 2021).

The second category centers on updating the trained opponent model during testing to reason and to respond against the current opponent. Some adopt the idea of *Bayesian Inference* (Bernardo & Smith, 2009), attempting to detect or infer the opponent's policy in real-time and generate accurate responses accordingly (Bard et al., 2013; Rosman et al., 2016; Hernandez-Leal et al., 2016; Zheng et al., 2018; Sessa et al., 2020; DiGiovanni & Tewari, 2021; Fu et al., 2022; Lv et al., 2023). Others utilize *Gradient-based Meta-learning* principles (Finn et al., 2017), capitalizing on the well-initialized solutions acquired in the parameter space during training to enable fast adaptation and fine-tuning for new opponents encountered during testing (Al-Shedivat et al., 2018; Kim et al., 2021; Wu et al., 2022; Banerjee et al., 2023). Research pertaining to the *Shaping of Opponents' Learning* (Foerster et al., 2018a;b; Letcher et al., 2019; Kim et al., 2021; Lu et al., 2022; Willi et al., 2022; Zhao et al., 2022; Fung et al., 2023; Souly et al., 2023; Duque et al., 2024; Qiao et al., 2024; Aghajohari et al., 2024; Zhou et al., 2024a), *Recursive Reasoning* (Wen et al., 2019; 2021; Dai et al., 2020; Yuan et al., 2020; Yu et al., 2022), and the *Theory of Mind* (Von Der Osten et al., 2017; Rabinowitz et al., 2018; Raileanu et al., 2018; Yang et al., 2019; Li et al., 2023a; Kosinski, 2023; Huang et al., 2024; Han et al., 2025) also falls within this category.

In prior work, it is typically assumed that the sets of training and testing opponents are fixed, and the testing opponent distribution often **lies around** the training opponent distribution (*i.e.*, testing opponents do not differ significantly from those in training). In contrast, this work adopts an OSO setting, where both the training and testing opponent sets are variable, and the types of policies they contain can have substantial differences in number and semantics. Under this challenging setting, existing approaches are unable to explicitly identify unseen opponents and, therefore, struggle to effectively respond to them. We propose a novel OSOM training approach to overcome this challenge.

Concurrent to our work, LOSI (Anonymous, 2025) proposes a latent opponent-strategy identification framework in cooperative SMAC-Hard, learning unsupervised embeddings of opponent scripts with a GRU encoder and a prototype-based contrastive objective that is integrated into QMIX (Rashid et al., 2020). However, LOSI assumes a fixed, closed set of opponent scripts shared between training and testing. In contrast, OSOM targets **open-set**, potentially non-stationary opponents with supervised OTEs and random OTE prompting, provides explicit per-type identification metrics, and is instantiated as a generic opponent-aware policy module that can be combined with different RL algorithms. We view LOSI as complementary, and exploring combinations of its unsupervised encoders with OSOM's open-set OTE framework is an interesting direction for future work.

**Open Set Learning.**   Our work is closely related to the field of *Open Set Recognition* (OSR) (Scheirer et al., 2014; Bendale & Boult, 2014; Geng et al., 2018; Yoshihashi et al., 2018; Mahdavi & Carvalho, 2021; Boult et al., 2019; Halász et al., 2023; Li et al., 2024a; Lang et al., 2024; Wang et al., 2024a; Miller et al., 2024; Bahavan et al., 2025). Traditional machine learning models mostly follow the 'closed world' assumption, where all classes encountered during testing are known during training. OSR aims to break this assumption, enabling models to not only accurately classify known classes but also effectively recognize and reject unknown class samples that were never seen during training. Moreover, many of these works have adopted the idea of *Contrastive Learning* (CL) (Xu et al., 2023; Li et al., 2024b; Zhou et al., 2024b).

Similar to the problem setting of OSR, the OSO we formalized assumes that the set of all possible policies opponents can adopt is variable, and the number and semantics of opponent policies in both

the training and testing sets can be significantly different. However, unlike the goal of traditional OSR, we do not simply hope to detect and reject '*unknown classes*.' Instead, we aim to achieve effective discrimination and recognition even for all '*unknown opponent classes*.'

Unlike prior OM work that adapts or generalizes to unseen opponents implicitly, our OSOM is the first to introduce an explicit identification mechanism. This concept also resonates with the more advanced field of *Open World Learning* (OWL) (Langley, 2020; Team et al., 2021; Zhu et al., 2022b;a; Kejriwal et al., 2024; Irfan et al., 2024; Brilhador et al., 2025; Zheng et al., 2025), which not only requires recognizing unknowns but also emphasizes incrementally learning new knowledge. Therefore, OSOM can be seen as a crucial step toward applying the ideas of OWL to multi-agent decision-making, leveraging the methodology of CL and the in-context generalization abilities of the Transformer model to achieve more effective opponent adaptation in open environments.

**RL with Transformers.** There has been a growing research interest in leveraging Transformers for decision-making tasks by reconceptualizing the problem as sequence modeling (Chen et al., 2021; Janner et al., 2021; Yang et al., 2023a; Li et al., 2023b; Yamagata et al., 2023; Wu et al., 2024; Wang et al., 2024b; Hu et al., 2024; Zhuang et al., 2024). Pioneering work like *Decision Transformer* (DT) (Chen et al., 2021) and Trajectory Transformer (Janner et al., 2021) introduced a novel paradigm, demonstrating that decision-making can be addressed through *Return-Conditioned Supervised Learning* (Brandfonbrener et al., 2022; Yang et al., 2023b). Specifically, DT utilizes a causal Transformer trained on offline data to predict action sequences based on desired returns. Subsequent research has built upon this foundation by exploring improvements such as more advanced conditioning techniques (Furuta et al., 2022; Paster et al., 2022) and architectural enhancements (Villaflor et al., 2022). Another research direction is the application of Transformers' versatility and scalability to multi-task learning (Lee et al., 2022; Reed et al., 2022).

When pre-trained for decision-making in some contextual manners, Transformers also demonstrate a robust ability for *In-Context RL* (ICRL) (Moeini et al., 2025; Wang et al., 2016; Duan et al., 2016; Grigsby et al., 2023; Dorfman et al., 2021; Mitchell et al., 2021; Pong et al., 2022; Laskin et al., 2023; Lee et al., 2023; Sinii et al., 2023; Lin et al., 2024; Grigsby et al., 2024; Mukherjee et al., 2024; Chen & Paternain, 2024; Dong et al., 2024; Schmied et al., 2024; Son et al., 2025; Polubarov et al., 2025; Tarasov et al., 2025; Chen et al., 2025; Tajwar et al., 2025; Liu et al., 2025; Wu et al., 2025). For example, Laskin et al. (2023) used an autoregressive supervised learning approach to distill the sub-traces of a single-task RL algorithm into a single model that is not tied to any specific task. Similarly, Lee et al. (2023) used supervised pretraining to show the ICRL capabilities of these models, both empirically and theoretically. Building on this, Lin et al. (2024) introduced a theoretical framework to analyze and explain the underlying principles and conditions required for ICRL to work. Moreover, the works of Dorfman et al. (2021); Mitchell et al. (2021); Li et al. (2020); Pong et al. (2022); Tarasov et al. (2025) are specifically centered on *Offline Meta-RL* and incorporate training objectives that are explicitly designed to address the challenges posed by distributional shift. It is worth mentioning that some studies, such as those by Wang et al. (2016); Duan et al. (2016); Melo (2022); Grigsby et al. (2023; 2024), are similar to our work in that they focus on the *Online Meta-RL setting*, where the primary objective during training is to maximize the total reward.

Inspired by the above studies, OSOM ingeniously reshapes the Transformer's sequence modeling abilities to perform contextual identity inference and response regarding opponents. The model processes historical interaction trajectories to infer the current opponent's type. This inference result then becomes a new context, which is used to guide the generation of the most appropriate response policy. The model design of OSOM facilitates its ICRL ability, making it possible to generalize the learned opponent identification and response knowledge to never-before-seen opponent types.

# B    ALGORITHMIC PSEUDOCODE FOR OSOM

## B.1    OSOM TRAINING

The complete training procedure is detailed in the pseudocode below.

---

**Algorithm 1** OSOM Training Procedure

---

**Initialize** model parameters: Encoder $\theta$, Decoder $\phi$, Identifier $\psi$, Responder $\omega$.
**Initialize** replay buffer $\mathcal{B}$ and historical OTE buffer $\mathcal{H}$.
**Initialize** hyperparameters: $K, C, \kappa, \alpha_1, \alpha_2, \alpha_3$.
**for** each training iteration **do**
    // **Sample a new set of opponents and generate corresponding random labels.**
    Sample $K$ opponent policies $\Pi^{\text{train}} = \{\pi^{-1,k}\}_{k=1}^{K}$ from the *opponent pool*.
    Generate $K$ random, pairwise orthogonal OTEs $\mathcal{Z}^{\text{train}} = \{z^{-1,k}\}_{k=1}^{K}$ on the unit sphere.
    // **Rollout phase: Collect interaction data.**
    **for** episode $h = 1, \ldots, N_{\text{train\_episodes}}$ **do**
        Sample an opponent policy $\pi^{-1,j} \in \Pi^{\text{train}}$ with its ground truth OTE $z^{-1,j} \in \mathcal{Z}^{\text{train}}$.
        Reset environment and get initial self-agent observation $o_0^1$.
        Initialize episode trajectory buffer $\tau_{\text{ep}}$.
        **for** timestep $t = 0, \ldots, T - 1$ **do**
            // **Aggregate context from historical identifications .**
            Retrieve recent selected OTEs $\{z^{\text{sel}}\}$ from $\mathcal{H}$ (current episode up to $t-1$ and previous $C$ episodes).
            Compute aggregated context $x_{t(h)} = \text{AveragePooling}(\{z^{\text{sel}}\})$.
            // **Self−agent acts based on observation and context.**
            Generate self-agent action $a_t^1 \sim \pi^1(\cdot | o_t^1, x_{t(h)}; \omega)$.
            Execute $a_t^1$ and opponent action $a_t^{-1} \sim \pi^{-1,j}(\cdot | o_t^{-1})$ in the environment.
            Receive $o_{t+1}^1, r_t^1, o_{t+1}^{-1}$.
            // **Distill opponent policy and perform identification .**
            Encode self-agent information: $e_t = f_\theta(o_t^1, a_t^1)$.
            Predict opponent OTE: $\hat{z}_t = \text{Identifier}_\psi(\{e\}, \mathcal{Z}^{\text{train}})$.
            Compute sampling probabilities $P(z^{-1,l}) = \text{softmax}(\hat{z}_t \cdot z^{-1,l})$ for all $l \in \{1, \ldots, K\}$.
            Sample a selected OTE $z_{t(h)}^{\text{sel}} \sim P(\cdot)$ and store in $\mathcal{H}$.
            // **Store all relevant data for updates.**
            Add $(o_t^1, a_t^1, r_t^1, o_{t+1}^1, x_{t(h)}, e_t, \hat{z}_t, z^{-1,j}, o_t^{-1}, a_t^{-1})$ to $\tau_{\text{ep}}$.
        **end for**
        Store completed trajectory $\tau_{\text{ep}}$ in replay buffer $\mathcal{B}$.
    **end for**
    // **Update phase: Optimize all model components.**
    Sample a batch of data from $\mathcal{B}$.
    Compute policy distillation objective $\mathcal{J}_{\text{distill}}$ using decoder $g_\phi$ on $(e_t, o_t^{-1}, a_t^{-1})$ per Eq. (1).
    Compute identification objective $\mathcal{J}_{\text{identify}}$ on $(\hat{z}_t, z^{-1,j}, \mathcal{Z}^{\text{train}})$ per Eq. (2).
    Compute response objective $\mathcal{J}_{\text{respond}}$ using an online RL algorithm (*e.g.*, PPO by Schulman et al. (2017)) on self-agent trajectories $(o_t^1, a_t^1, r_t^1, x_{t(h)})$ pe Eq. (4).
    Compute total objective $\mathcal{J}_{\text{total}} = \alpha_1 \mathcal{J}_{\text{distill}} + \alpha_2 \mathcal{J}_{\text{identify}} + \alpha_3 \mathcal{J}_{\text{respond}}$.
    Update $\theta, \phi, \psi, \omega$ using PPO policy-gradient updates on $\alpha_3 \mathcal{J}_{\text{respond}}$ combined with standard gradient updates on the auxiliary losses $\alpha_1 J_{\text{distill}}$ and $\alpha_2 J_{\text{identify}}$.
**end for**

---

## B.2 OSOM TESTING

The testing procedure formalizes the deployment of a trained OSOM agent against non-stationary OSOs, as detailed in the pseudocode below.

---

**Algorithm 2** OSOM Testing Procedure (On-the-Fly Identification and Response)

---

**Load** pre-trained model parameters $\theta, \phi, \psi, \omega$. Set models to evaluation mode.
**Define** the testing set of $M$ opponent policies $\Pi^{\text{test}} = \{\pi^{-1,m}\}_{m=1}^{M}$.
**Generate** $M$ new random, pairwise orthogonal OTEs $\mathcal{Z}^{\text{test}} = \{z^{-1,m}\}_{m=1}^{M}$.
**Initialize** historical OTE buffer $\mathcal{H}$.
**for** each test episode $h = 1, \ldots, N_{\text{test\_episodes}}$ **do**
    **// Opponent may switch its policy to simulate non-stationarity.**
    **if** $h \pmod{H} == 1$ **then**
        Sample a new opponent policy $\pi^{-1,j} \in \Pi^{\text{test}}$.
    **end if**
    Reset environment and get initial self-agent observation $o_0^1$.
    **for** timestep $t = 0, \ldots, T-1$ **do**
        **// Aggregate context from historical identifications.**
        Retrieve recent selected OTEs $\{z^{\text{sel}}\}$ from $\mathcal{H}$ (current episode up to $t-1$ and previous $C$ episodes).
        Compute aggregated context $x_{t(h)} = \text{AveragePooling}(\{z^{\text{sel}}\})$.
        **// Self-agent acts based on observation and context.**
        Generate self-agent action $a_t^1 \sim \pi^1(\cdot|o_t^1, x_{t(h)}; \omega)$.
        Execute $a_t^1$ in the environment and receive $o_{t+1}^1, r_t^1$.
        **// On-the-fly identification using the fixed, pre-trained model.**
        Encode self-agent information: $e_t = f_\theta(o_t^1, a_t^1)$.
        Predict opponent OTE: $\hat{z}_t = \text{Identifier}_\psi(\{e\}, \mathcal{Z}^{\text{test}})$.
        Compute sampling probabilities $P(z^{-1,l}) = \text{softmax}(\hat{z}_t \cdot z^{-1,l})$ for all $l \in \{1, \ldots, M\}$.
        Sample a selected OTE $z_{t(h)}^{\text{sel}} \sim P(\cdot)$ and store in $\mathcal{H}$.
    **end for**
**end for**

---

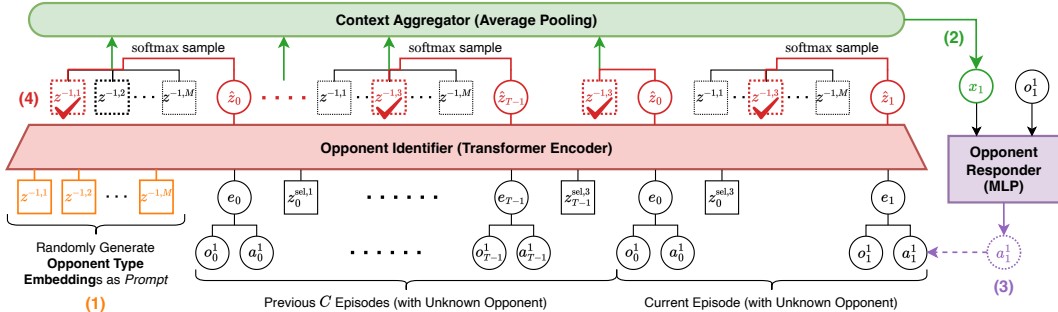

Figure 7: Testing procedure of OSOM. Taking timestep $1$ of the current episode against an unknown opponent policy as an example, we illustrate the four steps of OSOM's on-the-fly testing process: (1) *Embedding Generation and Prompting*: For the testing set of opponent policies $\{\pi^{-1,k}\}_{m=1}^{M}$ with $M$ opponent types, randomly generate $M$ corresponding OTEs $\{z^{-1,m}\}_{k=1}^{M}$, and use this set as the prompt for the input sequence. (2) *Context Aggregation*: The **Context Aggregator** aggregate the OTEs identified from recent interactions (the current episode up to timestep $0$ and the previous $C$ episodes) to form a compact context vector $x_1$ that represents the agent's current belief about the opponent's type. (3) *Action Selection (Response)*: The **Opponent Responder** takes the agent's current observation $o_1^1$ and the aggregated context vector $x_1$ as input to output an action $a_t^1$, allowing the agent's policy to be influenced by its belief about the opponent's identity. (4) *On-the-Fly Identification*: The agent encodes its own $(o_1^1, a_1^1)$ pair to produce a latent state $e_1$, which is then used by the **Opponent Identifier** along with the sequence of latent states from recent interactions and the full set of possible test-time OTEs to predict the opponent's identity $\hat{z}_1$. The agent probabilistically selects an OTE from the prompt set, updating its belief about the opponent's identity.

## C  TESTING PROCEDURE OF OSOM

The testing process evaluates the agent's ability to adapt to a set of $M$ unknown opponent policies. The model's parameters are frozen, and all adaptation occurs in-context by dynamically updating its understanding of the current opponent. An illustration of the OSOM testing procedure is provided in Fig. 7, and corresponding algorithmic pseudocode is supplemented in Algo. 2 to aid in comprehension.

Here is the breakdown of the procedure:

**Initialization and Setup.**  Before the testing episodes begin, the environment is prepared:

(1) *Load the Trained Model*: The final, trained parameters for the Encoder, Opponent Identifier, and Opponent Responder are loaded. The model is set to evaluation mode, meaning no further learning or gradient updates will occur.

(2) *Define the Opponent Set*: A set of $M$ new, potentially unseen opponent policies, denoted as $\Pi^{\text{test}}$, is established for the test.

(3) *Generate Random OTEs*: A corresponding set of $M$ new, random, and pairwise orthogonal OTEs, denoted as $\mathcal{Z}^{\text{test}}$, is generated. This set of OTEs acts as a 'prompt,' providing the agent with the possible identities of the opponents it might face in this new environment. The agent has no prior association between these specific OTEs and the opponent policies.

(4) *Initialize History*: A historical buffer for storing identified OTEs is cleared and prepared.

**Starting an Episode.**  At the beginning of each episode (or every $H$ episodes to simulate non-stationarity), an opponent is chosen:

(1) *Select Opponent Policy*: An opponent policy $\pi^{-1,j}$ is sampled from the test set $\Pi^{\text{test}}$. The agent does not know the identity of this opponent.

(2) *Reset Environment*: The environment is reset, and the agent receives its initial observation, $o_0^1$.

**The Interaction Loop (For each timestep $t$).** For every step within an episode, the agent performs a cycle of belief formation, action, and belief update.

(1) *Context Aggregation*: The agent first forms a belief about the current opponent's identity. The Context Aggregator retrieves the OTEs that were identified and stored in the historical buffer from recent interactions (*e.g.*, the current episode up to step $t-1$ and the previous $C$ episodes). It then calculates the average of these embeddings to produce a single, compact context vector, $x_{t(h)}$. This vector represents the agent's current, time-averaged belief about the opponent's type.

(2) *Action Selection (Response)*: The agent decides what action to take. The Opponent Responder network takes the agent's current observation $o_t^1$ and the aggregated context vector $x_{t(h)}$ as input. Based on this information, it outputs an action, $a_t^1$. This mechanism allows the agent's policy to be directly influenced by its belief about the opponent's identity.

(3) *On-the-Fly Identification*: After acting, the agent updates its belief about the opponent.

- The agent's own $(o_t^1, a_t^1)$ pair is passed through the pre-trained Encoder to produce a latent state, $e_t$. This state implicitly contains information about the opponent's behavior.
- The Opponent Identifier (a Transformer model) takes the sequence of latent states from the current episode up to $t$ and previous $C$ episodes $\left(e_{0(h-C)}, \ldots, e_{T-1(h-1)}, e_{0(h)}, \ldots, e_{t(h)}\right)$ and the full set of possible test-time OTEs ($\mathcal{Z}^{\text{test}}$) as input.
- It processes this information and outputs a predicted OTE, $\hat{z}_t$. This prediction represents the Identifier's best guess of the opponent's identity based on the interaction history.
- To make a choice, the agent probabilistically selects the OTE from the prompt set $\mathcal{Z}^{\text{test}}$ according to the dot-product similarities with its prediction $\hat{z}_t$. This selected OTE, $z_{t(h)}^{\text{sel}}$, is the agent's identification of the opponent at this timestep.

(4) *Update History*: The newly identified OTE, $z_{t(h)}^{\text{sel}}$, is stored in the historical buffer. This ensures it will be used by the Context Aggregator in subsequent timesteps to inform future actions.

This loop continues until the end of the episode, and the entire process is repeated for the desired number of test episodes. This procedure allows the agent to adapt its behavior on-the-fly by continually refining its belief about an opponent's identity and conditioning its actions on that belief, all without changing its underlying network weights.

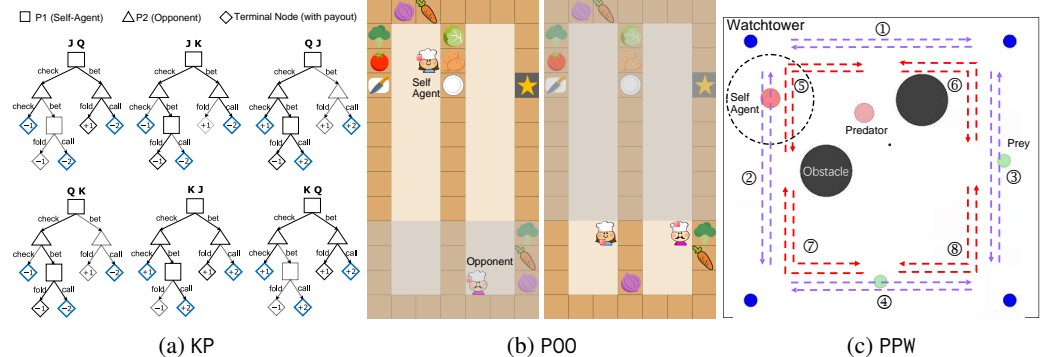

Figure 8: Illustrations of the multi-agent benchmarking environments (a) Kuhn Poker (KP), (b) Partially-Observable Overcooked (POO), and (c) Predator-Prey with Watchtowers (PPW).

# D  DETAILED INTRODUCTIONS OF ENVIRONMENTS

## D.1  KUHN POKER

The game of Kuhn Poker (KP) is a simplified, two-player (P1 and P2) version of poker, as detailed by Hoehn et al. (2005) and Kuhn (2016). The game's setup involves a three-card deck, from which each player is dealt a single card. The cards are ranked Jack, Queen, and King (from lowest to highest). KP has no suits, only ranks. Unlike games such as No-Limit Texas Hold'em, which permit multiple rounds of raising, the allowed actions in KP are restricted to simply "bet" or "pass."

The game unfolds in the following sequence: (1) Both players contribute one ante (chip) to the pot. (2) One card is dealt to each player from the deck, and the third, remaining card is not seen by either player. (3) Following the deal, P1 is the first to act, making a choice to either bet one chip or pass.

The trajectory of the game encompasses a total of three potential scenarios, contingent upon the sequence of actions executed by both players:

- P1 Bets: If Player 1 (P1) elects to bet, Player 2 (P2) is then presented with a choice: to bet (call), which results in P1's bet being matched and the game concluding in a showdown, or to pass (fold), thereby forfeiting the pot.

- P1 Passes, P2 Acts: Should P1 choose to pass, P2 can then either pass (check), leading directly to a showdown, or bet.

- P2 Bets After P1 Passes: If P2 places a bet after P1 has passed, P1 must then decide whether to bet (call), matching P2's bet and ending the hand in a showdown, or to pass (fold), which entails relinquishing the pot.

In this study, our primary focus is on learning the adaptation strategy for P1 against P2, where the opponent is modeled as P2. The specific card held by the opponent P2 is only disclosed at the point of a showdown (represented by the blue diamond nodes in Fig. 8a). Following the methodology for strategy simplification proposed by Hoehn et al. (2005), we systematically eliminate policies for P2 that are clearly dominated. For instance, P2 will never choose to bet with a Queen after P1 checks (passes), as P1 would consistently fold with a Jack and call with a King. The complete, simplified game tree can be referenced in the original paper by Hoehn et al. (2005). This simplification enables us to parameterize the P2 policy using only two variables, $\xi$ and $\eta$, each constrained to the range $[0, 1]$. Within, $\eta$ represents the probability that P2 'bets with a Queen' after P1 has bet. $\xi$ represents the probability that P2 'bets with a Jack' after P1 has passed. As a result, the full policy space of P2 can be partitioned into six distinct sections, with each section corresponding to the optimal best response strategy for P1.

**Observation Space.**  The agents within this game perceive a state that is represented by a 13-dimensional vector, which is constructed from three one-hot vectors.

(1) First One-Hot Vector (7-dimensions): This component is a 7-dimensional encoding of the current stage within the game tree.

(2) Second One-Hot Vector (3-dimensions): This component represents the hand card of the self-agent.

(3) Third One-Hot Vector (3-dimensions): This component represents the hand card of the opponent. Crucially, the opponent's hand is consistently represented by an all-zero vector until the game reaches a showdown stage.

**Action Space.** The permissible actions for each player are limited to either 'bet' or 'pass', which means the action space is a discrete space comprising two distinct actions.

**Reward.** The reward is not directly equivalent to the total amount in the pot. Instead, it is computed as the total chips in the pot minus the chips contributed by the winner. The loser's reward is the negative value of the winner's reward. The reward values are determined by the final size of the pot:

- Showdown (No Bets): If the game concludes in a showdown with no player betting (both pass), the pot contains 2 chips (one ante from each player). The player with the highest-ranked card wins the pot.

  – Reward: $\pm 1$ (Pot 2 - Winner's contribution 1).

- Showdown (With Bets): If the game concludes in a showdown after a sequence of bets (e.g., one player bets, and the other calls), the pot contains 4 chips (two chips from each player). The player with the highest-ranked card wins the pot.

  – Reward: $\pm 2$ (Pot 4 - Winner's contribution 2).

- Forfeit (Fold): If the game ends because one player forfeits the pot (folds), the pot contains 3 chips (one ante plus two bets/chips). The remaining player wins the pot.

  – Reward: $\pm 1$ (Pot 3 - Winner's contribution 2).

## D.2 Partially-Observable Overcooked

`Partially-Observable Overcooked` (POO) is a collaborative culinary simulation in which players assume the roles of chefs working together to accomplish multiple sub-tasks and serve prepared dishes, as described by Carroll et al. (2019). In this study, we present a more intricate *Multi-Recipe* variant, which incorporates modifications previously introduced by Charakorn et al. (2023) and Ma et al. (2024). To substantially increase the challenge and encourage a wider array of policy behaviors, we specifically incorporate two additional ingredients—potato and broccoli—and a corresponding increase in the number of available recipes. The game environment now features a total of six ingredients (Tomato, Onion, Carrot, Lettuce, Potato, and Broccoli) and nine recipes. A notable characteristic of this environment is a counter that divides the room, thereby making collaboration essential as chefs must exchange objects, such as ingredients and plates, across this counter. To successfully serve a dish, the required ingredients must first be taken to the cutting board and chopped. Once all the necessary chopped ingredients are placed onto a plate, the final dish must be transported to the delivery square to complete the task. Furthermore, we introduce partial observability by horizontally dividing the game scene into an 'upper room' and a 'lower room'. Each agent is restricted to observing only the objects present in the same room as itself. The agent situated in the left room is designated as the self-agent, and the masked gray area shown in Fig. 8b represents the portion of the environment that is unobserved by the self-agent.

**Observation Space.** The observation is structured as a 105-dimensional vector, which integrates a variety of features. These features encompass the agent's position, direction, currently held objects, objects immediately in front of the agent, and various other relevant attributes. To properly account for the partial observation, a flag is included for every relevant object to indicate its visibility status.

**Action Space.** Each agent is capable of selecting an action from a discrete space of six possible actions: moving left, moving right, moving up, moving down, interacting (with an object), and performing a no-operation (taking no action).

**Reward.**  Since POO is a fully cooperative game, all agents in the environment receive the same shared reward. There are three categories of rewards utilized within the game:

- Interactive Reward: An agent receives a reward of $0.5$ whenever it successfully interacts with an object. It should be noted that consecutive or repeated interactions with the same object do not yield any cumulative additional reward.

- Progress Reward: Each agent is given a reward of $1.0$ when the state of a recipe advances. For instance, if a chopped carrot is placed onto a plate, thereby changing the recipe state from "chopped carrot" to "carrot plate," all agents receive this reward.

- Completion Reward: When a prepared dish that fulfills the requirements of a recipe is successfully served to the delivery square, every agent receives a substantial reward of $10.0$.

### D.3  PREDATOR-PREY WITH WATCHTOWERS

Predator-Prey with Watchtowers (PPW) is a multi-agent environment where the agents face both collaborative and competitive dynamics. We utilize a modified version of the standard predator-prey scenario taken from the *Multi-agent Particle Environment* (MPE), which is commonly employed in the Multi-Agent RL (MARL) literature (Lowe et al., 2017).

As depicted in the game illustration of Fig. 8c, the environment contains the following components:

- Two Predators: represented by red circles, with the darker one designating the self-agent.
- Two Prey: represented by green circles.
- Multiple Landmarks: represented by gray and blue circles.

The core objectives are two-fold: the predators are tasked with chasing the prey, while the prey attempt to escape the predators. Furthermore, the predators operate in a collaborative manner, as they are required to coordinate their actions so that all prey are simultaneously covered by a predator. Each simulation episode is limited to a maximum duration of $40$ timesteps. However, if all the prey are tagged (or "touched") by the predators, the episode concludes immediately.

To augment the complexity of the task, we utilize the version modified by Ma et al. (2024), which incorporates both partial observability and the addition of 'four watchtowers': the blue circles situated at the corners of the figure. The self-agent's visual range is limited; it can only perceive other agents and landmarks that fall within its observation radius, which is fixed at $0.2$ for all experiments. The self-agent has the option to navigate to a watchtower to obtain full observability. When the self-agent is in contact with any of these watchtowers, its field of view expands to encompass all agents and landmarks present in the entire environment.

**Observation Space.**  The observation space is a 37-dimensional vector. It is composed of the positions and velocities of the various agents, as well as the positions of the landmarks. To implement the partial observability, an additional $0/1$ sign is appended for every landmark and every agent other than the self-agent, which indicates whether that entity is currently visible to the self-agent. For any entity that is invisible, its sign, positions, and velocities are all set to 0. Finally, all positions, except for the absolute position of the self-agent, are relative to the self-agent's location.

**Action Space.**  The action space employed for PPW is a discrete space comprising five actions, corresponding to the ability to move left, right, up, down, or stand still (take no movement action).

**Reward.**  The predators are assigned a common shared reward that is designed to incentivize collaboration and ensure that all prey are covered. Specifically, if $\mathcal{A}$ denotes the set of all predators and $\mathcal{B}$ denotes the set of all prey, the reward received by the predators at each timestep is defined by the following expression:

$$-c \sum_{b \in \mathcal{B}} \min_{a \in \mathcal{A}} \mathsf{dist}(a, b),$$

where dist is the *Euclidean distance function*, and the constant $c$ is set to $0.1$. Intuitively, this formulation encourages the predators to divide and conquer the task, ensuring that for every individual prey, there is at least one predator located in close proximity.

# E OPPONENT POOL DESIGN

In this section, we provide a detailed description of the design and construction of the opponent policy pools used in our experiments. We primarily follow the methodology outlined by Ma et al. (2024) to generate a diverse set of rule-based opponent policies for each environment.

Current work, as supported by research from Strouse et al. (2021); Charakorn et al. (2023); Lupu et al. (2021)u, predominantly employs RL algorithms enhanced with diversity objectives to cultivate a wide range of policy behaviors. However, this paper takes a different approach by generating a collection of rule-based policies that incorporate human-derived priors. The rationale for this decision is that the P2 policy within the KP framework can be effectively parameterized by two probabilities, denoted as $\xi$ and $\eta$. Furthermore, we posit that the preference-based policies utilized in POO and PPW more accurately represent human-like behaviors within the game environment. The specific details regarding the pool of these rule-based policies are provided in the following section.

## E.1 KUHN POKER

In accordance with the details provided in Sec. D.1, the dominant strategies for P2 have been eliminated. This allows the P2 policy to be defined by two key parameters: $\eta$ and $\xi$. The parameter $\eta$ represents the probability of P2 betting with a Queen when P1 has already bet, whereas $\xi$ denotes the probability of P2 betting with a Jack following a pass from P1.

This approach enables the generation of an arbitrary number of P2 policies by randomly sampling the parameters $\xi$ and $\eta$. For the purposes of this paper, a total of 40 P2 policies were sampled to form the *full training opponent set* Train Set, while 10 P2 policies were sampled for the *full unseen opponent set* Eval Set.

## E.2 PARTIALLY-OBSERVABLE OVERCOOKED

In this paper, we follow the method of Ma et al. (2024) for generating opponent policies that are based on individual preferences for specific recipes. Each opponent policy is thus tailored to a particular recipe, such as Tomato and Onion Salad. These opponents are spatially constrained to the right side of the kitchen environment and engage solely with ingredients and dishes relevant to their preferred recipe. For instance, a policy favoring Tomato and Onion Salad would concentrate on sub-tasks related to handling fresh or chopped tomatoes and onions, as well as delivering final dishes that exclusively contain these two ingredients.

At each timestep, the opponent's decision-making is as follows: it first assesses the completion status of its current sub-task. If the sub-task is not yet finished, the opponent computes the shortest path to its target location and proceeds to navigate along that path. Conversely, upon completion of a sub-task, the opponent samples a new sub-task from its predefined set of preferred options.

In addition, two parameters are utilized to control more fine-grained aspects of the policies. The first, $P_{\text{nav}}$, is the probability of selecting a lateral movement (right or left) over a vertical movement (up or down) when multiple shortest paths exist. The second, $P_{\text{act}}$, represents the probability of an agent choosing a random action from the action space instead of the optimal action for its current sub-task. To illustrate, consider an opponent aiming to place a Tomato on a counter. With a probability of $P_{\text{act}}$, the opponent will select a random action from the entire action space. Conversely, with a probability of $1 - P_{\text{act}}$, it will choose the optimal action for the task, such as navigating towards the counter or performing the necessary interaction.

We posit that rule-based agents exhibit behaviors that are more human-like than those of self-play agents trained with RL algorithms. This belief is supported by two main points. First, cognitive studies (Etel & Slaughter, 2019; Sher et al., 2014) suggest that human actions are indeed based on underlying intentions and desires. Additionally, self-play agents frequently develop arbitrary conventions (Hu et al., 2020). In the POO environment, such conventions may involve consistently placing or taking ingredients and plates from a specific counter and refusing to interact with objects at other locations. However, these self-play conventions are rarely seen in human behavior. As a result, a preference-based policy proves to be a more suitable choice.

The P00 scenario detailed in this paper is built upon a set of 9 recipes. Additionally, two parameters, $P_{\text{nav}}$ and $P_{\text{act}}$, are employed to control more granular strategic behaviors. The process for generating a new opponent policy involves first uniformly sampling its preferred recipe from the nine available options, and then randomly sampling values for $P_{\text{nav}}$ and $P_{\text{act}}$. The *full training opponent set* Train Set, consists of 18 policies, while the *full unseen opponent set* Eval Set, contains 9 policies.

### E.3 Predator-Prey with Watchtowers

Our approach to generating preference-based opponent policies in PPW follows the design strategy outlined by Ma et al. (2024).

For the predator opponent, we designed policies that have a preference for one specific prey from the two available options. When operating under full observation, the predator will invariably pursue its preferred prey.

For the prey opponents, we developed a total of 8 distinct movement patterns, which are illustrated as dotted lines and labeled ①-⑧ in Fig. 8c. Each prey opponent is designed to move back and forth along a single, preferred path. This collection of paths is partitioned into a training set, which includes the blue dotted lines (①-④), and an unseen set, which consists of the red dotted lines (⑤-⑧).

The final *full training opponent set* Train Set was created by sampling 16 combinations, with each combination comprising one predator opponent policy and two training prey opponent policies. Similarly, the final *full unseen opponent set* Eval Set was generated from 24 sampled combinations, each consisting of one predator opponent policy and two unseen prey opponent policies.

## F Training Recipes

The overall training pipeline for PACE (Ma et al., 2024), LILI (Xie et al., 2021), LIAM (Papoudakis et al., 2021), LIAMX (Papoudakis et al., 2021), and Generalist is fundamentally similar to the OSOM procedure . The key distinction lies in their optimization objectives. Specifically, these OM approaches do not incorporate the Contrastive Learning objective found in OSOM (Eq. (2) in the original paper) and may use different Representation Learning objectives (different from Eq. (1)). Furthermore, a major architectural difference is that these OM approaches do not utilize Transformers as their backbones. In contrast, the training procedure for GSCU is notably different from the other OM approaches mentioned, and its details can be found in the original paper (Fu et al., 2022).

For all of the baselines and ablation variants, we employ PPO (Schulman et al., 2017; Kostrikov, 2018) as the underlying RL training algorithm. The specific hyperparameters used for the architectures, training, and testing for each environment are detailed in the following sections: KP in Sec. G.1, P00 in Sec. G.2, and PPW in Sec. G.3.

### F.1 Specific Details for OSOM

**Architectural Design.** The choice of a Transformer as the model backbone is central to OSOM's functionality. Its attention mechanism is uniquely suited for both identification and response.

- Encoder: We implement the self-agent observation-action Encoder as a 2-layer Multi-Layer Perceptron (MLP) with ReLU (Agarap, 2018) activation functions. The number of hidden units for KP, P00, and PPW is 64, 128, and 128, respectively.
- Opponent Identifier: We choose an auto-regressive Transformer (Vaswani et al., 2017) implemented using the PyTorch Library (Paszke et al., 2019) for this component. The input sequence would consist of the latent states $(e_{0(h-C)}, \ldots, e_{T-1(h-1)}, e_{0(h)}, \ldots, e_{t(h)})$. The set of OTEs, $\mathcal{Z}$, can be provided as a prompt, prepended to the main sequence. The attention mechanism allows the model to perform a soft search over the OTEs in the prompt, comparing each one to the encoded history to produce the final prediction $\hat{z}_t$. Specifically, the backbone of OSOM is a 3-layer Transformer with 1 attention head. The number of hidden units (*i.e.*, dimension of the hidden state) of the model for KP, P00, and PPW is 64, 128, and 128, respectively. The number of hidden units of the feed-forward layer for KP, P00, and PPW is 256, 512, and 512, respectively. The dropout rate is set to 0.1 for all environments.

- Latent Layer: We also include a latent layer to project the output of the Transformer to the same dimension as the OTEs, which is the final output of the Opponent Identifier, *i.e.*, $\hat{z}_t$. This is implemented as a 2-layer MLP with ReLU (Agarap, 2018) activation functions. The number of hidden units for KP, POO, and PPW is $64$, $128$, and $128$, respectively.

- Opponent Responder: While a Transformer could also be used here, we adopt a simpler MLP, which is often sufficient and more computationally efficient. The input to this network is the concatenation of the current observation $o_t^1$ and the aggregated context vector $x_t$. This model allows the context to directly modulate the policy output for the given input. Specifically, the actor and critic of PPO are both implemented as 2-layer MLPs with ReLU (Agarap, 2018) activation functions. The number of hidden units for all environments is $128$.

**Masking Scheme.** At each timestep $t$ of episode $h$, the Opponent Identifier receives a sequence consisting of a prefix of $K$ OTE prompt tokens followed by the latent states from the previous $C$ episodes and the current episode up to $t$. We apply a **causal mask only over the temporal latent-state tokens**: each latent-state token can attend to all OTE prompt tokens and to latent states from earlier timesteps (across the previous $C$ episodes and the current episode), but not to any future latent states. The OTE prompt tokens themselves are visible from any position, since they represent a static set of candidate opponent types rather than a temporal process.

**Positional Embeddings.** We use learnable absolute positional embeddings implemented with 'nn.Embedding' of PyTorch (Paszke et al., 2019). For each latent state $e_{t(h)}$ we add (1) a timestep-level positional embedding reflecting its index within episode $h$, and (2) an episode-level embedding indicating which of the last $C$ episodes (or the current episode) it belongs to. OTE prompt tokens do not carry temporal position and thus are not assigned any positional embeddings. As a result, the Identifier is effectively insensitive to the ordering of the $K$ OTE tokens, which are randomly shuffled when we construct the prompt.

**Orthogonal OTEs Generation.** We suggest that OTEs should be random, pairwise (nearly) orthogonal, and reside on the unit sphere. A standard and numerically stable procedure to achieve is as follows:

(1) Random Sampling: Generate $K$ (or $M$) vectors of dimension $d$ by sampling each component from a standard Gaussian distribution, $\mathcal{N}(0, 1)$.

(2) Orthogonalization: Apply the Gram-Schmidt process to this set of random vectors. This procedure iteratively modifies the vectors to ensure they are mutually orthogonal.

(3) Normalization: Normalize each of the resulting orthogonal vectors to have a unit L2-norm ($\|\cdot\|_2 = 1$). This places them on the surface of a $d$-dimensional hypersphere.

This process ensures that the dot product similarity used in the CL objective (*c.f.*, Eq. (2)) is well-behaved. Orthogonality guarantees that a predicted vector $\hat{z}_t$ can have a high similarity score with one target OTE without inadvertently having a high score with others, which sharpens the optimization signal and prevents ambiguity during identification.

In this view, the prompt size $M$ and the embedding dimension $d$ should be understood as standard **capacity hyperparameters**: for a fixed $d$, very large $M$ will eventually force some opponent types to share similar OTEs, just as any finite-dimensional embedding has finite capacity to separate many distinct classes. In our experiments we choose $d \in \{64, 128\}$ with $M \leq 50$, so the apparent condition $d \geq M$ does not bind in practice.

### F.2 OTHER GENERAL DETAILS

**Training Budgets.** We keep the original hyperparameters for GSCU on KP. For KP, the training budget for all algorithms except GSCU is 5 million timesteps, while for GSCU embedding learning takes 1 million episodes and conditional RL takes 1 million episodes. For POO, the training budget for all algorithms except GSCU is 30 million timesteps, while for GSCU the embedding learning takes 2 million timesteps and the conditional RL takes 30 million timesteps. For PPW, the training budget for all algorithms including GSCU is 15 million timesteps. The embedding learning for GSCU takes an additional 2 million timesteps.

We have retained the original hyperparameters for GSCU on the KP environment. The training budgets for each environment are as follows:

- KP: For all algorithms except GSCU, the training budget is 5 million timesteps. For GSCU, the budget consists of 1 million episodes for embedding learning and an additional 1 million episodes for conditional RL.
- POO: For all algorithms except GSCU, the training budget is 30 million timesteps. For GSCU, the budget is 2 million timesteps for embedding learning and 30 million timesteps for conditional RL.
- PPW: The training budget for all algorithms, including GSCU, is 15 million timesteps. For GSCU, an additional 2 million timesteps are allocated specifically for embedding learning.

**Architectures.** For algorithms that employ a Recurrent Neural Network (RNN), including Generalist, LIAM, and LIAMX, the implementation uses a single-layer GRU (Chung et al., 2014) with 128 hidden units. The training for the RNN is conducted using Back-Propagation Through Time (BPTT), where gradients are detached every 20 timesteps. This means that all training trajectories are partitioned into segments of 20 timesteps for BPTT. The actor and critic networks share the same RNN, as well as the hidden layers that precede the RNN.

Similar to OSOM, the algorithms PACE and LILI also employ a 2-layer MLP as its Encoder, a 2-layer MLP as the Latent Layer to project the output of the Encoder to the same dimension of the output hidden states, and a 2-layer MLP for both the actor and critic networks of PPO. Unlike OSOM, the algorithms PACE and LILI include a *Aggregate Model* that is implemented as a 1-layer MLP. The number of hidden units for the Aggregate Model for KP, POO, and PPW is 64, 128, and 128, respectively.

**Auxiliary Tasks.** Algorithms that incorporate *auxiliary tasks* have an additional objective that is optimized concurrently with the primary RL objective. This auxiliary objective is computed using the same mini-batch of data as the RL training.

- OSOM: The auxiliary task is weighted at 1.0 for both observation and action prediction. In another word, the coefficient $\alpha_1$ for the Representation Learning objective in Eq. (5) is set to 1.0.
- PACE: The auxiliary task is also weighted at 1.0.
- LIAM: The auxiliary task is given a weight of 1.0 for both observation and action prediction.
- LILI: This approach uses the last episode as its context, as detailed by Xie et al. (2021). Its auxiliary task is weighted at 1.0 for both reward and next observation prediction.

**PACE's Exploration Reward.** For the implementation of PACE, we follow the original paper (Ma et al., 2024) to add an exploration reward to the environmental reward. The coefficient for the exploration reward undergoes a linear decay from an initial value, $c_{init}$, to 0 over a duration of $N_{decay}$ timesteps. The specific values for these parameters are:

- KP: $c_{init} = 0.01$ and $N_{decay} = 4 \times 10^6$ timesteps.
- POO: $c_{init} = 0.2$ and $N_{decay} = 2.5 \times 10^7$ timesteps.
- PPW: $c_{init} = 0.1$ and $N_{decay} = 1.5 \times 10^7$ timesteps.

Additionally, the context encoder is initially trained for $N_{warm}$ timesteps using only the auxiliary task, without any RL objective. The values for this warm-up period are:

- KP: $N_{warm} = 10^5$ timesteps.
- POO: $N_{warm} = 10^6$ timesteps.
- PPW: $N_{warm} = 5 \times 10^5$ timesteps.

# G    HYPERPARAMETER SETTINGS

## G.1    KUHN POKER

Table 2: Hyperparameters for all the algorithms in the KP environment.

| Parameter Name | Algorithms | | | | | |
|---|---|---|---|---|---|---|
| | Generalist | LILI | LIAM(X) | GSCU | PACE | OSOM |
| Learning Rate | 2e-4 | 2e-4 | 2e-4 | 5e-4 | 2e-4 | 2e-4 |
| PPO Clip $\epsilon$ | 0.2 | 0.2 | 0.2 | 0.2 | 0.2 | 0.2 |
| Entropy Coefficient | 5e-4 | 5e-4 | 5e-4 | 0.01 | 5e-4 | 5e-4 |
| Discount Factor $\gamma$ | 0.99 | 0.99 | 0.99 | 0.99 | 0.99 | 0.99 |
| GAE $\lambda$ | 0.95 | 0.95 | 0.95 | 0.95 | 0.95 | 0.95 |
| Batch Size | 80000 | 80000 | 80000 | 1000 | 80000 | 80000 |
| # of Update Epochs | 3 | 3 | 3 | 5 | 3 | 3 |
| # of Mini Batches | 80 | 80 | 80 | 30 | 80 | 80 |
| Gradient Clipping (L2) | 2.0 | 2.0 | 2.0 | 0.5 | 2.0 | 2.0 |
| Activation Function | ReLU | ReLU | ReLU | ReLU | ReLU | ReLU |
| Actor & Critic Hidden Dims | [128, 128] | [128, 128] | [128, 128] | [128, 128] | [128, 128] | [128, 128] |
| Encoder $f$ Hidden Dims | N/A | [64, 64] | N/A | N/A | [64, 64] | [64, 64] |
| Latent Layer Hidden Dims | N/A | [64, 64] | N/A | N/A | [64, 64] | [64, 64] |

**Specific Hyperparameters for OSOM.**    The specific hyperparameters for OSOM in the KP environment are as follows:

- Number of opponent types sampled during training ($K$): 8.

- Number of opponent types present during testing ($M$): 40 for Train Set, 10 for Eval Set, and 50 for All Set.

- Size of the sliding window (in episodes) for context aggregation ($C$): 20.

- Temperature coefficient for the Contrastive Learning objective in Eq. (2) ($\kappa$): 1.5.

- Coefficients for weighting the three objective components in Eq. (5) ($\alpha_1, \alpha_2, \alpha_3$): $1.0, 0.01, 1.0$, respectively.

- Number of timesteps for resampling the opponent policies and their corresponding OTEs during training: 3000.

## G.2 PARTIALLY-OBSERVABLE OVERCOOKED

Table 3: Hyperparameters for all the algorithms in the POO environment.

| Parameter Name | Algorithms | | | | | |
|---|---|---|---|---|---|---|
| | Generalist | LILI | LIAM(X) | GSCU | PACE | OSOM |
| Learning Rate | 1e-3 | 1e-3 | 1e-3 | 7e-4 | 1e-3 | 1e-3 |
| PPO Clip $\epsilon$ | 0.2 | 0.2 | 0.2 | 0.2 | 0.2 | 0.2 |
| Entropy Coefficient | 0.03 | 0.03 | 0.03 | 0.01 | 0.03 | 0.03 |
| Discount Factor $\gamma$ | 0.99 | 0.99 | 0.99 | 0.99 | 0.99 | 0.99 |
| GAE $\lambda$ | 0.95 | 0.95 | 0.95 | 0.95 | 0.95 | 0.95 |
| Batch Size | 72000 | 72000 | 72000 | 2500 | 72000 | 72000 |
| # of Update Epochs | 3 | 3 | 3 | 8 | 3 | 3 |
| # of Mini Batches | 90 | 90 | 90 | 2 | 90 | 90 |
| Gradient Clipping (L2) | 15.0 | 15.0 | 15.0 | 0.5 | 15.0 | 15.0 |
| Activation Function | ReLU | ReLU | ReLU | Tanh | ReLU | ReLU |
| Actor & Critic Hidden Dims | [128, 128] | [128, 128] | [128, 128] | [64 64] | [128, 128] | [128, 128] |
| Encoder $f$ Hidden Dims | N/A | [128, 128] | N/A | N/A | [128, 128] | [128, 128] |
| Latent Layer Hidden Dims | N/A | [128, 128] | N/A | N/A | [128, 128] | [128, 128] |

**Specific Hyperparameters for OSOM.** The specific hyperparameters for OSOM in the POO environment are as follows:

- Number of opponent types sampled during training ($K$): 4.

- Number of opponent types present during testing ($M$): 18 for Train Set, 9 for Eval Set, and 27 for All Set.

- Size of the sliding window (in episodes) for context aggregation ($C$): 5.

- Temperature coefficient for the Contrastive Learning objective in Eq. (2) ($\kappa$): 1.5.

- Coefficients for weighting the three objective components in Eq. (5) ($\alpha_1, \alpha_2, \alpha_3$): $1.0, 0.1, 1.0$, respectively.

- Number of timesteps for resampling the opponent policies and their corresponding OTEs during training: 3000.

## G.3 PREDATOR-PREY WITH WATCHTOWERS

Table 4: Hyperparameters for all the algorithms in the PPW environment.

| Parameter Name | Algorithms | | | | | |
|---|---|---|---|---|---|---|
| | Generalist | LILI | LIAM(X) | GSCU | PACE | OSOM |
| Learning Rate | 4e-4 | 4e-4 | 4e-4 | 5e-4 | 4e-4 | 4e-4 |
| PPO Clip $\epsilon$ | 0.2 | 0.2 | 0.2 | 0.2 | 0.2 | 0.2 |
| Entropy Coefficient | 0.03 | 0.03 | 0.03 | 0.01 | 0.03 | 0.03 |
| Discount Factor $\gamma$ | 0.99 | 0.99 | 0.99 | 0.99 | 0.99 | 0.99 |
| GAE $\lambda$ | 0.95 | 0.95 | 0.95 | 0.95 | 0.95 | 0.95 |
| Batch Size | 64000 | 64000 | 64000 | 2500 | 64000 | 64000 |
| # of Update Epochs | 2 | 2 | 2 | 8 | 2 | 2 |
| # of Mini Batches | 150 | 150 | 150 | 2 | 150 | 150 |
| Gradient Clipping (L2) | 15.0 | 15.0 | 15.0 | 0.5 | 15.0 | 15.0 |
| Activation Function | ReLU | ReLU | ReLU | Tanh | ReLU | ReLU |
| Actor & Critic Hidden Dims | [128, 128] | [128, 128] | [128, 128] | [64 64] | [128, 128] | [128, 128] |
| Encoder $f$ Hidden Dims | N/A | [128, 128] | N/A | N/A | [128, 128] | [128, 128] |
| Latent Layer Hidden Dims | N/A | [128, 128] | N/A | N/A | [128, 128] | [128, 128] |

**Specific Hyperparameters for OSOM.** The specific hyperparameters for OSOM in the PPW environment are as follows:

- Number of opponent types sampled during training ($K$): 4.

- Number of opponent types present during testing ($M$): 16 for Train Set, 24 for Eval Set, and 40 for All Set.

- Size of the sliding window (in episodes) for context aggregation ($C$): 5.

- Temperature coefficient for the Contrastive Learning objective in Eq. (2) ($\kappa$): 1.5.

- Coefficients for weighting the three objective components in Eq. (5) ($\alpha_1, \alpha_2, \alpha_3$): $1.0, 1.0, 1.0$, respectively.

- Number of timesteps for resampling the opponent policies and their corresponding OTEs during training: 3000.

## H THE USE OF LARGE LANGUAGE MODELS

This paper utilizes Large Language Models to polish the writing of certain sections.

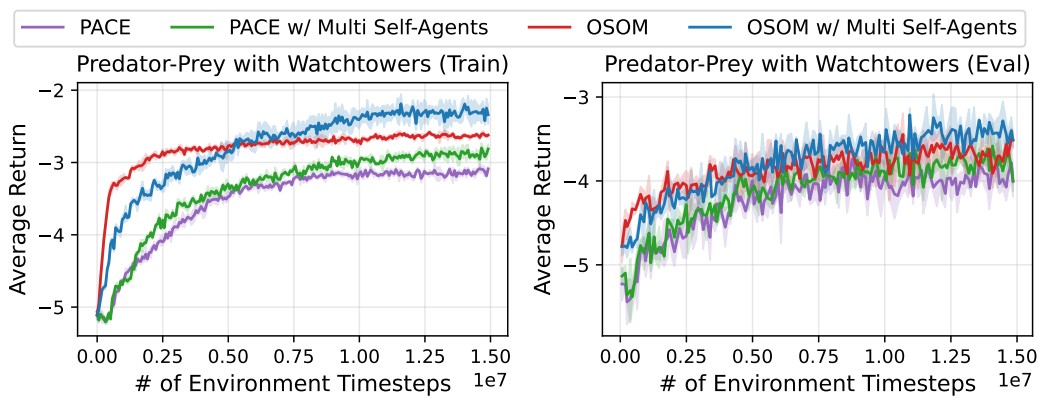

Figure 9: Training curves for centralized control of multiple self-agents in PPW. We adopt the same experimental setup as in Fig. 6 but with both predators being controllable self-agents.

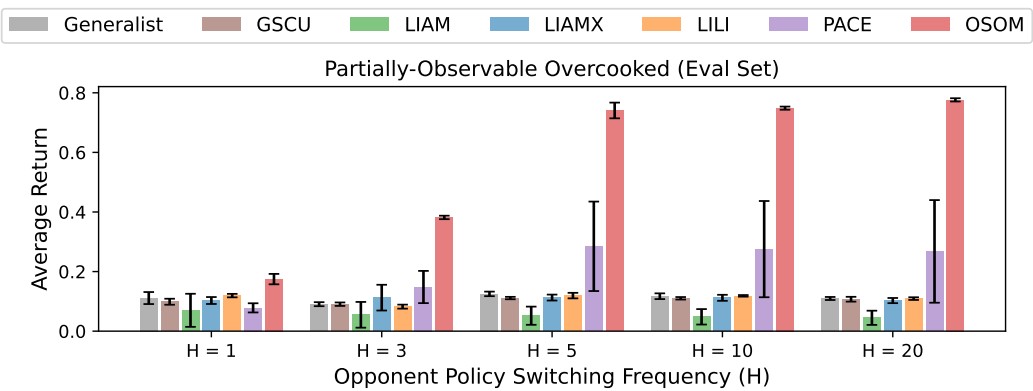

Figure 10: Ablation study on the opponent switch frequency $H$ in POO (Eval Set). We adopt the same experimental setup as in Fig. 3 but vary $H$ from $\{1, 3, 5, 10, 20\}$.

## I  ADDITIONAL EXPERIMENTAL RESULTS

**Question 6.**  *Can OSOM effectively control multiple self-agents?*

To directly test whether OSOM can control a *team* of agents, we consider a centralized-control variant of PPW in which both predators are controllable self-agents and a joint controller outputs their joint action at each timestep (see Sec. D.3 for details). We compare PACE and OSOM in this setting, including both the original single-self versions and their multi-self counterparts ('PACE w/ Multi Self-Agents' and 'OSOM w/ Multi Self-Agents'), where OSOM uses factorized per-agent OTEs for the two preys and aggregates them into a team-level context. As shown in Fig. 9, centralized team control yields slower early convergence for both approaches but higher asymptotic returns than their single-self variants, and **OSOM w/ Multi Self-Agents consistently outperforms PACE w/ Multi Self-Agents across training**, demonstrating that OSOM remains effective when extended to non-trivial multi-agent team control.

**Question 7.**  *How does the opponent switch frequency $H$ affect OSOM's performance?*

We further study the impact of the switch frequency $H$ on POO (Eval Set) by varying $H \in \{1, 3, 5, 10, 20\}$. The corresponding results are presented in Fig. 10. Across all choices of $H$, OSOM dominates all baselines in terms of average return and success rate, and the relative gap remains stable. This suggests that our conclusions are not tied to a specific non-stationarity level, and that OSOM is robust to both fast and slow opponent switches.

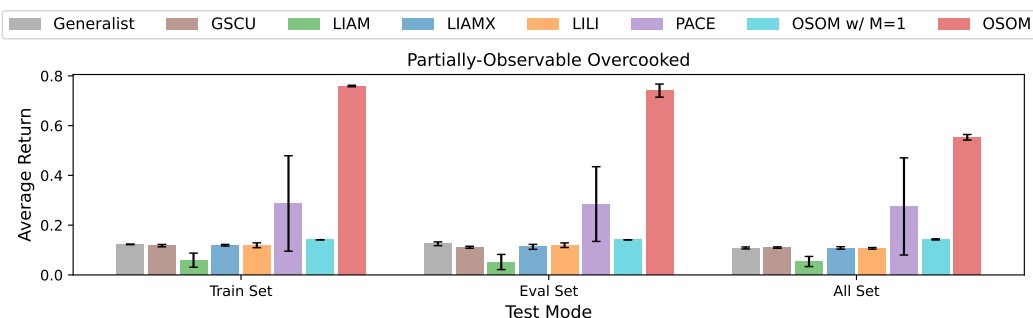

Figure 11: Ablation study on the degenerate case of a single OTE ($M = 1$) in P00. We adopt the same experimental setup as in Fig. 3 but set $M = 1$ for OSOM's OTE prompt.

**Question 8.** *How does OSOM perform in the degenerate case of a single OTE ($M = 1$)?*

We also investigate the degenerate case $M = 1$, where the prompt contains only a single OTE and OSOM deems that there is at most one opponent type. In this regime, the context vector collapses to a constant, and OSOM behaves like a generalist agent without meaningful type uncertainty. In a P00 experiment with $M = 1$ as shown in Fig. 11, OSOM performs comparably to other OM baselines but significantly worse than full OSOM with $M \geq 2$, which can exploit variation across opponent types. This behavior is consistent with our interpretation of OSOM as an open-set opponent model whose advantages emerge precisely when multiple types are possible.

