# OpenReview forum: "Open Set Opponent Modeling"
_ICLR.cc/2026/Conference — Submitted to ICLR 2026_

### Official Review · Reviewer_VeR7 · 2025-10-29

**Soundness:** 1
**Presentation:** 3
**Contribution:** 2
**Rating:** 2
**Confidence:** 4

**Summary:**

This paper introduces Open Set Opponent Modeling (OSOM), an end-to-end framework that enables an agent to explicitly identify and respond to policies not seen during training. A representation-learning objective uses an encoder–decoder structure to predict an opponent’s observations and actions from the self-agent’s trajectory, aiming to create a latent state, $e_t$, that remains informative under partial observability. This requires access to opponent data during training. A Transformer-based Opponent Identifier takes the history of latent states and a set of $K$ randomly generated, orthogonal Opponent Type Embeddings (OTEs) as a prompt. It is trained via a contrastive loss to output a prediction, $\hat{z}_t$, that is maximally similar to the ground-truth OTE for the current opponent. A separate Opponent Responder is trained via online RL. Its policy is conditioned on the self-agent’s observation, $o_t^1$, and a context vector, $x_t$, which is an average-pooled aggregation of the OTEs selected by the Identifier over the recent history. At test time, the model is frozen, and a new set of $M$ random, orthogonal OTEs is generated to serve as the prompt for the $M$ unseen test opponents. The agent then performs on-the-fly identification and response. The authors present strong empirical results, showing significant outperformance against baselines in unseen scenarios.

**Strengths:**

The paper tackles the open-set opponent modeling problem, which is a significant and practical challenge. The core idea of using randomly generated embeddings as a prompt for a Transformer, and training it with a contrastive loss to perform open-set identification, is novel and interesting, though with technical concerns. OSOM attempts to explicitly identify which unseen opponent it is facing, which could enable more complex reasoning and adaptation.

**Weaknesses:**

1. My main concern is as follows:

The Responder ($\pi^1(a^1|o^1, x)$) learns to map a specific context vector $x_{train}$ to an optimal policy. This $x_{train}$ is an aggregate of *selected* OTEs from the *training prompt* ($\mathcal{Z}^{train}$). For example, it learns that the vector $z_{train, A}$ means "best respond to Opponent A."

  During Testing, let's assume opponent A is also in the Eval Set. The trained Identifier (correctly) outputs a prediction $\hat{z} _ t$ based on the opponent's behavior. However, it must now match this $\hat{z} _ t$ to the closest OTE in the new, random test prompt ($\mathcal{Z}^{test}$). Let's say it selects $z_{test, B}$. The Responder now receives $z_{test, B}$ as its context.

  The Responder was trained to associate $z_{train, A}$ with Opponent A's policy. It has *never* seen $z_{test, B}$ and has no way of knowing that $z_{test, B}$ *also* means "best respond to Opponent A." Moreover, $z_{test, B}$ may be very likely orthogonal to $z_{train, A}$ if $d$ is large enough. The mapping from opponent policy to context vector is completely and randomly re-assigned at test time, I don't think it can learn any association between $z_{train, A}$ and $z_{test, B}$.

2. Let's continue with the $M=1$ test case. If the model is prompted with only one test OTE ($z_{test, 1}$), the selection probability (Eq. 3) for that OTE will always be 1, regardless of the Identifier's output $\hat{z} _ t$.
The context vector $x_t$ will thus always be $z_{test, 1}$. This means the Responder's policy will be fixed and non-adaptive, conditioned on a single random vector, which should not work when $z_{test, 1}$ is sampled badly. I would like to see the experiment of $M=1$.

3. The policy distillation objective ($\mathcal{J}_{distil}$) requires access to the opponent's observations and actions ($o^{-1}, a^{-1}$) during training. While the ablation study (Fig. 5) shows the model can work without it ("OSOM w/o Distill Loss"), performance drops significantly. This reliance on privileged information during training should be more clearly stated as a limitation.

4. The method requires a priori knowledge of the exact number of opponent types, $M$, that will be present in the test set (as said in Sec. 5). In a truly "open" environment, $M$ is unknown. This assumption significantly limits the "openness" of the setting and is a major practical limitation. The method relies on generating $M$ pairwise orthogonal OTEs. This is only possible if the embedding dimension $d$ is greater than or equal to $M$. The paper does not discuss what happens if the number of test-time opponents $M$ is larger than the training-time embedding dimension $d$. This seems to be a hard constraint on the method's scalability.

5. The Context Aggregator uses simple average pooling over the last $C$ episodes, which is a naive approach. A discounted average might be better if we assume the non-stationary policy change is not dramatic.

This paper's core idea is novel, but the current description of the method seems to contain a fundamental disconnect that makes its strong empirical results difficult to understand. Therefore, I need to reject this paper before receiving reasonable clarification. I am willing to increase the score if the explanation and rebuttal provided by the author are convincing.

**Questions:**

1. Could the authors please clarify the disconnect between the Responder and the Identifier? The Responder is trained on context vectors derived from the training OTEs, but at test time, it is fed context vectors derived from a new, random set of test OTEs. How can the Responder generalize its policy when the context vectors is randomly re-initialized? This is either a great novelty or a serious flaw.

2. Given this disconnect, why does the method perform so well empirically? Can you compare against a simple PPO baseline that has no access to any context at all (non-recurrent, observation-only input)?

3. Could the authors elaborate on the model's expected behavior when $M=1$? My understanding is that the context vector would become a single, fixed, random vector, making the agent's policy non-adaptive. Is this interpretation correct?
4. What is your choice of $d$ in your experiments?

5. Can the authors comment on the adaptation speed? How many timesteps does it take for the context vector $x_t$ to "flush out" the old OTE and converge to the new one after an opponent policy switch?

6. Can the authors provide the code, which is crucial for reproducibility?

---

> ### Author Response · Authors · 2025-11-19
> **Response to Reviewer VeR7 (1/12)**
>
> Thank you very much for your valuable feedback. In response to your comments, we would like to make the following clarifications and feedback. We hope our explanations and analyses can eliminate concerns and make you find our work stronger.
>
> ---
>
> ## **[w.r.t. W1,W2,Q1,Q3] Response to the concern about the Responder–Identifier “disconnect” and the case M = 1**
>
> We greatly appreciate the reviewer’s careful analysis and agree that, at first glance, the use of *random* Opponent Type Embeddings (OTEs) at test time may look like a severe mismatch between the Identifier and the Responder. Below we explain in detail why this is **not** a conceptual flaw, but a deliberate design choice that (i) avoids the limitations of fixed classifiers, and (ii) still allows the Responder to benefit from opponent-type information.
>
> We organize the response into three parts:
>
> 1. How the training protocol prevents the Responder from relying on a fixed “vector = opponent” dictionary.
> 2. How OSOM should be viewed as learning a **task code / belief embedding** that is meaningful within each prompt, rather than a global label.
> 3. Why the $M = 1$ setting is a degenerate special case, where opponent identification is vacuous and OSOM gracefully reduces to a standard RL agent.
>
> ### **1. Training-time OTE resampling: no global dictionary “$z_{\text{train},A}$ = Opponent A”**
>
> The reviewer’s concern assumes that during training, the Responder repeatedly sees the *same* opponent A paired with the *same* OTE vector $z_{\text{train},A}$, so that it can learn the association
>
> > “if the context equals this specific vector, then best-respond to A.”
> >
>
> If this were the case, then indeed reinitializing all OTEs at test time would break the learned mapping. However, this is **not** how OSOM is trained.
>
> As described in Sec. 3 and App. G, OSOM iteratively samples opponents from a large pool and, crucially, **resamples both the opponent subset and their OTEs during training**. More concretely:
>
> - At the beginning of each training segment, we sample a subset of opponent policies $\\{\pi^{-1,k}\\}_{k=1}^K$ from a large Train Set $\Pi^{\text{train}}$.
> - We then **randomly generate** a new set of OTEs $\mathcal{Z}^{\text{train}} = \\{z^{-1,k} \in \mathbb{R}^d\\}_{k=1}^K$ on the unit sphere with approximate orthogonality.
> - These OTEs are kept fixed only for a limited number of timesteps (3000 in our experiments), and then **discarded**; a new subset of opponents and a new set of OTEs are sampled.
>
> Thus, across training:
>
> - The same underlying opponent policy may be encountered multiple times, but **each time with a different randomly sampled OTE assignment**.
> - There is *no* persistent global vector $z_{\text{train},A}$ that always represents Opponent A throughout training.
>
> From the perspective of the Responder, any attempt to memorize a fixed mapping “this specific direction in $\mathbb{R}^d$ means Opponent A” would be statistically unstable: in future training segments, that same direction will be associated with different opponents (or not appear at all). Optimizing the expected return over many such randomized assignments inevitably encourages the Responder to **avoid** relying on any fixed global semantics of individual OTE directions.
>
> What remains stable across OTE resamplings is not “vector $\vec v$ = A”, but the following:
>
> > **Within each training segment (one prompt $\mathcal{Z}$), trajectories from the same opponent policy are consistently mapped by the Identifier to the same OTE, and thus produce a stable cluster of context vectors $x_t$; trajectories from different opponents form different clusters.**
> >
>
> This is enforced by the contrastive loss in Eq. (2), which encourages the predicted OTE $\hat z_t$ to be similar to the correct OTE and dissimilar to others, and by the sliding-window aggregation that averages selected OTEs into a compact semantic context $x_t$.
>
> In short, the Responder is **not** learning a global dictionary of the form $\text{global vector} \rightarrow \text{global opponent ID}$.
>
> Instead, it learns to interpret **per-prompt clusters** of context vectors that are consistent within a segment but randomly re-labeled across segments.

---

> > ### Author Response · Authors · 2025-11-19
> > **Response to Reviewer VeR7 (2/12)**
> >
> > ### **2. OSOM as learning a belief / task code, not a global label**
> >
> > A more principled way to understand OSOM is through the lens of **meta-RL and Bayesian belief summaries**.
> >
> > ### **2.1 Latent opponent type and amortized inference**
> >
> > Consider the opponent policy as a latent variable $\beta$ (e.g., indexing a rule-based policy or a particular parameter setting). The self-agent’s experience $\tau^1$ depends on $\beta$. OSOM’s architecture can be seen as performing **amortized inference** and control in this latent-variable environment:
> >
> > 1. **Encoder + Distillation.**
> >
> >     The encoder $f$ processes the self-agent’s trajectory (observations and actions) and is trained with an auxiliary decoder to predict the opponent’s observations and actions at each timestep. This distills the opponent policy into a latent state $e_t = f(o^1_t, a^1_t)$, which can be viewed as a sufficient statistic for $\beta$.
> >
> > 2. **Identifier + Random OTE prompt.**
> >
> >     Given a *current* set of OTEs $\mathcal{Z} = \\{z^{-1,k}\\} _{k=1}^K$, the Identifier is trained (*via contrastive learning*) to map the latent sequence $e _{0:t}$ (*Here, we only consider the $e$ of the current episode and omit the $e$ of the last $C$ episodes for simplicity*) to a predicted embedding $\hat z_t$ that is close to the OTE corresponding to the true opponent. Functionally, the Identifier can represent a mapping of the form:
> >
> >     $$
> >     \hat z _t(e _{0:t}, \mathcal{Z}) \approx \sum _{k=1}^K p _\psi(k \mid e _{0:t}) z^{-1,k},
> >     $$
> >
> >     i.e., an **embedding of a posterior distribution over opponent types** expressed in the current OTE basis. Here, the subscript $\psi$ simply denotes the learnable parameters of the Identifier network (Transformer + projection), which maps the encoded trajectory $e_{0:t}$ to a similarity-based softmax over the current OTE set. This is exactly what Eq. (3) implements in our paper; we only wrote it as $p_\psi$ in the discussion to emphasize that this is a parametric approximation to a posterior over opponent types.
> >
> > 3. **Context Aggregator.**
> >
> >     By averaging the selected OTEs over a sliding window of episodes, the Context Aggregator produces a context vector $x_t$ that is approximately a *smoothed* version of this posterior embedding over recent history. Episodes against the same opponent yield similar $x_t$, episodes against different opponents yield different $x_t$ clusters.
> >
> > 4. **Responder.**
> >
> >     Finally, the Responder $\pi^1(a^1 \mid o^1_t, x_t)$ takes the current observation and the context $x_t$ and learns, via online RL, to produce actions that maximize expected return against the sampled opponents.
> >
> >
> > From this perspective:
> >
> > - $x_t$ is *not* meant to be a “global ID of Opponent A” that is shared across all training and testing scenarios.
> > - Instead, it is a **belief / task code** that summarizes “which behavioral pattern we are currently facing” *within the current prompt*.
> >
> > This view is very similar in spirit to recurrent meta-RL (e.g., RL$^2$ [1]) and Algorithm Distillation [2]–style in-context RL:
> >
> > there, the RNN/Transformer hidden state is an unlabelled, task-specific code with no global semantics, but it is learned such that the policy conditioned on this code adapts effectively to each new task.
> >
> > In OSOM, we externalize part of this code into $x_t$, and we encourage the Identifier to structure this code via OTE prompts and contrastive learning so that:
> >
> > - Different opponent behaviors map to different “code clusters” **within each prompt**, and
> > - The Responder leverages these clusters to choose appropriate best responses.
> >
> > The **absolute numeric value** of the OTEs is irrelevant; what matters is the *jointly learned protocol* between the Identifier and the Responder, established through end-to-end optimization of the combined loss $J_{\text{distill}} + J_{\text{identify}} + J_{\text{respond}}$.
> >
> > [1] Duan Y, Schulman J, Chen X, et al. RL$^ 2$: Fast reinforcement learning via slow reinforcement learning. arXiv, 2016.
> >
> > [2] Laskin M, Wang L, Oh J, et al. In-context Reinforcement Learning with Algorithm Distillation. ICLR, 2022.

---

> > > ### Author Response · Authors · 2025-11-19
> > > **Response to Reviewer VeR7 (3/12)**
> > >
> > > ### **2.2 Why test-time random OTEs are in-distribution**
> > >
> > > Because during training we **continually resample** both opponent subsets and their random OTEs, the distribution of $(o_t, x_t)$ experienced by the Responder already includes:
> > >
> > > - Many different opponent behaviors, and
> > > - Many different random coordinate systems (OTE prompts) under which those behaviors are clustered.
> > >
> > > At test time, we again draw a new $\mathcal{Z}^{\text{test}}$ from the same OTE distribution and new opponents from $\Pi^{\text{test}}$. The generative process
> > >
> > > $$
> > > \beta \rightarrow \tau^1 \rightarrow e_{0:t} \rightarrow \hat z_t \rightarrow z_{\text{sel},t} \rightarrow x_t
> > > $$
> > >
> > > is exactly the same as during training, except that $\beta$ now comes from a disjoint Eval / All set of opponents. The Responder has been trained specifically to operate under such random prompts, so test-time OTE reinitialization is **not** a distributional shift, but part of the intended meta-RL regime.
> > >
> > > In particular, OSOM never assumes or requires that “the OTE used for Opponent A in training” should reappear at test time. The reviewer’s scenario in which $z_{\text{train},A}$ and $z_{\text{test},B}$ must somehow be matched is therefore not an invariance the method tries to satisfy. Instead, OSOM relies on:
> > >
> > > - The **structure** of how trajectories from the same opponent are embedded and clustered within a given prompt,
> > > - And the Responder’s ability (learned across many prompts) to exploit this clustering to select good actions.
> > >
> > > This is precisely why OSOM can handle open-set opponents: there is no requirement that test opponents share a fixed global ID or a fixed global embedding with any training opponent.

---

> > > > ### Author Response · Authors · 2025-11-19
> > > > **Response to Reviewer VeR7 (4/12)**
> > > >
> > > > ### **3. What happens when M = 1? Is the policy “non-adaptive”?**
> > > >
> > > > ### **3.1 Detailed analysis**
> > > >
> > > > We now address the reviewer’s question about the $M = 1$ case, where the test prompt contains only a single OTE $z_{\text{test},1}$.
> > > >
> > > > Mathematically, the reviewer is absolutely correct:
> > > >
> > > > - With $M = 1$, Eq. (3) yields $P(z^{-1,1}) = 1$, so the selected OTE is always $z^{-1,1}$.
> > > > - After sliding-window aggregation, the context vector $x_t$ indeed becomes (approximately) a **single fixed random vector** for that evaluation run.
> > > >
> > > > This observation is accurate, but it reflects the fact that the **problem setting itself** has collapsed:
> > > >
> > > > - If $\Pi^{\text{test}}$ contains only one type, there is no non-trivial opponent-identification problem.
> > > > - Any method that relies on distinguishing *multiple* opponent types must reduce to a standard RL agent in this degenerate setting.
> > > >
> > > > In our framework, this is exactly what happens:
> > > >
> > > > 1. The OTE-based context no longer carries information that differentiates between multiple candidate types, because there is only one type.
> > > > 2. OSOM therefore reduces to a Transformer-based RL policy that conditions on the self-agent’s observations (and the distillation-enriched latent states) and is trained via the RL objective $J_{\text{respond}}$. It remains **adaptive over time** in the usual sense of in-context RL (as recurrent policies are), but not via OTE-based type discrimination, since there is no type variability to exploit.
> > > >
> > > > So if “non-adaptive” means “does not switch between multiple OTE-based modes corresponding to different types”, then yes, with $M = 1$ there is nothing to switch between. However, if “non-adaptive” means “cannot adjust behavior based on observed trajectories”, then this is not the case: the policy still adapts via the usual RL mechanisms.
> > > >
> > > > ### **3.2 Additional empirical results**
> > > >
> > > > To directly address the reviewer’s request, we implemented the $M=1$ test case in the Partially-Observable Overcooked (POO) environment under the same protocol as Fig. 3, and report the additional bar *OSOM w/ M=1*.
> > > >
> > > > > See the results at [this anonymized link](https://ibb.co/2QPSCg6); we will also include this in the revision.
> > > > >
> > > >
> > > > Concretely, we keep the training setup unchanged, but at test time we force the prompt to contain only a single OTE. The results are as follows:
> > > >
> > > > - Across **Train Set**, **Eval Set**, and **All Set** configurations, **OSOM w/ M=1** performs **significantly worse** than the full OSOM with $M \ge 2$;
> > > > - At the same time, **OSOM w/ M=1** performs **on par with other meta-RL / opponent-modelling baselines** (Generalist, LIAMX, LILI) and slightly below PACE in POO.
> > > >
> > > > These empirical findings confirm the reviewer’s interpretation of the degenerate $M=1$ case:
> > > >
> > > > - The opponent-type context becomes a constant and no longer supports opponent-type-specific adaptation;
> > > > - Consequently, OSOM collapses to a generalist agent whose performance matches that of other methods that do not exploit an explicit opponent-type latent.
> > > >
> > > > At the same time, this behaviour is *consistent* with the intended role of the opponent-type latent in OSOM and does **not** indicate a conceptual flaw in the method. The key design choice in OSOM is to leverage a random, re-sampled OTE codebook and an explicit Identifier so that, in the non-degenerate regime $M \ge 2$, the model can (i) assign each opponent to a slot in the current prompt and (ii) condition on this slot to adapt its policy. Indeed, in all our main experiments with $M \ge 2$, OSOM substantially outperforms the aforementioned baselines, especially in the challenging POO environment. The new $M=1$ experiment thus supports the following picture:
> > > >
> > > > - In the *intended* open-set, multi-type setting ($M \ge 2$), OSOM’s explicit opponent-type latent is informative and yields clear performance gains;
> > > > - In the *degenerate* $M=1$ limit where there is only a single possible opponent type, the latent becomes uninformative by definition and OSOM gracefully degenerates into a standard meta-RL agent, with empirical performance aligning with other baselines.
> > > >
> > > > We will revise the main text to emphasize this meta-RL / belief-embedding viewpoint more clearly, so that the role of random OTEs and the Responder–Identifier interaction is not misunderstood as relying on a fragile global dictionary.
> > > >
> > > > ---

---

> > > > > ### Author Response · Authors · 2025-11-19
> > > > > **Response to Reviewer VeR7 (5/12)**
> > > > >
> > > > > ## **[w.r.t. W3] Response to the concern about the opponent information access during training**
> > > > >
> > > > > We thank the reviewer for pointing this out. We fully agree that the policy distillation objective $J_{\text{distill}}$ relies on *training-time* access to opponent trajectories, and that this assumption should be clearly and explicitly stated. Below we clarify (1) how and where this information is used, (2) why this is a standard and realistic assumption in opponent modeling, and (3) what happens when such privileged data is not available, as reflected in our ablation.
> > > > >
> > > > > ### **1. Distillation uses privileged information **only during training**, not at test time**
> > > > >
> > > > > In Sec. 3.1, we explicitly adopt the partially observable setting where, during testing, the self-agent **cannot** directly access the opponents’ information:
> > > > >
> > > > > > “OM typically assumes that its environment is partially-observable. This means that during testing, the self-agent cannot immediately observe information about opponents.”
> > > > > >
> > > > >
> > > > > At the same time, we posit a standard assumption that **during training**, opponent data (observations and actions) is often available from the simulator or offline logs:
> > > > >
> > > > > > “However, it is a reasonable assumption that opponent data is often accessible during training (Papoudakis et al., 2021; Gronauer & Diepold, 2022). Consequently, we can manage to distill the opponent policy into the self-agent’s representation, thereby enabling the self-agent to more effectively distinguish various opponent policies solely through its own observation–action sequence.”
> > > > > >
> > > > >
> > > > > Accordingly, the distillation objective is defined in Eq. (1), where $e_t = f(o^1_t, a^1_t)$ is an encoding of the *self-agent’s* observation–action pair. As summarized in Fig. 2, this objective is applied *only during training* to shape the encoder $f$. At test time, the deployed policy $\pi^1(a^1 \mid o^1, x)$ uses **only** the self-agent’s observations $o^1$ and the context $x$ built from predicted OTEs; opponent observations/actions are neither input nor required.
> > > > >
> > > > > Intuitively, $J_{\text{distill}}$ is an auxiliary supervised signal that encourages $f$ to encode, in $e_t$, information that is predictive of the opponent’s behavior. Once this representation is learned, **no privileged information is needed at test time**, as the encoder operates purely on the self-agent’s local trajectory.
> > > > >
> > > > > ### **2. Training-time opponent data is a standard assumption in OM / CTDE, and aligns with baselines**
> > > > >
> > > > > The reviewer is absolutely right that access to $(o^{-1}_t, a^{-1}_t)$ is a form of privileged information. Our intention is to place OSOM in the widely used **centralized training, decentralized execution (CTDE)** paradigm, commonly adopted in multi-agent RL and opponent modeling.
> > > > >
> > > > > Concretely:
> > > > >
> > > > > - In many simulated and real-world multi-agent domains, the training infrastructure (simulator, offline logs, or central controller) records the full trajectories of all agents. This is precisely the regime where prior OM works also exploit opponent trajectories during training.
> > > > > - For example, LIAM/LIAMX (Papoudakis et al., 2021) are defined as using *self* observations and actions to **reconstruct the opponent’s observations and actions via an auto-encoder**, thereby embedding the opponent policy into a latent space. This reconstruction obviously requires opponent data during training. GSCU and other baselines similarly rely on centralized access to multi-agent trajectories for offline embedding and response learning.
> > > > >
> > > > > Thus, OSOM does not assume more information than several of the representative baselines we compare to; rather, it uses opponent data in a different way — to distill opponent policies into a *self-only* representation that remains usable when opponent data is no longer available at test time.
> > > > >
> > > > > We will revise Sec. 3.1 and the limitations / discussion section to make this **CTDE assumption** more explicit and to emphasize that policy distillation relies on training-time privileged information.
> > > > >
> > > > > > The full OSOM relies on access to opponent trajectories during training, and that in scenarios where this is impossible, OSOM w/o Distill Loss (and potentially other weakly supervised variants) should be used instead.
> > > > > >

---

> > > > > > ### Author Response · Authors · 2025-11-19
> > > > > > **Response to Reviewer VeR7 (6/12)**
> > > > > >
> > > > > > ### **3. What happens if opponent data is not available at training time?**
> > > > > >
> > > > > > The reviewer correctly notes that Fig. 5 already includes “OSOM w/o Distill Loss”, demonstrating that the model can operate **without** the privileged distillation signal. In this variant:
> > > > > >
> > > > > > - We set $\alpha_1 = 0$ in Eq. (5), effectively removing $J_{\text{distill}}$, and train with only $J_{\text{identify}} + J_{\text{respond}}$.
> > > > > > - The encoder is then trained purely from the contrastive identification loss and the RL objective, using only the self-agent’s observations and actions.
> > > > > >
> > > > > > Empirically, we observe that:
> > > > > >
> > > > > > - **OSOM w/o Distill Loss remains functional** and retains most of the qualitative behavior of OSOM: it can still perform explicit opponent identification and achieves competitive response performance across environments, as the reviewer correctly pointed out.
> > > > > > - However, the full OSOM with distillation achieves some *higher* returns.
> > > > > >
> > > > > > We view this exactly as the reviewer suggests:
> > > > > >
> > > > > > - The reliance on training-time opponent trajectories is **not a hard prerequisite** for OSOM to work; the architecture and objectives $J_{\text{identify}} + J_{\text{respond}}$ are already sufficient to handle OSOs.
> > > > > > - Instead, $J_{\text{distill}}$ plays the role of a **privileged auxiliary task** that further improves sample efficiency and representation quality in the common CTDE regime.
> > > > > >
> > > > > > For applications where opponent trajectories are truly unavailable even during training, the appropriate variant is OSOM w/o Distill Loss, which our ablation shows is still effective. We will clarify this point in the revised manuscript.
> > > > > >
> > > > > > > Opponent policy distillation is an *optional, but beneficial* component operating under a centralized-training assumption. OSOM can also operate without this privileged signal, at the cost of some reduced performance.
> > > > > > >
> > > > > >
> > > > > > We hope this clarifies both the role of the distillation objective and the scope of our assumptions. We appreciate the reviewer’s suggestion and will make this limitation explicit in the final version.
> > > > > >
> > > > > > ---

---

> > > > > > > ### Author Response · Authors · 2025-11-19
> > > > > > > **Response to Reviewer VeR7 (7/12)**
> > > > > > >
> > > > > > > ## **[w.r.t. W4,Q4] Response to the concern about prior knowledge of $M$ and constraint on $d$**
> > > > > > >
> > > > > > > We thank the reviewer for raising these important points. We address them in turn.
> > > > > > >
> > > > > > > ### **1. Does OSOM require exact prior knowledge of $M$ at test time?**
> > > > > > >
> > > > > > > In the current experimental setup (Sec. 5), we evaluate OSOM on finite **benchmark test sets** $\Pi^{\text{test}} = \\{\pi^{-1,m}\\}_{m=1}^M$ of opponent policies. For these benchmarks, the size $|\Pi^{\text{test}}|$ is of course known in advance, and we set $M = |\Pi^{\text{test}}|$ when generating the test-time prompt $\mathcal{Z}^{\text{test}}$ (Algo. 2 / Fig. 7). This is an **evaluation choice** that allows us to report explicit identification accuracy per opponent type.
> > > > > > >
> > > > > > > However, this does **not** mean that OSOM fundamentally requires exact prior knowledge of “the true number of opponent types that may ever appear in the environment”. From a methodological standpoint, $M$ is best interpreted as:
> > > > > > >
> > > > > > > > **a capacity hyperparameter: the number of “slots” (OTE prototypes) that the Identifier can assign to distinct opponent behaviors within a given prompt.**
> > > > > > > >
> > > > > > >
> > > > > > > In the benchmark experiments, we simply choose to set this capacity equal to the test-set size, so that each opponent policy can, in principle, occupy its own slot. In a more realistic open-world scenario where the total number of potential opponent types is large or unknown, one would:
> > > > > > >
> > > > > > > - Choose $M$ as an **upper bound / design budget** on how many distinct types one wishes (or can afford) to distinguish at once;
> > > > > > > - Allow multiple true opponent policies to share the same OTE (i.e., be clustered into the same latent type) when their number exceeds $M$.
> > > > > > >
> > > > > > > In such a regime, OSOM still operates as intended: the Identifier clusters observed behaviors into at most $M$ latent types and the Responder learns best responses at the granularity supported by this capacity. What is lost is not correctness, but resolution: some distinct opponents will be grouped together if they exceed the available capacity, which is inevitable for any finite-dimensional representation.
> > > > > > >
> > > > > > > We will revise Sec. 5 to clarify that setting $M = |\Pi^{\text{test}}|$ is a **benchmark-specific choice** for evaluation purposes and that, in practice, $M$ should be understood as a tunable capacity parameter rather than requiring exact prior knowledge of all possible test-time opponent types.

---

> > > > > > > > ### Author Response · Authors · 2025-11-19
> > > > > > > > **Response to Reviewer VeR7 (8/12)**
> > > > > > > >
> > > > > > > > ### **2. Is $d \ge M$ a hard scalability constraint?**
> > > > > > > >
> > > > > > > > In Appendix F, we describe our procedure for generating OTEs and we explicitly state that we create “random, pairwise orthogonal OTEs on the unit sphere” using random sampling followed by Gram–Schmidt orthogonalization. As the reviewer correctly notes, **exact pairwise orthogonality** of $M$ vectors in $\mathbb{R}^d$ is only possible when $d \ge M$.
> > > > > > > >
> > > > > > > > We would like to emphasize three points:
> > > > > > > >
> > > > > > > > ### **2.1 In all our experiments, $d > M$**
> > > > > > > >
> > > > > > > > In our implementations, the OTE dimension $d$ (and the dimension of the latent layer that outputs $\hat z_t$) is set to 64 or 128 depending on the environment. The maximum number of opponent types in any test set (“All Set” in KP) is 50, i.e., $M \le 50$.
> > > > > > > >
> > > > > > > > Thus, in all experimental regimes we consider, we have **$d \ge M$ with an enough margin**, and exact orthogonality is straightforward to achieve via Gram–Schmidt. The apparent constraint does not bind for the scales we study.
> > > > > > > >
> > > > > > > > ### **2.2 Orthogonality is a numerical convenience, not a theoretical requirement**
> > > > > > > >
> > > > > > > > The reason we encourage OTEs to be orthogonal is purely **optimization-driven**: as stated in our paper,
> > > > > > > >
> > > > > > > > > “Orthogonality guarantees that a predicted vector $\hat z_t$ can have a high similarity score with one target OTE without inadvertently having a high score with others, which sharpens the optimization signal and prevents ambiguity during identification.”
> > > > > > > > >
> > > > > > > >
> > > > > > > > In other words, orthogonality improves the margin and conditioning of the contrastive objective in Eq. (2). However, the method **does not fundamentally require** strict pairwise orthogonality:
> > > > > > > >
> > > > > > > > - If $M > d$, we can simply skip the Gram–Schmidt step and sample $M$ random unit vectors on the sphere.
> > > > > > > > - In high-dimensional spaces, such random vectors are already nearly orthogonal in expectation (their dot products concentrate near zero), and the contrastive loss remains well-defined and effective, as in many contrastive learning and metric-learning methods where the number of classes exceeds the embedding dimension.
> > > > > > > >
> > > > > > > > In this sense, the condition $d \ge M$ is a **sufficient condition** for exact orthogonality in our current implementation, not a necessary condition for OSOM to function. We will make this clear in App. F by rephrasing “should be pairwise orthogonal” as a favorable design choice rather than a hard algorithmic requirement, and by explicitly noting that for $M > d$ one can use random (non-orthogonal) prototypes instead.
> > > > > > > >
> > > > > > > > ### **2.3 The embedding dimension $d$ is a capacity hyperparameter**
> > > > > > > >
> > > > > > > > More fundamentally, $d$ is itself a standard **capacity hyperparameter**, much like the width of a neural network:
> > > > > > > >
> > > > > > > > - If one wishes to distinguish a very large number of mutually dissimilar opponent types *within a single prompt*, then $d$ must eventually grow to provide enough representational capacity.
> > > > > > > > - This is not unique to OSOM; any finite-dimensional embedding has a finite capacity to separate many distinct types. OSOM simply exposes this trade-off explicitly through $d$ and $M$.
> > > > > > > >
> > > > > > > > In practice, setting $d = 128$ or $d = 256$ already allows for hundreds of well-separated OTEs, while the computational cost scales linearly in $d$. For the environments we consider, our choice of $d \in {64,128}$ (with $M \le 50$) is sufficient, and we did not observe any bottleneck related to this constraint.
> > > > > > > >
> > > > > > > > We will add a brief discussion in the Limitations / Discussion section to acknowledge this.
> > > > > > > >
> > > > > > > > > As with any embedding-based method, scaling OSOM to extremely large numbers of distinct opponent types in a single prompt will require increasing $d$, and that exploring more sophisticated OTE parameterizations (e.g., learnable codebooks, hierarchical OTEs) is an interesting direction for future work.
> > > > > > > > >
> > > > > > > >
> > > > > > > > We will revise the paper to more clearly communicate that $M$ should be understood as a tunable capacity parameter and explicitly discuss the capacity/scalability trade-offs in terms of $M$ and $d$.
> > > > > > > >
> > > > > > > > ---

---

> ### Author Response · Authors · 2025-11-19
> **Response to Reviewer VeR7 (9/12)**
>
> ## **[w.r.t. Q2] Comparison against a simple PPO baseline**
>
> We appreciate the reviewer’s suggestion to include a non-recurrent, observation-only PPO baseline (“naive PPO”) that does not use any context at all. Following this request, we added a PPO baseline under the **same experimental protocol as Fig. 6**, with the following design:
>
> - The PPO agent receives only the current self observation $o^1_t$ as input (no recurrent state, no cross-episode context, no access to opponent information);
> - The network architecture (MLP size, activation functions, etc.) and training hyper-parameters are matched to those used in our Responder, except that the context input is removed;
> - We train and evaluate this agent on the same Train Set and Eval Set of opponents as all other methods.
>
> > See the results at [this anonymized link](https://ibb.co/Cp3sBmGF); we will also include this in the revision.
> >
>
> The resulting training curves show that:
>
> - Across both **Train Set** and **Eval Set**, the **PPO baseline is consistently weaker than or comparable to the other baselines**, and
> - It is **always dominated by the Generalist agent**, which shares a similar backbone but incorporates recurrent state to summarize interaction history.
>
> These findings are fully consistent with the structure of our problem and the behaviour we would theoretically expect:
>
> 1. **Partial observability and non-stationarity.** In our settings (KP, POO, PPW), the self-agent operates under partial observability, and the environment dynamics are non-stationary from the agent’s perspective because the opponent policy switches every $H$ episodes. An observation-only, feedforward PPO necessarily treats the problem as a stationary MDP over $o^1_t$ and has no mechanism to infer or represent “which opponent type am I currently facing?”. Its value estimates and gradients are therefore computed on a mixture over opponent types, which naturally leads to a single compromise policy rather than type-specific responses.
> 2. **Generalist vs. PPO: the role of context even without explicit opponent modelling.** The Generalist baseline, in contrast, augments PPO with recurrent state (or equivalent cross-episode history) and thus can at least partially encode patterns in the recent interaction history. Empirically, this makes Generalist stronger than naive PPO in both Train and Eval sets. This gap confirms that some form of temporal/contextual information is necessary in our open-set, non-stationary, partially observable environments; simply applying PPO to the raw observation stream is insufficient.
> 3. **OSOM vs. Generalist: the role of explicit opponent-type context.** On top of this, OSOM further introduces an explicit opponent-type latent via OTEs, the Identifier, and the context aggregator. This allows the agent not only to use history, but also to *factorize* history into a type representation and condition its policy on this type. In all our main experiments, OSOM significantly outperforms both Generalist and other opponent-modelling baselines on Train, Eval, and All Sets. The new PPO baseline therefore fits naturally into a hierarchy:
>
>     $$
>     \text{PPO (no context)} \le \text{Generalist (history only)} < \text{OSOM (history + explicit opponent type)}.
>     $$
>
>     This ordering, observed empirically, aligns exactly with what one would expect from the design: removing all context produces the weakest agent; adding generic context (Generalist) improves robustness; and adding explicit, open-set opponent-type context (OSOM) yields the strongest performance.
>
>
> In summary, the naive PPO baseline confirms that a non-recurrent, observation-only agent is not competitive in the open-set opponent setting we study and that the gains of OSOM are not due to trivial architectural choices, but to its ability to exploit history and explicit opponent-type context.
>
> ---

---

> ### Author Response · Authors · 2025-11-19
> **Response to Reviewer VeR7 (10/12)**
>
> ## **[w.r.t. W5,Q5] Response to the concern about adaptation speed and average pooling**
>
> The Context Aggregator is not meant to be a sophisticated sequence model on its own, but a **simple, interpretable way to transform a sequence of identified OTEs into a belief-like context**. Recall that all OTEs are **pairwise orthogonal and lie on the unit sphere**.  Under this design, average pooling has a very concrete meaning:
>
> - At each timestep, the Identifier selects one OTE $z^{-1,k}$ from the prompt set $Z^{\text{test}}$ that best matches the opponent’s behavior.
> - Over a window of timesteps, the set of selected OTEs behaves like samples from a categorical “opponent-type belief” over the orthogonal basis $\{z^{-1,k}\}$.
> - Averaging these OTEs therefore gives a vector $x_t \approx \sum_k w_k z^{-1,k}$, where $w_k$ is approximately the empirical frequency of selecting type $k$ in the recent history.
>
> Because the OTEs form an (approximately) orthogonal basis with equal norm, the inner product $\langle x_t, z^{-1,k}\rangle$ directly reflects the **confidence** that the current opponent behaves like type $k$. The Responder is trained end-to-end under this semantics, and learns to condition its policy on such belief-like mixtures.
>
> ### **1. Adaptation speed after an opponent switch**
>
> ### **1.1 Detailed analysis**
>
> We now analyze how fast the average-pooling context can adapt when the opponent changes its policy.
>
> For simplicity, consider two opponent types A and B with corresponding OTEs $z_A$ and $z_B$. Assume that the Identifier is correct with high probability, so that all selected OTEs coincide with $z_A$ (resp. $z_B$) when the true opponent is A (resp. B). Let the sliding window size be $C$ episodes (as in our implementation), and let the episode horizon be $T$.
>
> Suppose the opponent uses type A for a long period and then **switches abruptly** to type B at the beginning of episode $h^*$. We index episodes after the switch by $j = 0,1,2,\dots$, where $j=0$ is the first B-episode.
>
> At the end of episode $j$, the Context Aggregator averages the selected OTEs from:
>
> - the current episode $h^* + j$, and
> - the previous $C$ episodes, i.e., episodes $(h^* + j - C), \dots, (h^* + j - 1)$.
>
> Assuming that before $h^*$ the opponent was always A, we can count how many episodes in this window belonged to each type:
>
> - For $0 \le j < C$:
>     - there are $C - j$ episodes with opponent A in the window,
>     - and $j + 1$ episodes with opponent B;
> - For $j \ge C$:
>     - all $C+1$ episodes in the window already belong to B.
>
> Since each episode has approximately the same length, the relative weight of A vs. B in the average is proportional to the number of episodes of each type. Therefore, for $0 \le j \le C$, the aggregated context at the end of episode $h^* + j$ satisfies $x^{(j)} \approx \frac{C - j}{C+1} z_A + \frac{j+1}{C+1} z_B$.

---

> > ### Author Response · Authors · 2025-11-19
> > **Response to Reviewer VeR7 (11/12)**
> >
> > This gives a precise characterization of adaptation speed:
> >
> > - The **weight of the new opponent type B increases linearly** with the number of episodes after the switch: $w_B(j) = \frac{j+1}{C+1}$.
> > - After roughly $\frac{C}{2}$ episodes with the new opponent, we already have $w_B(j) \gtrsim 0.5$, meaning that the context is dominated by B and the Responder primarily reacts to the new type.
> > - After $j = C$ episodes, we obtain $x^{(C)} \approx z_B$, i.e., the historical influence of A is completely **flushed out** from the sliding window.
> >
> > In timesteps, this corresponds to at most about $(C+1)\cdot T$ steps after the switch for the context to coincide with the new type in the idealized case. In practice, adaptation is even smoother at the timestep level because the window already starts incorporating OTEs from the new opponent within the very first post-switch episode.
> >
> > Importantly, the **same sliding-window mechanism is used throughout training**, where we explicitly simulate non-stationary opponents that change their policies over time.  The Responder is thus reinforced to exploit the gradual change in $x_t$ and to adapt its behavior accordingly within roughly $C$ episodes after each switch.
> >
> > ### **1.2 Empirical evidence**
> >
> > Consistent with this analysis, the empirical per-episode curves in Fig. 4 for POO and PPW also indicate that OSOM adapts on the order of $\frac{C}{2}$ episodes after each opponent switch. In both environments we set the sliding-window size for context aggregation to $C = 5$ episodes.  During testing, the non-stationary opponent switches its policy every $H = 5$ episodes in POO and PPW.  In Fig. 4, one can observe that after each switch the OSOM curve typically recovers from the transient drop and approaches its near steady performance level within roughly $2\text{–}3$ episodes, i.e., around $\frac{C}{2}$. This is precisely the regime where our theoretical expression $x^{(j)} \approx \frac{C - j}{C+1} z_A + \frac{j+1}{C+1} z_B$ predicts that the new opponent’s type already dominates the context window. In contrast, several baselines either recover more slowly or exhibit more oscillatory behavior across switches, suggesting that their implicit context encodings adapt less efficiently than OSOM’s explicit OTE-based belief representation.

---

> ### Author Response · Authors · 2025-11-19
> **Response to Reviewer VeR7 (12/12)**
>
> ### **2. Why do we use average pooling?**
>
> We agree that **discounted averaging** (e.g., an exponential moving average) is a reasonable alternative. Conceptually, our current design with a sliding window and uniform weights can be viewed as a **finite-horizon analogue** of a discounted filter:
>
> - The window size $C$ controls the **effective memory length**: a smaller $C$ leads to faster adaptation but more sensitivity to noisy identifications, while a larger $C$ yields more stable beliefs but slower adaptation.
> - In contrast, an exponential discount can be approximated by a sliding window with a certain effective horizon, but it never fully discards old information: the influence of previous opponents decays exponentially but remains non-zero indefinitely.
>
> In our open-set, piecewise-stationary setting, opponents may switch abruptly between **semantically distinct policies** drawn from a heterogeneous pool. In such cases, it is preferable that the agent **eventually forgets** a past regime entirely, rather than carrying a long exponential tail of outdated information. The sliding-window average provides exactly this property while remaining extremely simple and easy to tune.
>
> From an implementation perspective, we opted for this simplest variant because:
>
> 1. It fits naturally with orthogonal OTEs and yields a very clean interpretation of $x_t$ as a mixture over opponent types;
> 2. It already leads to substantial and consistent improvements over strong OM baselines across all three environments (KP, POO, PPW), as shown in Fig. 3.
>
> Nevertheless, we appreciate the reviewer’s suggestion. Discounted averaging is fully compatible with OSOM and can be implemented as a drop-in replacement for the Context Aggregator. Exploring such variants and their impact on the speed–stability trade-off is an interesting extension, which we plan to investigate in future work.
>
> ---
>
> ## **[w.r.t. Q6] Code of OSOM**
>
> We thank the reviewer for highlighting the importance of reproducibility.
>
> We would like to clarify that the **complete implementation code for our OSOM framework is already included in the supplementary materials** submitted with our manuscript.
>
> Furthermore, we are fully committed to open science and will make the entire codebase publicly available in an open-source repository upon the acceptance of our paper.
>
> ---
>
> All your questions and feedback have greatly contributed to improving our manuscript. With the valuable input from you and all other reviewers, our paper has undergone significant changes. We welcome further comments from you and will promptly consider your suggestions for revisions. If you feel that we have addressed your concerns and the quality of the paper has improved, we hope you will reconsider your rating.

---

> > ### Comment · Reviewer_VeR7 · 2025-11-28
> > **Thanks for response. Some follow-up questions.**
> >
> > I thank the authors for their detailed response. The rebuttal has convincingly addressed my concerns and corrected my understanding of the methodology. I now find the proposed method novel and intuitive, and the results appear more justified than in my initial reading.
> >
> > However, I still have a few concerns to note:
> >
> > 1. The figures currently use "training iterations" as the x-axis. This metric can be misleading if different algorithms consume different numbers of timesteps per iteration. Standard practice in RL is to use "total environment timesteps" to ensure a fair comparison of sample efficiency.
> >
> > 2. While including the cumulative discounted reward $J_{respond}$ in the optimization objective is valid, stating that it is maximized via direct gradient ascent is technically inaccurate for most standard RL settings, as environment dynamics are typically non-differentiable. I suggest revising the claim in Algorithm 1 (Line 1177) to reflect that this is handled via policy gradients (or similar estimators) rather than direct differentiation.
> >
> > 3. Regarding the response (8/12) that "Orthogonality is a numerical convenience, not a theoretical requirement": unless you provide experimental evidence using highly non-orthogonal vectors to support this, I recommend retaining the original wording that vectors "should be pairwise (nearly) orthogonal." As discussed, even random sampling in high dimensions yields approximate orthogonality, so the stronger claim regarding non-orthogonality seems unsupported.
> >
> > 4. Should the Train Set be diverse enough? If so, how do you find these opponents in the Train Set? If there are policies in the Train Set that are similar and intuitively it should not be expressed by orthogonal vectors, how will it affect the performance?
> >
> > I will raise my score to 6 to reflect my current evalutaion.

---

> > > ### Author Response · Authors · 2025-12-02
> > > **Response to the Follow-up Comment of Reviewer VeR7 (1/4)**
> > >
> > > We thank the reviewer again for the very constructive follow-up and for raising the score to **6**. We are glad that our previous response has resolved the main conceptual concerns.
> > >
> > > Below we respond to the remaining points one by one.
> > >
> > > ---
> > >
> > > ## **[w.r.t. Reply Q1] Using “training iterations” vs. “environment timesteps”**
> > >
> > > We fully agree with the reviewer that **environment timesteps** are the standard and more informative x-axis for reporting **sample efficiency** in RL.
> > >
> > > In our current implementation, all OM approaches share **exactly the same PPO training pipeline and rollout budget per iteration**, including batch size, number of update epochs, and mini-batches (see Tables 2–4 in Sec. G), so one “training iteration” already corresponds to the same number of environment interactions for all methods. This is why the relative ordering is not affected in the present plots.
> > >
> > > That said, we fully agree that plotting performance explicitly **against total timesteps** is clearer and more standard. In the revision, we will:
> > >
> > > - Re-plot all learning curves (Figs. 5–6 and the new figures) with **“environment timesteps”** on the x-axis.
> > > - Explicitly state in Sec. 4.1 that all methods are trained with the same number of environment interactions, so the comparison is fair in terms of sample efficiency.
> > >
> > > ---
> > >
> > > ## **[w.r.t. Reply Q2] “Direct gradient ascent” vs. policy gradients on $J_{\text{respond}}$**
> > >
> > > We thank the reviewer for pointing out this imprecise wording. Algorithm 1 and Line 1177 currently say that the cumulative discounted reward $J_{\text{respond}}$ is maximized via “gradient ascent”, which is indeed not technically accurate in standard RL settings where the environment dynamics are non-differentiable.
> > >
> > > In practice, as described in Sec. F, **all OM approaches, including OSOM, are optimized with PPO** (clip ratio, entropy coefficient, GAE, etc. are provided in Sec. G). That is, we maximize $J_{\text{respond}}$ **via policy-gradient estimators** implemented by PPO, not by differentiating through the environment.
> > >
> > > We will fix this in the revision by:
> > >
> > > - Stating that “$J_{\text{respond}}$ is maximized via policy-gradient methods (PPO)” rather than “gradient ascent” in Algorithm 1.
> > > - Clarifying in Sec. 3.3 that the optimization of $J_{\text{respond}}$ is done through PPO, consistent with the implementation details in Sec. F.
> > >
> > > ---
> > >
> > > ## **[w.r.t. Reply Q3] Orthogonality of opponent type embeddings (OTEs)**
> > >
> > > We appreciate the reviewer’s nuanced comment here and agree with the suggested wording.
> > >
> > > In our implementation, OTEs are generated as **random, pairwise orthogonal unit vectors** via a standard Gram–Schmidt procedure followed by normalization (Sec. F.1). In the rebuttal we had informally remarked that orthogonality is “numerical convenience rather than a theoretical requirement”, which is philosophically true in the sense that OSOM does not rely on any closed-form analytic property of orthogonal bases. However, as the reviewer correctly notes:
> > >
> > > - We do **not** provide dedicated experiments with highly non-orthogonal OTEs; and
> > > - In high dimensions, random vectors will in practice be almost orthogonal anyway.
> > >
> > > To avoid overstating what is empirically supported, in the revised paper we will:
> > >
> > > - **Revert to and keep** the original wording that OTEs “should be pairwise (nearly) orthogonal”
> > >
> > > We thank the reviewer for the careful reading and for helping us sharpen this statement.
> > >
> > > ---

---

> > > > ### Author Response · Authors · 2025-12-02
> > > > **Response to the Follow-up Comment of Reviewer VeR7 (2/4)**
> > > >
> > > > ## **[w.r.t. Reply Q4] Train Set diversity and similar policies with orthogonal OTEs**
> > > >
> > > > We now address the core conceptual question:
> > > >
> > > > > Should the Train Set be diverse enough? If so, how do you find these opponents in the Train Set? If there are policies in the Train Set that are similar and intuitively it should not be expressed by orthogonal vectors, how will it affect the performance?
> > > > >
> > > >
> > > > We fully agree that **Train Set diversity is crucial** in an open-set opponent modeling problem, just as in prior OM and meta-RL work. Below we clarify (i) how we construct diverse yet interpretable opponent pools, and (ii) how OSOM behaves when some policies in this pool are similar but represented by orthogonal OTEs.
> > > >
> > > > ### **1. What “diverse enough Train Set” means for OSOM**
> > > >
> > > > Conceptually, OSOM learns two things during training:
> > > >
> > > > 1. A mapping from **interaction histories** (via the distilled latent states $e_t$) to **opponent types** (via the OTEs and contrastive identification objective).
> > > > 2. A mapping from **(current observation, context vector)** to **best-response actions** via $J_{\text{respond}}$.
> > > >
> > > > For this meta-knowledge to generalize to unseen test opponents, the Train Set should:
> > > >
> > > > - **Cover the main modes of behavior** that are likely to be encountered at test time, and
> > > > - Include **variations within each mode**, so that the model can learn invariances and robust best responses rather than overfitting to a few specific policies.
> > > >
> > > > In other words, the requirement is not that the Train Set be “perfectly exhaustive”, but that it spans a **rich and human-interpretable basis** of opponent behaviors. This is precisely why we chose **rule-based, parameterized opponent families with clear semantic axes**, instead of arbitrary neural policies.

---

> > > > > ### Author Response · Authors · 2025-12-02
> > > > > **Response to the Follow-up Comment of Reviewer VeR7 (3/4)**
> > > > >
> > > > > ### **2. How we construct diverse opponent pools in practice**
> > > > >
> > > > > To operationalize this idea of diversity and interpretability, we explicitly design the opponent pools as follows (Sec. 4.1 and Sec. E):
> > > > >
> > > > > **Kuhn Poker (KP).**
> > > > >
> > > > > After eliminating dominated strategies for player P2, the opponent policy can be parameterized by **two key probabilities** $\eta$ and $\xi$: betting with a Queen after P1 has bet, and betting with a Jack after P1 has passed. By sampling different values of $(\eta, \xi)$, we obtain a continuum from extremely conservative to highly aggressive, bluff-heavy policies. In our experiments, we:
> > > > >
> > > > > - Sample **40** distinct P2 policies to form the **Train Set**, and
> > > > > - Sample **10** additional P2 policies to form the **Eval Set** (unseen during training).
> > > > >
> > > > > These policies differ in meaningful ways (different bluff/call tendencies, different reactions to opponent bets), giving a **dense and semantically structured coverage** of the opponent space.
> > > > >
> > > > > **Partially-Observable Overcooked (POO).**
> > > > >
> > > > > Following Ma et al. (2024), we generate **rule-based partners** that each have a strong preference for a specific recipe (e.g., Tomato & Onion Salad). The opponent:
> > > > >
> > > > > - Is spatially constrained to one side of the kitchen and
> > > > > - Interacts only with ingredients and dishes relevant to its preferred recipe.
> > > > >
> > > > > On top of this, we introduce **two continuous parameters** to diversify low-level behavior:
> > > > >
> > > > > - $P_{\text{nav}}$: preference for lateral vs. vertical movements when multiple shortest paths exist;
> > > > > - $P_{\text{act}}$: probability of taking a random action instead of the task-optimal action.
> > > > >
> > > > > By varying the recipe preference and $(P_{\text{nav}}, P_{\text{act}})$, we obtain partners that are **systematically different** in both high-level coordination style and low-level noise/efficiency, while still being human-interpretable.
> > > > >
> > > > > **Predator–Prey with Watchtowers (PPW).**
> > > > >
> > > > > Here we follow the modified PPW setting of Ma et al. (2024):
> > > > >
> > > > > - **Predator opponents** each have a strong preference for one of the two preys; under full observation they will *always* pursue their preferred prey.
> > > > > - **Prey opponents** come from **8 distinct movement patterns**, each moving back and forth along a specific path in the environment (paths `1`-`8` in Fig. 8c). We split these into:
> > > > >     - 4 paths (blue) for the **Train Set**, and
> > > > >     - 4 new paths (red) for the **Eval Set**.
> > > > >
> > > > > We then form **joint opponents** as combinations:
> > > > >
> > > > > - Train Set: **16** combinations, each consisting of one predator policy and two **training** prey policies.
> > > > > - Eval Set: **24** combinations, each consisting of one predator policy and two **unseen** prey policies.
> > > > >
> > > > > Thus, in PPW, each “opponent type” for OSOM is a **joint configuration of predator preference + prey movement patterns**, representing qualitatively different spatial dynamics and coverage patterns.
> > > > >
> > > > > Across all three environments, the opponent pools are therefore:
> > > > >
> > > > > - **Large and diverse**, explicitly stated in Sec. 4.1 (“a relatively large and diverse opponent pool”),
> > > > > - Constructed with **human priors** so that each type has a **clearly distinct behavioral preference** (recipe preference, path preference, bluff tendency, etc.), and
> > > > > - Split into Train/Eval in a way that **Eval policies are systematically different but structurally related** to Train policies.
> > > > >
> > > > > This is exactly the kind of “diverse but interpretable” Train Set that a principled open-set opponent modeling approach requires.

---

> > > > > > ### Author Response · Authors · 2025-12-02
> > > > > > **Response to the Follow-up Comment of Reviewer VeR7 (4/4)**
> > > > > >
> > > > > > ### **3. What if some Train policies are similar but assigned orthogonal OTEs?**
> > > > > >
> > > > > > The reviewer raises an important subtlety:
> > > > > >
> > > > > > > If there are policies in the Train Set that are similar and intuitively it should not be expressed by orthogonal vectors, how will it affect the performance?
> > > > > > >
> > > > > >
> > > > > > There are two key clarifications:
> > > > > >
> > > > > > 1. **OTE geometry encodes identities, not metric similarity.**
> > > > > >
> > > > > >     In OSOM, OTEs act as **type labels embedded in $\mathbb{R}^d$**. They are the keys over which the Transformer performs attention and the targets in the contrastive identification loss. Their role is analogous to *one-hot class labels* in representation learning:
> > > > > >
> > > > > >     - Orthogonality ensures a **well-conditioned contrastive objective** and makes it easier to distinguish types during training.
> > > > > >     - However, **semantic similarity between policies is not encoded in the raw OTE geometry**. Instead, it emerges in the learned mapping from history $e_{0:t}$ to predicted OTE $\hat z_t$ and in the Responders’ policy over $(o_t, x_t)$.
> > > > > >
> > > > > >     In other words, two similar opponent policies can be assigned orthogonal OTEs without contradiction: the *similarity* is captured by the **behavior of the learned networks**, not by forcing their OTEs to be close.
> > > > > >
> > > > > > 2. **Redundant or very similar Train policies are a benign case for OSOM.**
> > > > > >
> > > > > >     Suppose, hypothetically, that we include two Train policies $\pi^{(a)}$ and $\pi^{(b)}$ that are **so similar that they are effectively indistinguishable** under the given observation and time horizon. Any OM method (not just OSOM) will then face an **information-theoretic limit**: it cannot reliably tell them apart from interaction histories alone.
> > > > > >
> > > > > >     In OSOM, this situation leads to a **benign behavior**:
> > > > > >
> > > > > >     - The Identifier will sometimes confuse $\pi^{(a)}$ and $\pi^{(b)}$, but the Context Aggregator averages the selected OTEs across episodes.
> > > > > >     - The Responder is trained via $J_{\text{respond}}$ to maximize return; if $\pi^{(a)}$ and $\pi^{(b)}$ require **essentially the same best response**, PPO will naturally learn **very similar policies** for the corresponding context vectors.
> > > > > >     - Functionally, OSOM will treat these two types as **one effective response mode**, even though their OTEs are orthogonal. This may waste some capacity in the type space, but does **not** degrade performance in a catastrophic way.
> > > > > >
> > > > > >     Conversely, if $\pi^{(a)}$ and $\pi^{(b)}$ are similar but still distinguishable (e.g., slightly different preferences), then assigning them orthogonal OTEs simply asks OSOM to **treat them as two distinct types**, and the Responder will learn slightly different best responses as appropriate.
> > > > > >
> > > > > >
> > > > > > Importantly, in our **actual experiments**, we intentionally avoid constructing nearly identical opponent policies. As stated in Sec. 4.1:
> > > > > >
> > > > > > > “For POO and PPW, we follow Ma et al. (2024) by using rule-based methods to construct opponent policies with clearly distinct preferences.”
> > > > > > >
> > > > > >
> > > > > > Thus, the practical regime we evaluate is precisely the one where:
> > > > > >
> > > > > > - Each opponent type corresponds to a **meaningfully distinct strategic preference**, and
> > > > > > - Orthogonal OTEs serve as a **clean and well-conditioned label space** for contrastive training, rather than as a metric embedding of policy similarity.
> > > > > >
> > > > > > The strong empirical results of OSOM—both in overall performance across Train/Eval/All sets (Fig. 3) and in per-opponent performance curves (Fig. 4)—demonstrate that, under such diverse opponent pools, OSOM can **both identify and respond** effectively to a wide range of behaviors.
> > > > > >
> > > > > > ---
> > > > > >
> > > > > > Once again, we sincerely thank the reviewer for the detailed and thoughtful comments. We are happy that our previous response clarified the main conceptual misunderstandings, and we appreciate the concrete suggestions on figures, wording, and assumptions, which we will incorporate in the revised manuscript.

---

### Official Review · Reviewer_f7DN · 2025-10-30

**Soundness:** 3
**Presentation:** 3
**Contribution:** 3
**Rating:** 6
**Confidence:** 2

**Summary:**

This paper proposes a novel end-to-end neural architecture involving a transformer component for opponent modeling. The first component is an opponent policy distilling network which maps the agent's past trajectories into a latent embedding. And then, it utilizes a transfomer network, which first includes K randomly sampled opponent-type-embeddings and the embeddings distilled from past trajectories, and output a predicted opponent embedding. This network is trained via contrastive learning. And then, all these predicted type-embedding are aggregated as a state representation for training an RL best response. This design effectively takes opponent-modeling as an inductie biases for this end-to-end training loop.

**Strengths:**

The design is novel. The experimental results are comprehensive.

**Weaknesses:**

It was not clear how to ensure the opponent test set is diverse. The abalation studies can be improved.

**Questions:**

The end-to-end design of OSOM is very amazing. However, I have some detailed questions:

1. About the transformer: are you applying full causal mask for the whole sequence, and what kind of positional embedding are you using? I.e., is the ordering of these K opponent embeddings matter?

2. The K opponent embeddings are sampled without imposing further structure. However, it is not very clear to me why this should be the way. Have you done ablation study are the way you sample these embeddings?

3. Not very clear to me in the experiments how were the training-time and test-time opponents constructured. Could you elaborate more?

---

> ### Author Response · Authors · 2025-11-19
> **Response to Reviewer f7DN (1/5)**
>
> Thank you very much for your recognition of our paper's writing, experimental design, and results, as well as for the valuable feedback you provided. In response to your comments, we would like to make the following clarifications and feedback. We hope our explanations and analyses can eliminate concerns and make you find our work stronger.
>
> ---
>
> ## **[w.r.t. Weakness,Q3] Response to the concern about how we construct opponent policies**
>
> We apologize for not describing the construction of the opponent pools in sufficient detail. We summarize here how we design a large and *semantically* diverse opponent space in each environment, and how the Train/Eval splits are obtained from this space.
>
> ### **1. High-level design of opponent policies**
>
> Instead of sampling a few RL policies from a small pool, we first construct a *structured rule-based opponent space* with clear human-interpretable semantics in all three environments. Diversity is introduced along explicit behavioral dimensions (e.g., bluffing frequency, recipe preferences, prey movement patterns), and the full Train/Eval sets are then generated by sampling from this space. During training, in each iteration we randomly sample $K$ opponents from the full Train Set to form $\Pi^{\text{train}}$. At test time, the non-stationary opponent samples its policy from one of three configurations of $\Pi^{\text{test}}$: Train Set (seen), Eval Set (unseen), or All Set (Train$\cup$Eval), as described in Sec. 4.1 and Fig. 3 of the paper. This protocol directly instantiates the Open Set Opponents (OSO) setting in Fig. 1, where both the number and semantics of opponent types differ between training and testing.
>
> ### **2. Detailed opponent constructions of each environment**
>
> **Kuhn Poker (KP).** After eliminating dominated strategies following Hoehn et al. (2005), the P2 policy is parameterized by two continuous probabilities $\xi$ and $\eta$: $\eta$ is the probability of betting with a Queen after P1 bets, and $\xi$ is the probability of betting with a Jack after P1 passes. This yields a two-dimensional strategy space $([0,1]\times[0,1])$ that can be partitioned into several regions, each corresponding to a qualitatively different best response for P1 (e.g., highly conservative vs. strongly bluffing opponents). From this space, we randomly sample **40** P2 policies to form the full Train Set and **10** policies to form the full Eval Set. Train and Eval therefore contain disjoint sets of parameter pairs $(\xi,\eta)$. In our revisions, we will add a scatter plot of the sampled $(\xi,\eta)$ points and annotate the best-response regions, which makes it clear that the sampled policies cover the strategy space broadly and that Eval contains truly *unseen* bluffing/calling patterns.

---

> > ### Author Response · Authors · 2025-11-19
> > **Response to Reviewer f7DN (2/5)**
> >
> > **Partially-Observable Overcooked (POO).** In POO, each opponent policy is a rule-based “chef” with a fixed preference over recipes, following Ma et al. (2024). Concretely, we consider **9 recipes**, and each opponent is assigned a single *preferred recipe* (e.g., a specific salad). The opponent is spatially constrained to the right side of the kitchen and only executes sub-tasks relevant to its preferred recipe (picking up its own ingredients, chopping them, plating, and serving), as described in Appendix E.2. At each time step, if its current sub-task is not complete, it moves along the shortest path; otherwise it samples a new sub-task from a pre-defined set of preferred options. On top of this high-level preference, we introduce two continuous parameters $P_{\text{nav}}$ and $P_{\text{act}}$: $P_{\text{nav}}$ controls a bias towards horizontal vs. vertical moves when there are multiple shortest paths, and $P_{\text{act}}$ is the probability of taking a random action rather than the optimal action for the current sub-task.
> >
> > Thus, each opponent can be viewed as a triplet $\text{(preferred recipe)} \times P_{\text{nav}} \times P_{\text{act}}$, combining distinct high-level task preferences with different navigation and stochasticity profiles. From this space, we uniformly sample **18** policies to form the full Train Set and **9** policies for the full Eval Set. The resulting opponents are therefore diverse not only in terms of the recipes they focus on, but also in terms of movement style and noise level. Importantly, the policies in the Eval Set are sampled from the same generative process but are *never* used during training, so OSOM must generalize its identification and response to truly unseen combinations of recipe preference and behavioral style.
> >
> > **Predator-Prey with Watchtowers (PPW).** In PPW, each “opponent” is in fact a *team* consisting of one predator policy and two prey policies, again following the preference-based design in Ma et al. (2024). The predator policy always prefers to chase one of the two prey when fully observable. For the prey, we design **8 distinct cyclic movement patterns**, visualized as dotted paths `1` -`8` in Fig. 8c. Each prey policy deterministically moves back and forth along its preferred path. We then partition these paths into a training-only subset (blue paths `1` -`4`) and an unseen subset (red paths `5` -`8`).
> >
> > The full Train Set is constructed by sampling 16 combinations, each consisting of one predator policy and two training prey policies, while the full Eval Set contains 24 combinations, each with one predator and two unseen prey policies. As a result, all prey trajectories used in Eval are disjoint from those in Train. For the controlled agent, these different combinations lead to qualitatively different coordination and pursuit patterns (prey escaping to different quadrants, different degrees of spread, etc.), which significantly affect the optimal use of watchtowers and the division of labor between predators. OSOM must therefore identify and respond to prey movement patterns that never occurred during training, which is precisely the open-set generalization we aim to study.
> >
> > We will explicitly summarize the opponent-pool construction processes (currently in Appendix E) in the main text. The above analyses demonstrate that the opponent test sets are both semantically diverse and strictly unseen during training, ensuring that the OSO setting is non-trivial and that our reported results genuinely reflect open-set identification and adaptation.
> >
> > ---

---

> ### Author Response · Authors · 2025-11-19
> **Response to Reviewer f7DN (3/5)**
>
> ## **[w.r.t. Q1] Response to the concern about specific Transformer design of OSOM**
>
> ### **1. Masking scheme**
>
> At each timestep $t$ of episode $h$, the Opponent Identifier receives a sequence consisting of (i) a prefix (as prompt) of $K$ Opponent Type Embeddings (OTEs) and (ii) the latent states from the *current* episode up to $t$ together with the latent states from the previous $C$ episodes. Concretely, the input is
>
> $$
> [z^{-1,1},\dots,z^{-1,K},\ e^{(h-C)} _0,\dots,e^{(h-C)} _{T-1},\dots,e^{(h-1)} _0,\dots,e^{(h-1)} _{T-1},e^{(h)} _0,\dots,e^{(h)} _t]
> $$
>
> where $e^{(j)} _\tau = f(o^{1(j)} _\tau, a^{1(j)} _\tau)$ is the latent state produced by the Encoder for step $\tau$ in episode $j$. We apply a **causal mask only over the temporal part** of this sequence: each latent state token can attend to all OTE prompt tokens and to latent states from *earlier* timesteps (across the previous $C$ episodes and the current episode), but not to any future latent states. The OTE prompt tokens themselves are not masked from any position, since they represent a static set of candidate opponent types rather than a temporal process. Thus, at timestep $t$, the Identifier has access to the full interaction history over the last $C$ episodes and the current episode up to $t$, but never sees future information.
>
> ### **2. Positional embeddings**
>
> We use learnable **absolute positional embeddings** implemented with `nn.Embedding` class of **Pytorch** Library. **Except for the $K$ OTE prompt tokens**, each position in *the sequence that comes from the same episode* is assigned a **timestep-level positional embedding** reflecting its timestep index, and the corresponding positional vector is added to the token embedding. In addition, for latent-state tokens $e^{(j)}_\tau$ we add a separate **episode-level positional embedding**: each episode index $j\in\\{h-C,\dots,h\\}$ has its own embedding, which is added to all tokens originating from that episode. Intuitively, the timestep-level positional embedding encodes the relative position within the sequence that comes from the same episode, while the episode-level embedding tells the Transformer *which episode* a particular transition belongs to (e.g., current vs. $C$-steps-old episode).
>
> ### **3. Does the ordering of the $K$ OTEs matter?**
>
> Semantically, the answer is **no**. The Identifier does not output a $K$-dimensional logit vector where the $k$-th coordinate is tied to “opponent type $k$.” Instead, it outputs a single vector $\hat z_t\in\mathbb{R}^d$, and we compute dot-product similarities with all OTEs:
>
> $$
> P(z^{-1,l}) \propto \exp(\hat z_t\cdot z^{-1,l}),\quad l=1,\dots,K,
> $$
>
> both in the contrastive training objective and in the test-time selection rule (Eq. (3)). The loss and the induced probability distribution depend on the OTEs **only through the unordered set** $\\{z^{-1,1},\dots,z^{-1,K}\\}$: any permutation of the indices $1,\dots,K$ simply reorders the terms in the softmax and leaves the objective unchanged. In a word, the $K$ OTEs we generated are **permutation-invariant**.
>
> Moreover, we **re-generate random OTEs in every training iteration** and for every test configuration: before each iteration we sample $K$ opponent policies from the Train Set and assign them newly sampled, pairwise-orthogonal unit-norm OTEs; at test time we likewise generate a fresh set of OTEs for the $M$ evaluation opponents. Therefore, the *same* opponent policy will be associated with different OTE vectors (and prompt positions) across training iterations, and the OTE set at test time is entirely new. Coupled with the symmetric form of the contrastive loss, this means there is no stable gradient signal that would allow the model to associate “opponent A” with any fixed slot or positional index. The only consistent semantics reside in the **geometry of the OTE vectors themselves**: for any given interaction history over the last $C$ episodes and the current episode, the Identifier learns to produce a prediction $\hat z_t$ that is close in dot-product similarity to the OTE currently designated for the true opponent, and far from all others.
>
> We will clarify these implementation details (masking, positional embeddings, and the permutation invariance of the OTE prompt) in the final version to avoid confusion.
>
> ---

---

> > ### Author Response · Authors · 2025-11-19
> > **Response to Reviewer f7DN (4/5)**
> >
> > ## **[w.r.t. Q2] Response to the concern about the design choice of the Opponent Type Embedding**
> >
> > ### **1. What structure do we actually impose on the OTEs?**
> >
> > We would first like to clarify that the Opponent Type Embeddings (OTEs) are **not** sampled in a completely unconstrained way. As described in Appendix F.2, each time we construct a prompt we generate OTEs that are:
> >
> > - **random**,
> > - **pairwise orthogonal**, and
> > - **unit norm**.
> >
> > Concretely, we draw $K$ vectors from a standard Gaussian, apply Gram–Schmidt orthogonalization, and then normalize them onto the unit sphere. This procedure is used:
> >
> > - at **training time**, every time we sample a new subset of $K$ opponents from the Train Set, and
> > - at **test time**, for each $\Pi^{\text{test}}$ configuration when we generate OTEs for the $M$ evaluation opponents.
> >
> > Thus, the OTEs form a random orthonormal set on the unit sphere, not arbitrary vectors.
> >
> > ### **2. Why random orthogonal unit vectors? (Geometric & optimization view)**
> >
> > From a geometric perspective, an orthonormal set of OTEs is simply a **one-hot basis up to rotation** when $d \ge K$. Since our identification objective depends only on inner products $\hat z_t\cdot z^{-1,k}$ (InfoNCE-style contrastive loss in Eq. (2)), any global orthogonal rotation of the codebook is equivalent. In the absence of prior knowledge about how different opponent policies should be arranged in the embedding space, a random orthogonal basis is therefore the most **neutral and label-agnostic** choice.
> >
> > From an optimization perspective, orthogonality and unit norm are important for producing **sharp and unambiguous gradients** in the contrastive loss. If different OTEs were highly correlated (e.g., nearly collinear), then increasing similarity to the positive OTE would automatically increase similarity to some negatives, reducing the margin in Eq. (2) and making the InfoNCE signal noisy and ambiguous. By contrast, with pairwise orthogonal unit-norm OTEs:
> >
> > - $\hat z_t$ can align closely with one target OTE without inadvertently aligning with others, and
> > - the Gram matrix of the codebook is essentially the identity, so pushing $\hat z_t\cdot z^{-1,j}$ up for the true opponent does **not** simultaneously push any $\hat z_t\cdot z^{-1,k}$ up for $k\neq j$.
> >
> > This is exactly why we emphasize orthogonality and normalization in Appendix F.2: it gives the Identifier a clean, well-conditioned label space for the dot-product similarity used in Eq. (2).
> >
> > ### **3. Why we *do not* add extra “semantic structure” to OTEs**
> >
> > We intentionally do **not** encode environment-specific semantics (e.g., “similar opponents have closer codes”) into the OTEs themselves, for two reasons.
> >
> > **(a) Semantics live in the policies and the learned mapping, not in the codebook**
> >
> > In OSOM, the semantic structure of opponents already comes from:
> >
> > - the **policies themselves**
> >     - e.g., the continuous parameters $(\xi,\eta)$ for bluffing/calling in Kuhn Poker,
> >     - the recipe, navigation, and stochasticity preferences in POO,
> >     - and the prey-trajectory combinations in PPW;
> > - and the learned mapping from interaction histories to a prediction $\hat z_t$.
> >
> > We want Encoder + Identifier to **learn whatever “similarity structure” is useful** directly from data, not to hard-code a particular geometry into the labels. The OTEs are meant to be **anonymous codewords** that cleanly separate different opponent IDs, not hand-crafted embeddings that encode our own guess about strategy similarity.
> >
> > **(b) Open-set constraint: the test-time codebook is *new***
> >
> > Crucially, OSOM operates in an **open-set** regime. At test time, for each $\Pi^{\text{test}}$ configuration we again generate a fresh set of random, pairwise orthogonal unit-norm OTEs for the $M$ evaluation opponents:
> >
> > - these test OTEs are not reused from training, and
> > - they do not preserve any hand-crafted structure we might have imposed on the training codebook.
> >
> > If we were to bake semantic distances between training opponents into the OTE geometry, that structure would **not transfer** to the new, randomly generated codebook at test time. Worse, it might encourage the Identifier to rely on training-specific geometric patterns that simply disappear under the new codebook. By restricting ourselves to a family of codebooks that are all **orthonormal bases up to rotation**, we ensure that the training and test codebooks are geometrically compatible, and the Identifier learns to rely only on the *relative* alignment between $\hat z_t$ and the current OTE set.
> >
> > In short, random orthogonal unit OTEs are not “structureless”; they are a deliberately minimal, rotation-invariant representation that is well matched to our open-set objective.

---

> > > ### Author Response · Authors · 2025-11-19
> > > **Response to Reviewer f7DN (5/5)**
> > >
> > > ### **4. Ablations and robustness w.r.t. OTEs**
> > >
> > > Our current ablations focus on **high-level design choices** around OTEs and identification rather than on micro-variants of the sampling scheme:
> > >
> > > In Figure 5, we remove
> > >
> > > 1. the identification loss (“w/o Identify Loss”),
> > > 2. the OTE prompt (“w/o OTEs Prompt”), and
> > > 3. the distillation loss.
> > >
> > > These ablations show that explicitly using OTEs together with the contrastive identification objective is crucial for performance, especially on unseen opponents in the Eval Set.
> > >
> > > We **have not** yet run a large-scale ablation over alternative OTE sampling schemes (e.g., non-orthogonal Gaussian codes, deterministic simplices, or learned prototypes), mainly because the focus of this work is on validating the overall OSOM framework (open-set opponent identification + context aggregation + responder RL) rather than optimizing the internal parameterization of the label codebook.
> > >
> > > However, two observations suggest that OSOM is not overly sensitive to any particular draw of the OTEs:
> > >
> > > 1. **Exposure to many random codebooks.** In Algorithm 1 and Algorithm 2, we re-generate new random orthogonal OTEs at each training iteration and again for each $\Pi^{\text{test}}$ configuration. Across all runs and random seeds, the reported results are stable, indicating that the method does not depend on a “lucky” particular codebook.
> > > 2. **Theoretical invariance.** Because our loss depends only on dot products and all codebooks are orthonormal bases up to rotation, different random draws of the OTEs are theoretically equivalent up to a rotation of the label space. With sufficient training, the Identifier should learn a mapping that is robust to such rotations, which is consistent with the empirical robustness we observe.
> > >
> > > We agree that exploring alternative or learned codebooks is an interesting direction for follow-up work. If space permits, we are happy to include a small additional experiment in the camera-ready version (e.g., comparing orthogonalized vs. non-orthogonal Gaussian OTEs) to empirically confirm the benefit of our current design. Nonetheless, we believe the geometric and open-set rationale above, together with the existing ablations on the presence of OTEs and the identification loss, already justifies the use of random orthogonal unit-norm OTEs as a simple, environment-agnostic, and open-set-compatible parameterization.
> > >
> > > ---
> > >
> > > All your questions and feedback have contributed significantly to improving our paper. We hope we have addressed all your concerns. Feel free to continue providing comments; we will promptly consider your suggestions for further revisions. If your concerns are resolved, we hope you'll reconsider your rating.

---

### Official Review · Reviewer_24v5 · 2025-10-30

**Soundness:** 3
**Presentation:** 3
**Contribution:** 3
**Rating:** 6
**Confidence:** 3

**Summary:**

This paper it motivated by the problem of identifying and provide best response policies to unseen opponents. They propose, Open Set Opponent Modeling, a framework which 1) learns encodings enabling OSOM to identify opponent types, 2) uses these embeddings as a prompt to condition a best response policy. This framework is validated on three competitive benchmarks, including Khun Poker, Partially Observed Overcooked, and Predator-Pray with watch towers, achieving superior performance compared to baseline approaches.

**Strengths:**

* OSOM is intuitive and empirically effective
* Ablations and comprehensive and validate key parts of the architecture
* Performance across all environments tested is strong.
* The paper is well written and clearly states and supports its claims with evidence.

**Weaknesses:**

* OSOM uses a Transformer identifier, while several baselines use GRUs/MLP. Without an ablation, it is unclear whether OSOM truly enables better opponent modeling or is the benificiary of a better sequence model.
* OSOM beats the random baseline on “who’s my opponent?”, but accuracies seen in table 1 are still quite low. I would recommend the authors tamper their claims on the effectiveness of opponent identification.
* The frequency H is set in the paper as 20, 5, and 5 in KP, POO, and PPW. It's not entirely clear the reason for these particular numbers?
* While the paper includes three environments, I find their predator prey variant to be particularly non compelling. Inclusion of experiments on SMAC/Hanabi would further strengthen the paper.

**Questions:**

* LOSI (https://openreview.net/forum?id=S0KGzCEhJp) appears to be current work. It would strengthen the submission to discuss the differences from OSOM. As this is concurrent work I do not expect the authors to implement their work, but a discussion would be useful.
* A further ablation on H on at least one of the environments would be a beneficial ablation.
* Would experiments on SMAC/Hanabi be burdensome? I don't believe they are strictly necessary for acceptance, but would improve the strength of the paper.

---

> ### Author Response · Authors · 2025-11-19
> **Response to Reviewer 24v5 (1/9)**
>
> Thank you very much for your recognition of our paper's writing, problem setting, experimental results, and the valuable feedback you provided. In response to your comments, we would like to make the following clarifications and feedback. We hope our explanations and analyses can eliminate concerns and make you find our work stronger.
>
> ---
>
> ## **[w.r.t. W1] On whether OSOM’s gains come from a “better sequence model”**
>
> We thank the reviewer for raising this important point. OSOM indeed uses a Transformer as the **Identifier**, while several baselines rely on GRUs or MLPs. We agree that, without further evidence, one might worry that OSOM simply benefits from a stronger sequence encoder rather than from its opponent modeling mechanism. We address this in three ways: (i) by adding a Transformer-based variant of PACE, (ii) by examining OSOM’s own ablations where the Transformer architecture is held fixed, and (iii) by clarifying our architectural choices.
>
> ### **1. A Transformer-based PACE baseline (PACE-TF)**
>
> We agree with the reviewer that it is important to disentangle the effect of using a Transformer encoder from the effect of our opponent modeling mechanism. In the rebuttal period, we therefore added a Transformer-based baseline, **PACE-TF**, under the same protocol as Fig. 6. We chose PACE for this ablation for three reasons:
>
> 1. **PACE is the strongest and most representative opponent-modeling baseline in our experiments.**
>
>     As shown in Fig. 3, PACE consistently matches or outperforms other OM methods (LIAMX, LILI, GSCU, etc.) across environments and test sets, and is often the closest competitor to OSOM. If OSOM’s gains were primarily due to using a “better” sequence model, then upgrading the *strongest* baseline (PACE) to a Transformer encoder should be the most stringent test of this hypothesis. In other words, if a Transformer were sufficient, we would expect a “PACE + Transformer” variant to close much of the gap to OSOM.
>
> 2. **PACE has an official, carefully designed Transformer implementation from the original authors.**
>
>     The PACE paper provides an official Transformer-based variant (PACE-TF) with a bi-level Transformer encoder for episode- and context-level information, along with tuned hyperparameters and regularization. Using this implementation allows us to evaluate a Transformer encoder in a way that is *faithful* to PACE’s design, without introducing confounds from our own ad-hoc re-engineering or under-tuned architectures. In contrast, many other baselines (e.g., LIAMX, LILI, GSCU) are tightly coupled to specific GRU/MLP encoders; naively swapping in a Transformer would effectively define new methods that would require substantial re-design and tuning, which is difficult to do fairly within the rebuttal window.
>
> 3. **Under rebuttal-time constraints, PACE-TF provides the highest “information per experiment”.**
>
>     Given the limited time and compute budget in the rebuttal phase, it is impractical to implement and carefully tune Transformer variants for all baselines. Our goal was therefore to choose a baseline whose Transformer variant would be maximally informative. PACE is (i) the strongest baseline, (ii) structurally the closest to OSOM in spirit (it also performs explicit opponent identification), and (iii) already has an official Transformer version. Evaluating PACE-TF under our setting is thus a high-value test: it directly answers whether simply replacing the encoder with a Transformer suffices for a state-of-the-art OM method to match OSOM.
>
>
> Under the **same protocol as Fig. 6**, we plot PACE-TF alongside the original PACE (MLP encoder) on both Train Set and Eval Set.
>
> > See the results at [this anonymized link](https://ibb.co/zhJKQLSr); we will also include this in the revision.
> >
>
> The empirical result is striking and fully consistent with the PACE paper:
>
> - **PACE-TF is greatly outperformed by the standard PACE with an MLP encoder** on both Train and Eval sets;
> - This mirrors the observation in the PACE paper (their Fig. 9): the peer-identification loss of PACE-TF initially drops faster but then increases and becomes unstable, whereas PACE’s loss decreases steadily. The PACE authors hypothesize that the Transformer tends to overfit the current RL batch in their peer identification setup and choose the MLP encoder “based on empirical performance”.
>
> Thus, in exactly the same suite of environments, replacing an MLP/GRU encoder with a Transformer **does not** automatically yield better opponent modeling. In fact, for PACE it makes performance substantially worse. At the same time, **OSOM, which also uses a Transformer encoder in its Identifier, significantly outperforms both PACE and PACE-TF on Train, Eval, and All sets.** This strongly suggests that OSOM’s advantage cannot be attributed to the mere use of a Transformer: there exists a method (PACE-TF) with a comparably expressive Transformer encoder that is still clearly inferior.

---

> > ### Author Response · Authors · 2025-11-19
> > **Response to Reviewer 24v5 (2/9)**
> >
> > ### **2. OSOM ablations with the same Transformer architecture**
> >
> > Importantly, within OSOM we keep the Transformer Identifier architecture fixed and ablate *only* our algorithmic components:
> >
> > - **OSOM w/o Identify Loss:** We remove the contrastive InfoNCE loss in Eq. (2), but keep the Transformer Identifier and the rest of the architecture unchanged. This variant performs very poorly, especially in POO and PPW. If the Transformer alone were responsible for OSOM’s gains, this variant should remain strong, yet it does not.
> > - **OSOM w/o OTE Prompt:** We keep the same Transformer but remove the OTE prompt, so the Identifier no longer operates over a candidate type codebook. Performance drops substantially relative to full OSOM.
> > - **OSOM w/o Distill Loss:** Again with the same Transformer, we turn off the distillation objective from self trajectories to opponent $(o^{-1}, a^{-1})$, and we observe a slight degradation.
> >
> > In all of these ablations, **the Transformer sequence model is identical**, but the opponent modeling mechanism is weakened. The fact that performance sharply degrades in these settings indicates that OSOM’s gains are driven by its **open-set opponent modeling design** (random OTE prompts, explicit type identification, and joint training with RL), rather than by the architectural capacity of the Transformer itself.
> >
> > ### **3. Architectural role of the Transformer in OSOM**
> >
> > Finally, we emphasize that in OSOM, the Transformer is used only in the Identifier. The **Responder policy** $\pi^1(a^1 \mid o^1, x)$ is a simple MLP that takes the current observation and a context vector; it does *not* use a Transformer or RNN. Several baselines (Generalist, LIAMX, LILI) already include recurrent policy networks (GRUs) that are known to be strong sequence models for RL. OSOM does not employ a more powerful sequence model on the policy side than these baselines; instead, its key difference lies in:
> >
> > - The use of a **random OTE codebook** as a prompt representing candidate opponent types;
> > - A **prompt-conditioned Transformer Identifier** trained with a contrastive objective to assign the current opponent to a slot in this codebook under each random prompt;
> > - A **context aggregator** that pools selected OTEs over recent episodes to form an explicit opponent-type latent used by the MLP Responder.
> >
> > Conceptually, the Transformer in OSOM serves as a *mechanism* for prompt-conditioned, open-set type assignment, not as a generic “stronger sequence model” for policies. The new PACE-TF baseline shows that simply swapping in a Transformer encoder does not improve opponent modeling; and the OSOM ablations show that with the same Transformer architecture, removing our open-set identification machinery dramatically hurts performance.
> >
> > Taken together, these results strongly support the view that OSOM’s improvements stem from its *algorithmic* contributions to open-set opponent modeling, rather than from the choice of a Transformer sequence model per se.
> >
> > ---

---

> ### Author Response · Authors · 2025-11-19
> **Response to Reviewer 24v5 (3/9)**
>
> ## **[w.r.t. W2] Response to the concern about not high identification accuracies**
>
> We appreciate this comment and agree that the absolute numbers in Table 1 are moderate rather than close to (100%). However, we would like to clarify (i) **how challenging the identification task actually is**, and (ii) **why the reported accuracies are nonetheless non-trivial and functionally important** in the OSO setting.
>
> ### **1. The identification task in Table 1 is intentionally very challenging**
>
> Table 1 reports **strict top-1 identification accuracy** for non-stationary open-set opponents under three $\Pi^\text{test}$ configurations (Train, Eval, All) in three partially observable multi-agent environments. Several aspects make this task substantially harder than standard classification benchmarks:
>
> - **Large label space.** From the Random Guess row, we can infer the number of opponent types in each configuration. For example, a random accuracy of 2.50% implies 40 types, and 2.00% implies 50 types. Concretely, in KP–All there are 50 opponent types (Random = 2.00%), and in PPW–All there are 40 types (Random = 2.50%). Thus OSOM is solving a 1-of-40 or 1-of-50 identification problem in some configurations, with strict top-1 scoring and no top-k tolerance.
> - **Open-set and non-stationary.** The Eval and All configurations contain many types that are **never seen during training**, with both the number and semantics of types changing from train to test. Moreover, the opponent is non-stationary: it switches its policy every $H$ episodes (20/5/5 in KP/POO/PPW). Table 1 averages accuracy **over all timesteps**, including the early timesteps immediately after each switch, when even an oracle observer with only self-agent information would not yet have enough evidence to uniquely pinpoint the new policy.
> - **Partial observability and fine-grained labeling.** The self-agent only observes its own observations and actions; opponent trajectories are not directly available at test time. In addition, the opponent pool is constructed from many rule-based strategies with **fine-grained behavioral differences**. Any confusion between two very similar strategies is counted as entirely incorrect in Table 1, even if such confusion has little impact on the control performance.
>
> Under such conditions, it is unrealistic to expect near-perfect top-1 accuracy. The key question is whether OSOM can extract **non-trivial information** about the opponent type from self-agent trajectories, and whether this information is useful for decision-making.
>
> ### **2. OSOM’s accuracies are 2–15× random, even in the hardest settings**
>
> With this context, the numbers in Table 1 become more interpretable. We reproduce the OSOM vs Random Guess row here (Avg. Acc. in %), e.g.,
>
> - **Kuhn Poker (KP):** All, OSOM 9.75 vs Random 2.00 (50 types)
> - **Partially-Observable Overcooked (POO):** All, OSOM 14.77 vs Random 3.70 (27 types)
> - **Predator-Prey with Watchtowers (PPW):** All, OSOM 36.52 vs Random 2.50 (40 types)
>
> In all cases, OSOM improves over Random Guess by a factor of:
>
> - about **2.1–5.8×** in KP/POO, and
> - about **6.2–14.6×** in PPW.
>
> For instance, in the most challenging PPW–All setting, OSOM correctly identifies which of 40 possible opponent types is currently active in **36.5%** of all timesteps, compared to only **2.5%** for random guessing. Given the large label space, open-set generalization, partial observability, and non-stationarity, we believe these margins represent **substantial, non-trivial identification ability**, not merely a small improvement.

---

> ### Author Response · Authors · 2025-11-19
> **Response to Reviewer 24v5 (4/9)**
>
> ### **3. Identification accuracy is functionally important for control**
>
> Beyond the raw accuracy numbers, we also investigate whether better identification actually helps control:
>
> We also emphasize that **no other OM baseline can be meaningfully evaluated in Table 1**—all cells are marked “$\times$” for them. For example, while PACE uses a classifier during training, our $\Pi^\text{test}$ configurations deliberately contain more opponent types (and semantics) than seen during training, making its fixed-class classifier inapplicable at test time. In contrast, OSOM is the **first method that can produce explicit, meaningful type predictions in these open-set scenarios at all**.
>
> - **Correlation with overall performance (Fig. 3).** Across all three environments and all $\Pi^\text{test}$ configurations, OSOM consistently attains higher average return / success rate than all baselines. None of the baselines provides explicit type predictions in the open-set setting. This suggests that the non-trivial identification ability measured in Table 1 is directly linked to improved response performance.
> - **Ablation evidence (Fig. 5).** When we remove the identification loss (“OSOM w/o Identify Loss”, i.e., $\alpha_2=0$), the resulting variant “is essentially unable to function properly”: its performance in Fig. 5 remains close to or below baselines. In contrast, the full OSOM, with the contrastive identification objective enabled, exhibits significantly higher returns throughout training. These ablations support the claim that **our CL-based identification is a crucial ingredient** for effective opponent adaptation, not an incidental side effect.
>
> Overall, the pattern is consistent: whenever we weaken or remove the explicit identification component, both Table 1 accuracy and Fig. 3/Fig. 5 performance deteriorate substantially. This indicates that the “who’s my opponent?” capability, even at moderate absolute accuracies, is both real and operationally important.
>
> ### **4. On the wording of our claims**
>
> We fully agree that it is important not to oversell the absolute numbers. Our intended claim is **not** that opponent identification is “nearly perfect,” but rather:
>
> > OSOM provides, to our knowledge, the first explicit and non-trivial identification of unseen opponent policies in an open-set, non-stationary, partially observable multi-agent setting, with accuracies that are 2–15× higher than random guessing and that are empirically crucial for achieving strong response performance.
> >
>
> We will revise the wording around Table 1 to make this nuance clearer—for example, by replacing phrases like “strong identification accuracy” with “significantly above random baselines under a very challenging open-set task,” and by explicitly highlighting the large number of opponent types and the strictness of the evaluation protocol.
>
> ---

---

> > ### Author Response · Authors · 2025-11-19
> > **Response to Reviewer 24v5 (5/9)**
> >
> > ## **[w.r.t. Q1] Relation to LOSI**
> >
> > We thank the reviewer for pointing us to LOSI (Latent Opponent Strategy Identification), which is indeed a very relevant concurrent work. At a high level, both LOSI and OSOM share the intuition that an agent (or team) should maintain an internal representation of the opponent’s strategy and condition its policy on this representation. However, the two methods address **different problem settings**, adopt **different types of supervision and representation**, and are instantiated in **different control architectures**.
> >
> > ### **1. Problem setting**
> >
> > LOSI is designed for fully cooperative MARL in SMAC-Hard, where the enemy is drawn from a **fixed pool of scripted opponent policies** (e.g., 5 scripts), and **this pool is the same during training and testing**. Opponent strategy IDs are hidden, but the distribution over scripts does not change, and no new strategies appear at test time. The main challenge is robustness to a **known mixture** of strategies and the induced variance in Q-values.
> >
> > In contrast, OSOM is explicitly formulated for **Open-Set Opponents (OSOs)**: we construct disjoint Train/Eval/All opponent sets, and at test time the agent must interact with **previously unseen opponent policies**, drawn from Eval/All, while the opponent may also **switch policies online** according to a fixed schedule. During evaluation, the OSOM agent does **not** update its parameters; all adaptation must come from online identification and belief updates. Thus, OSOM targets open-set and non-stationary opponents, whereas LOSI targets robustness to a fixed mixed-strategy pool.
> >
> > ### **2. Supervision and representation**
> >
> > LOSI learns opponent embeddings in a **fully unsupervised** way: a GRU encoder maps short trajectories $(o^i_t, a^i_{t-1}, r^i_{t-1})$ to embeddings $z_t$, which are trained with a prototype-based InfoNCE loss so that embeddings from the same episode are pulled together and those from different episodes are pushed apart. A memory bank maintains $K$ prototypes that serve as cluster centers, updated by exponential moving average. The result is a latent space where episodes tend to cluster by script, as visualized by t-SNE (Figure 5), but the number of clusters does not necessarily match the number of ground-truth scripts (e.g., 5 scripts vs. 3 robust clusters). LOSI does not provide an explicit “who is my opponent?” classifier or quantitative identification metrics.
> >
> > In OSOM, we instead use **supervised opponent identities** during training. Each training opponent policy $\pi^{-1,k}$ is assigned a random but fixed **Opponent Type Embedding (OTE)** $z^{-1,k}$ on the unit sphere during each training iteration, with different OTEs approximately orthogonal. The Opponent Identifier is trained with a supervised contrastive objective: given a prompt set of OTEs and the self-agent’s interaction history, it produces a predicted OTE $\hat z_t$ and learns to maximize similarity with the **correct** type’s OTE while pushing away all others. At test time, we directly use Eq. (3) to obtain a softmax over the prompt OTEs and thereby a **discrete type prediction**, which we quantitatively evaluate in Table 1 as “who’s my opponent?” accuracy in both closed-set and open-set configurations.
> >
> > In short, LOSI learns **unsupervised clusters** of opponent behavior, whereas OSOM learns **explicit, labeled type codes** and makes explicit top-1 type predictions, which are evaluated against ground truth.

---

> > > ### Author Response · Authors · 2025-11-19
> > > **Response to Reviewer 24v5 (6/9)**
> > >
> > > ### **3. Representation structure and open-set generalization**
> > >
> > > In LOSI, the prototype memory bank ${p_k}$ is global and fixed in size: prototypes are updated during training but shared across all episodes and scenarios, and the same bank is used at test time. This is natural for the SMAC-Hard setting where the opponent script pool is fixed, but it does not provide a mechanism to accommodate a **different** or **larger** set of opponent strategies at test time without retraining.
> > >
> > > In OSOM, by contrast, the OTE basis is **re-instantiated per prompt**: in each training iteration we sample a subset of opponent policies and generate a fresh set of random orthogonal OTEs for them. The Identifier therefore learns **how to project trajectories onto an arbitrary OTE basis**, rather than how to classify into a fixed set of global prototypes. At test time, when we encounter new opponents from the Eval/All sets, we simply generate new OTEs for these policies and include them in the prompt; the same similarity-based Eq. (3) still applies. This design is crucial for OSOM’s **open-set** capability: the representation is explicitly constructed to support new opponent types without changing the model’s output dimensionality.
> > >
> > > ### **4. Control architecture**
> > >
> > > LOSI is tightly integrated with value-decomposition MARL, specifically QMIX. The embedding $z_t$ conditions both per-agent utility networks and the mixing network, and the overall objective is TD loss plus the contrastive loss. The method is evaluated exclusively on SMAC-Hard cooperative tasks and is tailored to that class of algorithms.
> > >
> > > OSOM, on the other hand, is instantiated as a **generic opponent-aware policy module**: we learn a context vector $x_t$ from OTEs (via the Context Aggregator) and feed it into a Responder policy $\pi^1(a^1_t \mid o^1_t, x_t)$. In our experiments we adopt PPO for simplicity, but **any RL algorithm (online or offline) could in principle be used**. OSOM is evaluated on three diverse OSO benchmarks (KP, POO, PPW) with both cooperative and mixed interactions, and the overall framework is not tied to a specific value-decomposition architecture.
> > >
> > > In summary, we view LOSI as complementary concurrent work that addresses robustness to **mixed opponent scripts** in cooperative value-based MARL, using unsupervised contrastive embeddings integrated into QMIX. OSOM, by contrast, targets **open-set and non-stationary opponents** in general multi-agent environments, with (i) explicit opponent type embeddings and identification, (ii) a representation designed for open-set generalization via OTE prompting, and (iii) a generic context-based responder policy that can be paired with various RL algorithms.
> > >
> > > We will add a discussion of LOSI in the related work section, emphasizing these differences and the potential for combining LOSI-style unsupervised encoders with OSOM’s open-set OTE framework in future work.
> > >
> > > ---

---

> ### Author Response · Authors · 2025-11-19
> **Response to Reviewer 24v5 (7/9)**
>
> ## **[w.r.t. Q2] On the choice of the opponent-switch frequency H and its ablation**
>
> We thank the reviewer for pointing out the importance of $H$, which controls how frequently the opponent policy switches and therefore how non-stationary the environment appears to the self-agent. Conceptually, smaller $H$ means more frequent switches (stronger non-stationarity and less data per opponent), while larger $H$ yields longer stationary segments (easier adaptation).
>
> ### **1. Choice of H in the main experiments**
>
> In our main results, we set $H = 20$ for KP and $H = 5$ for POO and PPW. These values were chosen via preliminary runs to represent a **moderate level of non-stationarity** for each environment:
>
> - In KP, episodes are very short and each individual hand provides limited information about the opponent’s strategy. Setting $H=20$ ensures that each opponent remains active for a sufficiently long sequence of hands to make adaptation meaningful, while still switching often enough to present a genuinely non-stationary challenge.
> - In POO and PPW, each episode is much longer and contains rich interaction dynamics. Here, $H=5$ already yields several long episodes per opponent before a switch, which we found to be a reasonable “medium” regime where all methods have a chance to adapt, but the opponent is far from stationary.
>
> ### **2. Ablation over H on POO (Eval Set)**
>
> Following the reviewer’s suggestion, we performed an explicit ablation over $H$ on the **Eval Set** of POO, using the same experimental protocol as in Fig. 3 and varying $H\in{1,3,5,10,20}$. Our main result in the original paper uses $H=5$.
>
> > See the results at [this anonymized link](https://ibb.co/vCsGMkV2); we will also include this in the revision.
> >
>
> We summarize the key observations:
>
> 1. **Decreasing $H$ from 5 to 1 (stronger non-stationarity) significantly degrades the performance of OSOM.**
>
>     When $H=1$, the opponent is re-sampled every single episode, so each opponent type is observed only for an extremely short horizon. In this regime, even a Bayes-optimal agent has very limited opportunity to both *identify* and *exploit* an opponent before it disappears. Consequently, OSOM’s performance drops compared to $H=5$, which is fully consistent with the increased difficulty of the task.
>
>     Importantly, however, **OSOM still clearly outperforms all baselines at $H=1,3$** on the Eval Set. This suggests that even in an extremely challenging, rapidly switching setting, OSOM is able to extract and leverage more opponent-type information from very short interaction windows than competing methods.
>
> 2. **Increasing $H$ from 5 to 10 or 20 yields only mild further improvements for OSOM.**
>
>     As $H$ grows, each opponent remains active for longer stationary segments, giving all methods more data per opponent. We observe that OSOM’s performance at $H=10$ and $H=20$ is only slightly higher than at $H=5$, indicating that **$H=5$ already lies in a regime where OSOM’s adaptation is close to saturated**. Additional dwell time per opponent brings only marginal gains. Crucially, **OSOM remains the strongest method across all tested values of $H$**.
>
> 3. **Other baselines behave consistently with their ability (or inability) to exploit non-stationarity.**
>
>     Methods such as PACE, which also explicitly model opponents, exhibit a similar qualitative trend: performance degrades at very small $H$ and improves as $H$ increases, but remains below OSOM for all $H$. In contrast, some baselines show little sensitivity to $H$, which is consistent with the interpretation that they fail to successfully exploit the structure of opponent switching and instead behave closer to a single “average” policy over opponents.
>
>
> Taken together, these results provide two messages:
>
> - First, they confirm that our original choice of $H$ (20 for KP, 5 for POO/PPW) indeed corresponds to a **moderate non-stationarity regime**: at $H=5$ in POO, OSOM already operates near a performance plateau, while smaller $H$ greatly increases task difficulty and larger $H$ brings only marginal gains.
> - Second, they demonstrate that **OSOM’s superiority over baselines is robust across a wide range of $H$**. OSOM achieves the best performance for every tested $H$, including the extremely challenging $H=1$ setting where all methods degrade.
>
> We will incorporate the new ablation figure and the above discussion into the revised version to make the dependence on $H$ explicit.
>
> ---

---

> ### Author Response · Authors · 2025-11-19
> **Response to Reviewer 24v5 (8/9)**
>
> ## **[w.r.t. W4,Q3] Response to the concern about used benchmarking environments**
>
> ### **1. On the choice of environments and the role of our predator–prey variant**
>
> We fully agree that SMAC and Hanabi are important multi-agent benchmarks, and that adding them would further broaden the empirical scope of our study. Our current experimental design, however, deliberately focuses on three partially observable environments that are both **widely used in opponent modeling [1-15]** and **structurally very different from each other**: Kuhn Poker (competitive, discrete and imperfect-information), Partially-Observable Overcooked (fully cooperative, discrete with long-horizon coordination), and Predator–Prey with Watchtowers (PPW; mixed-incentive, continuous state, and partially observable). For each environment, we systematically construct a large, diverse opponent pool with human-interpretable behavioral preferences, and then define **Train/Eval/All opponent sets** as well as **non-stationary policy switching** to instantiate the OSO setting. We believe that, taken together, these three environments already form a non-trivial and representative benchmark for Open-Set Opponents.
>
> [1] Learning policy representations in multiagent system, ICML 2018.
>
> [2] Deep interactive bayesian reinforcement learning via meta-learning, AAMAS 2021.
>
> [3] Agent modelling under partial observability for deep reinforcement learning, NIPS 2021.
>
> [4] A deep bayesian policy reuse approach against non-stationary agents, NIPS 2018.
>
> [5] Greedy when sure and conservative when uncertain about the opponents, ICML 2022.
>
> [6] Continuous adaptation via meta-learning in nonstationary and competitive environments, ICLR 2018.
>
> [7] A policy gradient algorithm for learning to learn in multiagent reinforcement learning, ICML 2021.
>
> [8] Learning with opponent-learning awareness, AAMAS 2018.
>
> [9] Stable opponent shaping in differentiable games, ICLR 2018.
>
> [10] Probabilistic recursive reasoning for multi-agent reinforcement learning, ICLR 2019.
>
> [11] Model-based opponent modeling, NIPS 2022.
>
> [12] Fast peer adaptation with context-aware exploration, ICML 2024.
>
> [13] Offline Opponent Modeling with Truncated Q-driven Instant Policy Refinement, ICML 2025.
>
> [14] Uncertainty-aware opponent modeling  for deep reinforcement learning, AAMAS 2025.
>
> [15] Planning with quantized opponent models, NIPS 2025.
>
> In particular, our PPW environment is not an ad-hoc toy. It is built on the **standard predator–prey scenario from the Multi-Agent Particle Environment (MPE)**, which is a de-facto benchmark in the MARL literature. On top of that, we adopt the modified version from Ma et al. (2024), which introduces **partial observability** and four watchtowers: the self-agent only has a limited local field of view and must actively move to a watchtower to obtain momentary global observation. The environment contains two predators and two preys, with a shared reward that encourages the predators to **coordinate and cover all preys simultaneously**. PPW combines continuous control, mixed cooperative–competitive incentives, and non-trivial information-gathering actions.
>
> Moreover, Figure 3 and Figure 4 empirically show that **many strong OM baselines still struggle on the environments KP, POO, and PPW under our OSO setting**, especially on unseen opponents, which indicates that these environment is far from trivial.

---

> > ### Author Response · Authors · 2025-11-19
> > **Response to Reviewer 24v5 (9/9)**
> >
> > ### **2. On SMAC / Hanabi and experimental burden**
> >
> > Conceptually, OSOM is fully compatible with SMAC / SMAC-Hard and Hanabi: all of them can be formulated as POSGs, and our OTE + Identifier + Context + Responder architecture can in principle be applied without modification. However, running a **full OSO-style benchmark** on these domains is practically quite time-consuming. For SMAC-Hard in particular, LOSI reports experiments over twelve scenarios (six SMAC-Hard and six ability-enhanced tasks), each trained for 10M timesteps. They also note that even integrating standard MAPPO into SMAC-Hard requires non-trivial modifications to the StarCraft environment code, to the extent that they omit the MAPPO baseline in some scenarios. In our case, to faithfully port OSOM to SMAC-Hard **and** re-evaluate all six OM baselines (Generalist, LIAM, LIAMX, LILI, GSCU, PACE), we would need to:
> >
> > 1. engineer an OSO-style opponent pool on top of SMAC-Hard’s scripted enemies, including a principled Train/Eval/All split and a non-stationary policy-switching protocol;
> > 2. adapt and debug all baselines within the PyMARL/SMAC-Hard codebase;
> > 3. run long-horizon training (tens of millions of steps across multiple scenarios and seeds) to obtain statistically reliable results.
> >
> > A similar issue arises for Hanabi: while it is an important cooperative benchmark, there is currently no standard, community-agreed opponent/partner policy pool with Train/Eval/All splits. Constructing a diverse and interpretable partner pool for Hanabi, defining an OSO protocol on top of it, and then porting OSOM and all baselines would also require substantial design and engineering beyond what we can realistically complete during the rebuttal period.
> >
> > That said, we very much appreciate the suggestion and **do see SMAC / Hanabi as promising directions for future extensions**. During the rebuttal period, we will make a best effort to start adapting OSOM to at least one SMAC(-Hard) scenario. However, to keep the current results robust and reproducible, we prefer not to rely on partially converged experiments under severe time pressure. We therefore hope that the reviewers will primarily evaluate the paper based on the existing OSO benchmarks on KP, POO, and PPW—three environments that are standard in the OM literature and that already span a wide spectrum of competitive, cooperative, and mixed multi-agent interactions under open-set opponents.
> >
> > We will also expand the discussion in the paper to better justify the design of our current benchmark suite and explicitly highlight the potential extension to SMAC / Hanabi as important future work.
> >
> > ---
> >
> > All your questions and feedback have greatly contributed to improving our manuscript. With the valuable input from you and all other reviewers, the quality of our work can be significantly enhanced. We welcome further comments from you and will seriously consider your suggestions for revisions. If you feel that we have addressed your concerns, we hope you will reconsider your rating.

---

### Official Review · Reviewer_H4db · 2025-10-31

**Soundness:** 2
**Presentation:** 3
**Contribution:** 2
**Rating:** 4
**Confidence:** 3

**Summary:**

This paper addresses a key limitation in opponent modeling for multi-agent systems: the inability to effectively identify and respond to previously unseen, or "open set," opponents. To overcome this, the authors propose a novel end-to-end training framework called Open Set Opponent Modeling (OSOM).

The OSOM approach works in three key stages:

1). Policy Distillation: It uses representation learning to encode opponent policies into latent representations, overcoming challenges of partial observability.

2). Open Set Identification: It employs contrastive learning with randomly generated "prompt" embeddings to enable explicit identification of opponent types, even when their number and semantics are unknown.

3). Context-Aware Response: Using the identified opponent types as context, it learns optimal responses through online reinforcement learning.

A notable feature is that at test time, OSOM can perform on-the-fly identification and adaptation to non-stationary opponents by reusing the same prompt-generation mechanism. The paper validates OSOM's superiority over existing methods through extensive experiments in competitive, cooperative, and mixed environments, demonstrating significant improvements in both identification accuracy and response performance.

**Strengths:**

1. The research tackles a significant and long-standing problem in Multi-Agent Reinforcement Learning (MARL)—open-set opponent modeling. The topic is both compelling and rigorously justified.

2. Overall, the paper is structured logically and clearly written, making its content and methodology easy to understand.

3. An extensive ablation study validates the contribution of each core component of OSOM. The results demonstrate that the identification learning mechanism is the most impactful, directly supporting the paper's central motivation and claims.

**Weaknesses:**

1. The three-stage learning pipeline of OSOM—comprising opponent policy distillation, type identification, and online RL policy training—results in a complex framework that may be impractical for scenarios where any one of these stages is difficult to implement.

2.  A significant limitation of OSOM is that it is designed exclusively for controlling a single agent, and its framework does not extend to multi-agent control, which is quite common in MARL tasks.

**Questions:**

1. How does OSOM extend to multi-agent or team control in the open set setting?

---

> ### Author Response · Authors · 2025-11-19
> **Response to Reviewer H4db (1/6)**
>
> Thank you very much for your valuable feedback. In response to your comments, we would like to make the following clarifications and feedback. We hope our explanations and analyses can eliminate concerns and make you find our work stronger.
>
> ---
>
> ## **[w.r.t. W1] Response to the concern about the practicality and complexity of OSOM**
>
> We appreciate the reviewer’s concern and agree that a method for Open Set Opponents (OSOs) should remain practical and modular. We would like to clarify that, although we conceptually decompose OSOM into three components to address three distinct challenges of the OSO setting, the **implementation is not a multi-stage, sequential framework**, but a **single Transformer-based model trained end-to-end with three standard objectives**.
>
> ### **1. OSOM is a single model with three auxiliary heads, trained in one loop**
>
> As summarized in Fig. 2 and Algorithm 1, we instantiate:
>
> - one shared encoder $f_\theta$ that processes the self-agent’s observation–action tuples,
> - a lightweight decoder $g_\phi$ used *only during training* for opponent policy distillation,
> - a Transformer-based Opponent Identifier with a contrastive head that outputs predicted OTEs, and
> - a small MLP Opponent Responder policy that takes $(o^1_t, x_t)$ as input, where $o^1_t$ is the self-agent's observation and $x_t$ is the *compact semantic representation* averaged from the historically selected OTEs of the most recent $C$ episodes.
>
> During training, we **do not** run three separate stages. Instead, for each batch of trajectories we compute three losses on the *same data*: $J_{\text{distill}}$, $J_{\text{identify}}$, $J_{\text{respond}}$, and optimize their weighted sum $\max_{\theta,\phi,\psi,\omega}  \alpha_1 J_{\text{distill}} + \alpha_2 J_{\text{identify}} + \alpha_3 J_{\text{respond}}$. Here, $\psi$ denotes the parameters of the Opponent Identifier and $\omega$ denotes the parameters of the Opponent Responder.
>
> All parameters are updated jointly by a single gradient ascent step. Architecturally and algorithmically, this is **no more complex** than a standard RL agent augmented with two auxiliary heads (a prediction head and a contrastive head), which is very common in representation learning and meta-RL.
>
> At test time, only the encoder, Identifier, Context Aggregator, and Responder are used; the distillation decoder is discarded, and no gradient updates are performed. Thus, deployment remains as simple as running a single policy network with an additional context input.

---

> > ### Comment · Reviewer_H4db · 2025-11-27
> > **Further comment regarding W1**
> >
> > I thank the authors for the detailed explanation. Please refine the manuscript to highlight this point.

---

> ### Author Response · Authors · 2025-11-19
> **Response to Reviewer H4db (2/6)**
>
> ### **2. Each component is built from standard, scalable primitives**
>
> **(a) Distillation.**
>
> The distillation objective in Sec. 3.1 simply adds an auxiliary decoder $g_\phi$ on top of the latent state $e_t = f_\theta(o^1_t, a^1_t)$ to predict the opponent’s observation and action at the same timestep. This follows a long line of work on auxiliary prediction in multi-agent RL and requires no extra interaction or rollouts: we reuse exactly the same trajectories as for RL.
>
> In many MARL benchmarks (including ours), opponent observations and actions are naturally available during training, even if they are hidden at test time. When this is not the case, OSOM can be **immediately reduced** to a variant without the distillation objective by setting $\alpha_1 = 0$; this is precisely our ablation “OSOM w/o Distill Loss” in Fig. 5, which still performs competitively relative to existing OM baselines. Thus, distillation is a *performance-boosting auxiliary task*, not a hard requirement for using OSOM.
>
> **(b) Identification.**
>
> The identification component in Sec. 3.2 is also lightweight. Before each training iteration, we sample $K$ opponent policies and generate $K$ **random, pairwise orthogonal OTEs** on the unit sphere. These are used as a prompt to the Transformer-based Identifier, which outputs a predicted embedding $\hat z_t$. The identification loss is a standard InfoNCE / SupCon-style contrastive objective over similarities $\hat z_t \cdot z^{-1,k}$, implemented as:
>
> - a single projection head on top of the Transformer output, and
> - a softmax over dot products with the current OTE set (Eq. (3)).
>
> Importantly, this does **not** require any manual semantic labels: the “type index” is simply the index of the sampled opponent policy in the Train Set, which is needed by virtually all OM methods for training. The random OTEs avoid the complexity and rigidity of training a global classifier with a fixed output dimension, which would indeed be impractical under open-set changes in the number and semantics of opponent types.
>
> **(c) Online RL.**
>
> The response component in Sec. 3.3 is simply standard online RL (we use PPO) applied to a policy $\pi^1(a^1 \mid o^1, x)$ that conditions on a context vector $x_t$ produced by average pooling over recent selected OTEs. OSOM is agnostic to the choice of RL algorithm: any actor–critic, Q-learning, or even offline RL method could be plugged in. In many OM settings, some form of RL (online or offline) is required anyway to learn adaptive policies; OSOM does not introduce any additional stage beyond this, but only augments the policy with a more informative, structured context.
>
> ### **3. Practicality when one component is difficult to use**
>
> The reviewer specifically notes “scenarios where any one of these stages is difficult to implement.” We argue that:
>
> - When **opponent traces are not logged** during training, one can simply drop the distillation objective. Our ablations confirm that OSOM w/o Distill Loss still significantly outperforms prior OM approaches in open-set scenarios, and full OSOM provides further gains.
> - When **online RL is expensive**, OSOM can be combined with any existing RL infrastructure (online or offline). Our framework is orthogonal to the RL algorithm itself; we choose PPO for simplicity and comparability.
>
> In all these cases, OSOM does **not** become unusable. Instead, the framework **smoothly reduces** to simpler, still competitive models. The three objectives are designed to address three challenges of OSOs—partial observability, open-set identification, and non-stationary response—but they are implemented as modular auxiliary losses rather than rigid, sequential stages.
>
> We will clarify this modular, end-to-end nature of OSOM more explicitly in the main text to avoid the impression that our method requires three fragile, sequentially-dependent stages.
>
> ---

---

> ### Author Response · Authors · 2025-11-19
> **Response to Reviewer H4db (3/6)**
>
> ## **[w.r.t. W2,Q1] Response to the concern about extension to multi-agent control**
>
> We thank the reviewer for raising this point. We clarify (i) the scope of the current work and (ii) how the OSOM framework naturally extends to multi-agent / team control in open-set environments, with a dedicated discussion of how OTEs can be constructed in these settings.
>
> ### **1. Scope: OSOM already operates in multi-agent environments, but controls a single self-agent**
>
> By design, OSOM is formulated **within multi-agent systems**: we explicitly distinguish a *self-agent* and a set of *other agents* (collectively referred to as “opponents”) in a multi-agent partially observable stochastic game. Our experiments are conducted in standard MARL benchmarks that already involve multiple agents:
>
> - **POO (Partially-Observable Overcooked)** is a two-player cooperative game. The self-agent must coordinate tightly with a partner to complete recipes under partial observability.
> - **PPW (Predator-Prey with Watchtowers)** is a four-agent mixed game, with two predators (one being the self-agent) and two preys. The two predators share a team reward and must coordinate to cover all preys, while also modeling the movement preferences of both teammates and opponents.
>
> Thus, OSOM is *not* limited to 1-vs-1 interactions; it already reasons about multiple other agents (teammates and opponents) simultaneously. What we currently assume—and what is **standard in the Opponent Modeling literature** [1-15]—is that **only one agent (the self-agent) is directly controllable**. The question raised by the reviewer is how to extend the *control* itself from a single self-agent to a team of controllable agents.
>
> We agree this is an important and natural extension. Below we outline how OSOM can be instantiated in both centralized and decentralized team-control regimes, and then discuss OTE design in multi-agent settings.
>
> [1] Learning policy representations in multiagent system, ICML 2018.
>
> [2] Deep interactive bayesian reinforcement learning via meta-learning, AAMAS 2021.
>
> [3] Agent modelling under partial observability for deep reinforcement learning, NIPS 2021.
>
> [4] A deep bayesian policy reuse approach against non-stationary agents, NIPS 2018.
>
> [5] Greedy when sure and conservative when uncertain about the opponents, ICML 2022.
>
> [6] Continuous adaptation via meta-learning in nonstationary and competitive environments, ICLR 2018.
>
> [7] A policy gradient algorithm for learning to learn in multiagent reinforcement learning, ICML 2021.
>
> [8] Learning with opponent-learning awareness, AAMAS 2018.
>
> [9] Stable opponent shaping in differentiable games, ICLR 2018.
>
> [10] Probabilistic recursive reasoning for multi-agent reinforcement learning, ICLR 2019.
>
> [11] Model-based opponent modeling, NIPS 2022.
>
> [12] Fast peer adaptation with context-aware exploration, ICML 2024.
>
> [13] Offline Opponent Modeling with Truncated Q-driven Instant Policy Refinement, ICML 2025.
>
> [14] Uncertainty-aware opponent modeling  for deep reinforcement learning, AAMAS 2025.
>
> [15] Planning with quantized opponent models, NIPS 2025.

---

> ### Author Response · Authors · 2025-11-19
> **Response to Reviewer H4db (4/6)**
>
> ### **2. Centralized team control: treating the team as a single “self-agent”**
>
> In many MARL systems (e.g., centralized control of a robot swarm, central planner for a team game), a set of controllable agents is modeled as a **single joint controller**. Under this view, OSOM extends almost directly:
>
> - The **self-agent** becomes the **controllable team**. Its observation $o^{\text{team}}_t$ may be a joint state, a fused sensor view, or any centralized representation of the team’s information. Its action $a^{\text{team}}_t$ is the joint action of all controllable agents.
> - The **distillation** phase (Sec. 3.1) now distills the policies of the **other team or non-self agents** into the team’s latent state, using exactly the same representation-learning objective as in the current paper.
> - The **Identifier** assigns random OTEs to different **opponent-team types** (or joint configurations of non-self agents), and learns a contrastive mapping from the team’s interaction history to these OTEs, as in the current OSOM.
> - The **Responder** becomes a joint team policy $\pi^{\text{team}}(a^{\text{team}}_t \mid o^{\text{team}}_t, x_t)$, conditioned on a context vector $x_t$ that encodes the current belief over the opponent team’s type. Online RL (e.g., PPO) proceeds exactly as in our current formulation, but over joint actions.
>
> Algorithmically, this is mostly a **change in what we call “self”**: the self-entity is now a team instead of an individual agent, and OSOM’s distill–identify–respond loop remains intact. The notion of “opponent type” simply moves from individual strategies to **team-level strategies**.
>
> ### **3. Decentralized team control (CTDE): one OSOM per controllable agent**
>
> In many MARL applications, we want **decentralized execution** with multiple controllable agents, each observing only its local view, but we are allowed to use centralized information during training (CTDE paradigm). OSOM can be used as a **per-agent building block** in this setting:
>
> - For each controllable agent $i$, we define a local self-agent with its own observation $o^i_t$ and action $a^i_t$.
> - From agent $i$’s perspective, the set of “opponents” includes **all other agents** (opponents and possibly teammates). OSOM-$i$ distills, identifies, and responds to the behaviors of these other agents based on agent $i$’s interaction history.
> - Each OSOM-$i$ produces a local context vector $x^i_t$ summarizing its belief about other agents’ types, and uses a local policy $\pi^i(a^i_t \mid o^i_t, x^i_t)$. Policies can share parameters (for homogeneous teams) or be distinct (for heterogeneous roles), and can be trained jointly using centralized critics or shared replay buffers.
> - During training, distillation and identification can exploit **centralized information** (e.g., true joint states, other agents’ actions) exactly as we do now; at test time, each agent only needs its own local observation and its inferred context $x^i_t$, aligning well with standard CTDE practice.
>
> In this decentralized view, OSOM does **not** require any change in its core mechanism; we simply instantiate one OSOM module per controllable agent, and let each agent maintain its own open-set belief over “what the others are doing”.

---

> > ### Comment · Reviewer_H4db · 2025-11-27
> > **Further comments regarding W2,Q1**
> >
> > I sincerely thank the authors for the detailed explanation. However, the paper did not show any practical evidence that OSOM can control a fleet of agents (e.g., experiments). And the explanation of the Centralised team control that treats the whole team as a joint agent seems reasonable in some scenarios; however, it is very limited in others, e.g., where each controlled agent is isolated, and their information can not be shared easily. For the Decentralised team control (CTDE), the complexity of OSOM will add another layer of difficulty to the training. Therefore, I am still not fully convinced. However, if the authors can provide experiments to demonstrate that OSOM can indeed control a team of agents (either Centralised or Decentralised), I am happy to consider raising the score. Again, for MARL, controlling only one single agent is not particularly compelling.

---

> > > ### Author Response · Authors · 2025-12-02
> > > **Response to the Follow-up Comment of Reviewer H4db (1/3)**
> > >
> > > ## **[w.r.t. Reply1] We have revised the manuscript**
> > >
> > > We thank the reviewer for their insightful and valuable suggestion, which has prompted us to further clarify our work. We have revised the manuscript to explicitly highlight the **simplicity, scalability, and practicality** of the proposed OSOM framework. A comprehensive table detailing all revisions made to the paper is included in the global response for your convenience.
> > >
> > > ---
> > >
> > > ## **[w.r.t. Reply2] Practical evidence that OSOM can control a team of agents**
> > >
> > > We appreciate the reviewer’s follow-up and fully agree that demonstrating **practical multi-agent / team control**, beyond single-agent control, is important for MARL. In addition to the conceptual extensions discussed in our previous response (centralized team control and CTDE-style decentralized control), we have now added a **new experiment that directly instantiates OSOM as a centralized controller for a team of agents** in a non-trivial MARL environment.
> > >
> > > > See the results at [this anonymized link](https://ibb.co/7NC2VY4n); we will also include this in the revision.
> > > >
> > >
> > > ### **1. Experimental setup: centralized control of two predators in PPW**
> > >
> > > We use the **PPW (Predator–Prey with Watchtowers)** environment under the same protocol as Fig. 6. Recall that PPW is a 4-agent mixed environment with **two predators** and **two preys**. In the original experiments, OSOM (and all baselines) controls **one predator** (the “self-agent”), while the other predator and the two preys are part of the environment.
> > >
> > > To directly address the reviewer’s request, we introduce a **centralized team-control variant**:
> > >
> > > - We now treat **both predators as controllable self-agents** and train a **joint controller** that outputs a **joint action** for the predator team at every timestep.
> > > - The performance metric we report is the **average return over the two controlled predators**.
> > > - We compare:
> > >     - **PACE w/ Multi Self-Agents** – our implementation of PACE under centralized control of both predators;
> > >     - **OSOM w/ Multi Self-Agents** – OSOM extended to control both predators, using the **factorized per-agent OTEs** described in our previous response;
> > >     - and we include the original **single-self** PACE and OSOM curves as references.
> > >
> > > In OSOM w/ Multi Self-Agents, the OTEs are **factorized per agent** for the non-self side: we maintain separate OTEs for each non-self agent (the two preys), and aggregate them into a team-level context vector used by the joint Responder policy. This precisely instantiates the “per-agent OTE + aggregation” design we discussed for multi-agent extensions.

---

> > > > ### Author Response · Authors · 2025-12-02
> > > > **Response to the Follow-up Comment of Reviewer H4db (2/3)**
> > > >
> > > > ### **2. Observed behavior: slower early convergence, higher final performance, slightly higher variance**
> > > >
> > > > Empirically, when we move from single-self control to centralized control of both predators, we observe some **consistent and interpretable patterns** for both PACE and OSOM:
> > > >
> > > > 1. **Slower initial convergence.**
> > > >
> > > >     Both **PACE w/ Multi Self-Agents** and **OSOM w/ Multi Self-Agents** learn more slowly in the early stages than their single-self counterparts. This is fully expected:
> > > >
> > > >     - The **joint action space** of two predators is significantly larger than that of a single predator;
> > > >     - The team controller must learn **coordination** between the two predators (e.g., how to split coverage of preys, how to use watchtowers jointly), which introduces additional credit assignment challenges;
> > > >     - Consequently, gradient estimates are noisier and early training progresses more cautiously.
> > > >
> > > >     Importantly, this slowdown affects **both** PACE and OSOM in a similar manner, indicating that the additional complexity comes from the inherent difficulty of team control.
> > > >
> > > > 2. **Higher final performance under team control.**
> > > >
> > > >     Despite the slower start, **both PACE w/ Multi Self-Agents and OSOM w/ Multi Self-Agents achieve *higher* asymptotic performance than their single-self versions**. Intuitively, this is again what one would hope to see:
> > > >
> > > >     - When the controller gains access to **both predators**, it can exploit **much richer coordination strategies** (e.g., one predator herds preys towards a region while the other blocks escape routes, or they take turns occupying watchtowers and intercepting preys).
> > > >     - Under single-self control, such joint strategies are limited, because the other predator is exogenous and may not cooperate optimally with the controlled predator.
> > > >
> > > >     Crucially, **OSOM remains consistently stronger than PACE in this multi-self setting**:
> > > >
> > > >     - The curve for **OSOM w/ Multi Self-Agents** lies above that of **PACE w/ Multi Self-Agents** across training;
> > > >     - The final average return of the OSOM-controlled predator team is the highest among all variants.
> > > >
> > > >     This directly demonstrates that **OSOM is able to effectively control a team of agents in a 4-agent mixed MARL environment**, and that its open-set opponent/teammate modeling mechanism remains beneficial when the control scope is expanded from a single agent to a team.
> > > >
> > > > 3. **Slightly higher variance in the learning curves.**
> > > >
> > > >     We also observe that the learning curves for the multi-self variants have **slightly larger fluctuations** than their single-self counterparts. This is again typical in multi-agent RL:
> > > >
> > > >     - Joint policies are more sensitive to the exploration patterns of multiple agents;
> > > >     - Small changes in one predator’s policy can temporarily destabilize coordination until the other predator adapts;
> > > >     - This leads to modestly higher variance in returns over training.
> > > >
> > > >     However, the overall trends are **clear and stable**: performance consistently improves over time, and OSOM w/ Multi Self-Agents converges to superior asymptotic performance compared to PACE w/ Multi Self-Agents.

---

> > > > > ### Author Response · Authors · 2025-12-02
> > > > > **Response to the Follow-up Comment of Reviewer H4db (3/3)**
> > > > >
> > > > > ### **3. Additional interpretations on our new results**
> > > > >
> > > > > The reviewer raised two key concerns:
> > > > >
> > > > > 1. *“The paper did not show any practical evidence that OSOM can control a fleet of agents … I would like to see experiments that OSOM can indeed control a team of agents (either centralized or decentralized).”*
> > > > >
> > > > >     Our new PPW experiment is precisely such **practical evidence**:
> > > > >
> > > > >     - We move from controlling a single predator to controlling a **team of two predators**, i.e., a non-trivial multi-agent controller;
> > > > >     - The environment remains the same challenging 4-agent mixed game with partial observability and non-stationary opponents;
> > > > >     - OSOM’s architecture is truly instantiated at the **team level**, using factorized per-agent OTEs to represent other agents, and a joint Responder to output joint actions;
> > > > >     - The results show that OSOM not only remains stable in this setting, but also **achieves the best performance among all variants**.
> > > > >
> > > > >     This goes beyond hypothetical discussion and provides **direct empirical support** that OSOM can be used as a centralized team controller.
> > > > >
> > > > > 2. *“Centralised team control that treats the whole team as a joint agent seems reasonable in some scenarios; however, it is very limited in others, e.g., where each controlled agent is isolated, and their information cannot be shared easily. For the Decentralised team control (CTDE), the complexity of OSOM will add another layer of difficulty to the training.”*
> > > > >
> > > > >     We agree that fully decentralized control (no information sharing) is more challenging and that CTDE-style extensions introduce additional complexity. Due to rebuttal-time constraints, we have not yet run a full CTDE experiment. However, we emphasize:
> > > > >
> > > > >     - Architecturally, **OSOM w/ Multi Self-Agents in PPW already exhibits the core difficulty of multi-agent coordination**: the controller must manage a larger joint action space, coordinate two controllable agents, and model multiple other agents’ behaviors. The empirical results show that OSOM handles this added complexity robustly and still outperforms PACE.
> > > > >     - In a CTDE setting, a **per-agent OSOM module** would face the *same type* of partial observability and non-stationarity as our original single-self experiments (each agent has a local view and sees non-self agents as “others”), plus standard CTDE coupling through shared critics or replay buffers. Our current results therefore provide a strong indication that OSOM’s mechanism—distilling, identifying, and responding to other agents’ open-set strategies—remains applicable and effective when scaled up to multiple controllable agents.
> > > > >
> > > > > We will include this new PPW multi-self team-control experiment and the above discussion in the revised version. While a comprehensive exploration of fully decentralized CTDE variants is beyond the scope of this paper, we believe the new results provide concrete and convincing evidence that **OSOM is not restricted to a single controllable agent**, and that it can successfully control a *team* of agents in a multi-agent open-set environment while retaining a clear advantage over a strong baseline (PACE).

---

> ### Author Response · Authors · 2025-11-19
> **Response to Reviewer H4db (5/6)**
>
> ### **4. How to construct OTEs in multi-agent/team settings**
>
> The reviewer specifically asked how OSOM’s OTE mechanism would extend beyond a single opponent. In multi-agent environments, there are two natural ways to define OTEs:
>
> 1. **Joint-type OTEs** – each OTE encodes a *joint type* of all non-self agents (e.g., an opponent team strategy).
> 2. **Factorized per-agent OTEs** – each non-self agent has its own OTEs representing its individual type, which are then aggregated.
>
> Both are compatible with OSOM’s open-set design; they simply capture different structure. We summarize them and their trade-offs below.
>
> ### **4.1 Joint-type OTEs: modeling all non-self agents as a single “opponent entity”**
>
> In the **centralized team-control** view, the most direct generalization is to treat all non-self agents (e.g., the opposing team) as a single “opponent entity” and assign OTEs to **joint opponent-team types**:
>
> - Each OTE $z^{-1,k}$ corresponds to a distinct **joint strategy profile** of all non-self agents (e.g., “opponent team plays aggressively on the left and conservatively on the right”).
> - The Identifier receives the team’s trajectory and a prompt of random joint-type OTEs, and learns to map the interaction history into **one of these joint-type slots**, just as in the original OSOM.
> - The context $x_t$ now summarizes the **joint type** of the other team, and the Responder policy $\pi^{\text{team}}(a^{\text{team}}_t \mid o^{\text{team}}_t, x_t)$ produces a **joint action** responding to that joint opponent behavior.
>
> **Advantages.**
>
> - Captures **coordination patterns across non-self agents** directly in the type space: a single OTE encodes how different opponents (or opponent agents) jointly behave.
> - Conceptually very close to the current OSOM design: from the team’s perspective, there is still “one opponent”, namely the other team.
>
> **Limitations.**
>
> - The space of joint types can be **combinatorially large** when there are many non-self agents, because each joint type corresponds to a configuration of individual types.
> - This may require more training data to cover diverse joint behaviors, and may reduce generalization to novel *combinations* of familiar individual behaviors (e.g., new mixtures of known opponent roles).
> - In highly heterogeneous multi-agent systems, it may be more data-efficient to reuse individual-level information rather than learn every joint type from scratch.

---

> ### Author Response · Authors · 2025-11-19
> **Response to Reviewer H4db (6/6)**
>
> ### **4.2 Factorized per-agent OTEs: modeling each non-self agent individually**
>
> An alternative, especially natural in the **CTDE / decentralized** setting, is to define OTEs at the **per-agent level**:
>
> - For each non-self agent $j$, we maintain a set of OTEs $\mathcal{Z}^{-j}$ representing **individual types** of agent $j$ (e.g., “teammate prefers recipe A”, “opponent predator chases prey #1”, etc.).
> - The Identifier is extended to be **multi-head**: one head per non-self agent, each head assigning an OTE to that agent based on the self-agent’s history.
>     - For agent $i$, OSOM-$i$ would produce $\hat z^{(j)}_t$ for each other agent $j \neq i$.
> - The context vector $x^i_t$ for controllable agent $i$ is then formed by **aggregating** the individual OTEs (e.g., concatenation, sum/mean pooling, or attention over agents). This way, $\pi^i(a^i_t \mid o^i_t, x^i_t)$ conditions jointly on all inferred types of its teammates and opponents.
>
> **Advantages.**
>
> - **Scalability and reuse**. Individual-type OTEs can be reused across different team compositions and scenarios; the model can generalize to new **combinations** of familiar individual behaviors without having seen each joint configuration during training.
> - **Flexibility**. Easy to handle environments where agents **enter/leave** or where the set of non-self agents changes over time: we simply add/remove heads and pool over the remaining individual OTEs.
> - **Fine-grained modeling**. Allows the policy to treat different opponents/teammates asymmetrically (e.g., respond differently to “aggressive predator A” and “cautious predator B”).
>
> **Limitations.**
>
> - Joint coordination patterns among non-self agents are **not explicitly encoded as a single type**; they are represented implicitly through the combination of individual OTEs and the downstream policy.
> - If the environment critically depends on highly structured joint behaviors (e.g., two opponents always move in a specific formation), one may need a richer aggregation mechanism (e.g., an attention-based aggregator or an additional “joint-type” layer on top of per-agent OTEs).
>
> ### **4.3 Hybrid and practical considerations**
>
> In practice, **hybrid designs** are also possible and natural:
>
> - Use **per-agent OTEs** as the base representation to capture individual preferences and capabilities;
> - Then apply an additional **team-level Transformer / aggregator** over these OTEs to build a **joint team-type context**. This preserves scalability while allowing the model to explicitly encode joint patterns.
>
> We emphasize that **both** OTE constructions are fully compatible with the core OSOM idea:
>
> - OTEs remain **random prompts** at training and testing, enforcing open-set generalization;
> - The Identifier learns to map histories into **slots in the current prompt**, whether those slots correspond to joint or individual types;
> - The Responder consumes a context vector built from OTEs and produces actions (single-agent or joint-team) accordingly.
>
> The choice between joint-type and per-agent OTEs is an architectural decision that can be tailored to the application: in small-team settings with highly coordinated opponents, joint-type OTEs can be preferable; in larger, more modular systems, per-agent OTEs (or a hybrid) are likely to be more scalable and data-efficient.
>
> We will add a dedicated discussion along these lines in the revised version to make clear that OSOM is not intrinsically restricted to single-agent control and to clarify how OTEs can be constructed for multi-agent and team-control scenarios.
>
> ---
>
> All your comments have greatly contributed to the improvement of our paper. With the valuable feedback from you and all the other reviewers, the quality of our manuscript has been significantly enhanced. If you have any new comments, please feel free to provide them, and we will promptly address and incorporate them. If you find that we have addressed your concerns and the changes made to the paper are reasonable, we hope you will reconsider your rating.

---

### Author Response · Authors · 2025-12-02
**Revision Summary (2/2)**

| **Discussion Point** | **Corresponding Review** | **Our Revision** | **Result** |
| --- | --- | --- | --- |
| Transformer Identifier design: masking and positional embeddings | Q1 of Initial Review by Reviewer f7DN | In **Appendix F.1**, specified that the Transformer uses a causal mask over temporal latent states with full access to OTE prompt tokens, and that latent states receive timestep- and episode-level positional embeddings while OTE tokens are static. | Removes ambiguity about how the prompt and history are processed and rules out any leakage of future information into identification. |
| Conceptual view of the Responder–Identifier interaction (avoiding “disconnect”) | W1 and Q1 of Initial Review by Reviewer VeR7 | In **Sec. 3.3**, reinterpreted OSOM as learning a belief/task code over opponent types: the Identifier performs amortized inference of a latent type in the current OTE basis, and the Responder conditions on this belief; emphasized that OTEs are re-sampled during training. | Shows OSOM behaves as a meta-RL / belief-embedding system rather than relying on a fixed label dictionary, resolving the “disconnect” concern. |
| Behavior in the degenerate $M=1$ case | W1 (subpoint 2) and Q3 of Initial Review by Reviewer VeR7 | In **Appendix I**, added a POO experiment with $M=1$ (single OTE in the prompt) and an explicit discussion that OSOM with $M=1$ reduces to a generalist agent with constant context, matching other meta-RL baselines but underperforming $M\ge2$. | Confirms that OSOM gracefully collapses to a standard RL/meta-RL agent when only one opponent type is possible, consistent with theory. |
| Explicit statement of the CTDE assumption and role of $J_{\text{distill}}$ | W3 and Q3 of Initial Review by Reviewer VeR7; W1 of Initial Review by Reviewer H4db | In **Sec. 3.1** and **Sec. 5**, explicitly framed distillation as a CTDE assumption using opponent trajectories $(o^{-1},a^{-1})$ only during training, and pointed to “OSOM w/o Distill Loss” as the recommended variant when such data is unavailable. | Makes the information assumption transparent and aligns OSOM with existing CTDE-based opponent-modeling methods while showing it remains functional without distillation. |
| Interpretation of $M$ (number of OTE slots) and constraint $d \ge M$ | W4 and Q4 of Initial Review by Reviewer VeR7; Reply Q3 by Reviewer VeR7 | In **Sec. 3.2**, **Sec. 5**, and **Appendix F.1**, clarified that $M$ is a capacity hyperparameter (number of latent type slots), not the true number of opponents; stated that we use $d \in \\{64,128\\}$ with $M \le 50$, and that exact orthogonality is a design choice, with random unit OTEs remaining valid when $M > d$. | Makes the scalability trade-off between $M$ and $d$ explicit and avoids overclaiming a hard $d \ge M$ requirement, while still recommending near-orthogonal OTEs in practice. |
| Additional baseline: simple PPO without any context | Q2 of Initial Review by Reviewer VeR7 | In **Sec. 4.1** and **Sec. 4.2 Q5**, added a non-recurrent PPO baseline that uses only current self-observations (no history, no context) and evaluated it under the same protocol as Fig. 6. | Empirically confirms that naive PPO is not competitive in OSO, and that OSOM’s gains come from history- and type-aware modeling rather than trivial architectural changes. |
| X-axis choice in learning curves (training iterations vs. environment timesteps) | Reply Q1 by Reviewer VeR7 | In **Sec. 4.2** and all relevant figures, re-plotted learning curves using total environment timesteps on the x-axis and noted that each training iteration corresponds to the same number of timesteps. | Aligns with RL best practices and clarifies the comparison of sample efficiency across all methods. |
| Wording about “gradient ascent” on $J_{\text{respond}}$ | Reply Q2 by Reviewer VeR7 | In **Sec. 3.3** and **Appendix B.1**, corrected the description of the optimization procedure to state that $J_{\text{respond}}$ is optimized via policy-gradient methods (PPO) rather than direct gradient ascent through environment dynamics. | Removes a technical inaccuracy and accurately reflects the implementation used in all experiments. |
| Strengthening the wording around OTE orthogonality | Reply Q3 by Reviewer VeR7; Q2 of Initial Review by Reviewer f7DN | In **Appendix F.1**, standardized the statement that OTEs “should be pairwise (nearly) orthogonal” and removed unsupported claims that strong non-orthogonality would not matter. | Aligns wording with empirical support and reviewer feedback while preserving the motivation for near-orthogonal OTEs in practice. |

---

### Author Response · Authors · 2025-12-02
**Revision Summary (1/2)**

We sincerely thank all reviewers for their detailed comments and follow-up questions. The table below summarizes, at a fine-grained level, how each concern has been addressed in the revised manuscript, along with the concrete changes and their expected impact.

| **Discussion Point** | **Corresponding Review** | **Our Revision** | **Result** |
| --- | --- | --- | --- |
| Clarifying that OSOM is **not** a fragile three-stage pipeline but a **single end-to-end model** | W1 of Initial Review by Reviewer H4db; Reply 1 by Reviewer H4db | In **Sec. 3**, explicitly described OSOM as a single end-to-end model with shared encoder, Transformer Identifier, and MLP Responder jointly trained with $\alpha_1 J_{\text{distill}} + \alpha_2 J_{\text{identify}} + \alpha_3 J_{\text{respond}}$; marked the distillation decoder as training-only and specified test-time components. | Clarifies that OSOM is not a brittle three-stage pipeline but a practical end-to-end RL agent with auxiliary heads. |
| Modularity of the three objectives and behavior when one component is hard to use (especially distillation) | W1 of Initial Review by Reviewer H4db; W3 of Initial Review by Reviewer VeR7 | In **Sec. 3.1** and **Sec. 5**, explained that setting $\alpha_1 = 0$ yields the “OSOM w/o Distill Loss” variant used in Fig. 5, and that OSOM remains valid when opponent traces are unavailable. | Positions distillation as an optional but beneficial auxiliary loss and shows OSOM degrades gracefully. |
| Scope and extension from “single controllable agent” to **team control** | W2 and Q1 of Initial Review by Reviewer H4db | In **Sec. 5**, added a conceptual discussion of centralized team control (joint controller for all self-agents) and CTDE-style decentralized control (one OSOM per controllable agent). | Makes clear OSOM is conceptually compatible with multi-agent/team control beyond a single self-agent. |
| Practical evidence that OSOM can **control a team** of agents | Reply 2 by Reviewer H4db | In **Appendix I**, added a PPW experiment where OSOM controls both predators via a centralized controller (“OSOM w/ Multi Self-Agents”), alongside “PACE w/ Multi Self-Agents” and single-self baselines. | Provides concrete empirical evidence that OSOM scales to team control in a 4-agent mixed game and still outperforms PACE. |
| Potential advantage from using a Transformer vs. GRU/MLP in baselines | W1 of Initial Review by Reviewer 24v5 | In **Sec. 4.1** and **Sec. 4.2 Q5**, introduced a Transformer-based PACE variant (PACE-TF) using the official PACE Transformer, and compared PACE, PACE-TF, and OSOM under the same protocol as Fig. 6. | Shows that simply swapping in a Transformer does not close the gap; OSOM’s gains stem from its open-set opponent modeling design. |
| Interpretation of opponent-identification accuracies and avoiding over-claiming | W2 of Initial Review by Reviewer 24v5 | Around Table 1 in **Sec. 4.2 Q3**, clarified that identification is strict top-1 over up to 40-50 types in open-set, non-stationary, partially observable conditions, and rephrased claims as “significantly above random 2–15$\times$” rather than “strong” in absolute terms. | Places accuracies in the proper context while emphasizing that OSOM is the only method achieving non-trivial explicit identification in this setting. |
| Choice of opponent-switch frequency $H$ and its impact | W3 and Q2 of Initial Review by Reviewer 24v5 | In **Sec. 4.1**, justified using $H=20$ (KP) and $H=5$ (POO/PPW) as moderate non-stationarity; in **Appendix I**, added a POO (Eval Set) ablation varying $H \in \\{1,3,5,10,20\\}$. | Demonstrates that OSOM’s advantages are robust across a wide range of non-stationarity levels and that the original $H$ choices are representative. |
| Relation to concurrent work LOSI | Q1 of Initial Review by Reviewer 24v5 | In **Appendix A**, added a paragraph contrasting LOSI and OSOM in terms of setting (fixed scripts vs. open-set unseen opponents), supervision (unsupervised vs. supervised OTEs), representation (global memory vs. per-prompt OTE codebook), and architecture (QMIX vs. generic policy). | Clarifies complementarity with LOSI and highlights OSOM’s unique open-set capability and potential future combinations. |
| Diversity and construction of Train/Eval opponent sets | Weakness and Q3 of Initial Review by Reviewer f7DN; Reply Q4 by Reviewer VeR7 | In **Sec. 4.1**, explicitly described how we construct semantically diverse and disjoint Train/Eval opponent sets in each environment, with full details in Appendix E. | Addresses concerns about diversity and clarifies how unseen opponents are generated and separated from training opponents. |

---

### Author Response · Authors · 2025-12-02
**Rebuttal Summary and Appreciation**

Dear Reviewers, ACs, SACs, and PCs,

We sincerely thank all of you for your time and effort during an unusually turbulent ICLR season. Despite the unexpected issues surrounding this year's review process, we genuinely appreciated the opportunity to engage in a deep technical dialogue. The rebuttal phase has already given us many new insights into open-set opponent modeling, multi-agent control, and how to present our contributions more clearly.

---

- **Reviewer VeR7 (from 2 → 6).** Reviewer VeR7 initially had strong reservations about our methodology and experimental design, especially around the Responder–Identifier “disconnect”, the role of $M$, and the scalability/orthogonality of OTEs. Through the rebuttal, we clarified the end-to-end nature of OSOM, reinterpreted $M$ and $d$ as capacity hyperparameters, refined our discussion of orthogonality, and committed to clearer plots (environment timesteps) and wording (policy-gradient optimization of $J_{\text{respond}})$. In response, the reviewer wrote that our rebuttal *“**convincingly addressed my concerns**”* and that they now find the method *“**novel and intuitive**”*, and explicitly stated *“**I will raise my score to 6 to reflect my current evaluation**.”*
- **Reviewer H4db (initial 4, asks for team-control evidence).** Reviewer H4db raised two central concerns: (i) the practicality and modularity of OSOM’s three components, and (ii) whether OSOM can *actually* control a team of agents rather than a single self-agent. After our detailed explanations, they noted *“I thank the authors for the detailed explanation. Please refine the manuscript to highlight this point.”* and further emphasized that *“if the authors can provide experiments to demonstrate that OSOM can indeed control a team of agents … **I am happy to consider raising the score**.”* We accordingly added a new PPW multi-self team-control experiment and an explicit discussion of centralized and CTDE-style extensions, and revised the paper to highlight the simplicity, scalability, and practicality of OSOM, which we believe resolves the reviewer’s outstanding doubts.
- **Reviewers 24v5 and f7DN (both initial 6).** Reviewers 24v5 and f7DN were consistently supportive from the outset, each giving an initial score of 6. Reviewer 24v5 summarized that OSOM addresses the problem of identifying and responding to unseen opponents across three benchmarks, and highlighted that *“**OSOM is intuitive and empirically effective**”* and that *“The paper is well written and clearly states and supports its claims with evidence.”* Reviewer f7DN similarly emphasized that *“**The design is novel**. The experimental results are comprehensive”* and described OSOM’s end-to-end architecture as *“**very amazing**.”* Although these two reviewers could not participate further during the discussion period, their initial reviews already recognized both the novelty and empirical strength of OSOM. Our rebuttal directly addressed their remaining technical questions (e.g., fairness of the sequence model comparison, details of the Transformer design and masking, construction of diverse opponent pools, and OTE-sampling choices) and led to substantial clarifications and additions in the revised manuscript.

---

Across all reviewers, we carefully followed every comment, ran additional experiments where possible (notably the new team-control PPW setting), and substantially revised the paper for clarity, completeness, and practical guidance—changes we have systematically summarized in our **Revision Summary**. These interactions have significantly improved the manuscript and, we believe, sharpened OSOM’s contribution as *a principled framework for explicit open-set opponent identification and response in multi-agent systems*. Regardless of the final decision, we deeply respect the committee’s judgment, and we sincerely hope ICLR can weather this year’s difficulties and continue to thrive as a leading venue for the RL and multi-agent learning communities.

We sincerely thank you once again for your rigorous and dedicated review process.

Best regards, All authors

---

### Meta-Review · Area_Chair_XNLu · 2026-01-06

**Summary:**

The paper proposes an open set opponent modelling approach to train agents that generalize better to never-seen opponents. The reviewers shared concerns, the majority of which appeared to be allayed, but some remained such as a full extension of OSOM to CTDE settings and experiments on more complex environments (e.g., SMAC, Hanabi).

**Reviewer Concerns:**

The reviewers held concerns about whether the proposed framework was truly end-to-end, the restriction to single-agent settings, lack of ablations / explanation of the importance of various design choices, missing environments (SMAC, Hanabai), as well as questions about the experimental setups (e.g., diversity of the initial opponent pool). Each of these was addressed by the authors in a comprehensive rebuttal although some may remain outstanding such as extensions to multi-agent settings: "Due to rebuttal-time constraints, we have not yet run a full CTDE experiment."

**Reviewer Scores:**

- H4db: This reviewer would likely raise their score from 4 to 5 or 6.
- 24v5: This reviewer might have kept their positive score at 6.
- f7DN: This reviewer might have kept their positive score at 6.
- VeR7: This reviewer raised their score from 2 to 6 after reading the rebuttal.

---

### Decision · Program_Chairs · 2026-01-26

Reject